# On the Value of Cross-Modal Misalignment in Multimodal Representation Learning

**Yichao Cai**[*]    **Yuhang Liu**[*†]    **Erdun Gao**    **Tianjiao Jiang**
**Zhen Zhang**    **Anton van den Hengel**    **Javen Qinfeng Shi**
Australian Institute for Machine Learning
The University of Adelaide, SA 5000, Australia

## Abstract

Multimodal representation learning, exemplified by multimodal contrastive learning (MMCL) using image-text pairs, aims to learn powerful representations by aligning cues across modalities. This approach relies on the core assumption that the exemplar image-text pairs constitute two representations of an identical concept. However, recent research has revealed that real-world datasets often exhibit *cross-modal misalignment*. There are two distinct viewpoints on how to address this issue: one suggests mitigating the misalignment, and the other leveraging it. We seek here to reconcile these seemingly opposing perspectives, and to provide a practical guide for practitioners. Using latent variable models we thus formalize cross-modal misalignment by introducing two specific mechanisms: *Selection bias*, where some semantic variables are absent in the text, and *perturbation bias*, where semantic variables are altered—both leading to misalignment in data pairs. Our theoretical analysis demonstrates that, under mild assumptions, the representations learned by MMCL capture exactly the information related to the subset of the semantic variables invariant to selection and perturbation biases. This provides a unified perspective for understanding misalignment. Based on this, we further offer actionable insights into how misalignment should inform the design of real-world ML systems. We validate our theoretical findings via extensive empirical studies on both synthetic data and real image-text datasets, shedding light on the nuanced impact of cross-modal misalignment on multimodal representation learning. [2]

## 1 Introduction

Modern multimodal learning has achieved remarkable success in jointly modeling information from heterogeneous sources such as vision, language, and audio. Multimodal contrastive learning (MMCL) on paired data has emerged as a dominant strategy for aligning modalities [55, 27, 72]. This approach is exemplified by vision-and-language models like CLIP, which attempt to identify a common representation space that maximizes the similarity of real image-text pairs while minimizing that of incorrect pairs [55]. One of the assumptions underpinning this approach is that the training pairs are aligned across modalities, meaning that they convey exactly the same semantic information [73, 45]. This assumption, though convenient, is often violated in real-world scenarios, where multimodal data is inherently noisy or imprecisely paired [49, 50]. More critically, text taken from image captions typically only partially describes the paired image content, and often contains descriptive elements that are irrelevant or misleading, a situation referred to as *cross-modal misalignment*. For example, in a large-scale video-text dataset, over 50% of the purportedly aligned clip-caption pairs were found to be misaligned [49]. It is worth noting that the impact of cross-modal misalignment is mainly

---

[*]Equal contribution. [†]Correspondence to: `yuhang.liu01@adelaide.edu.au`.

[2]The project page is available at `https://yichaocai.com/misalignment.github.io`

39th Conference on Neural Information Processing Systems (NeurIPS 2025).

concentrated in the text modality, as text captions, often derived from images, are more susceptible to semantic incompleteness and interpretive variability than the typically consistent visual semantics.

The misalignment[3] discussed above has led to two seemingly opposing viewpoints. On one hand, misalignment is viewed as a form of disruption that should be mitigated [40, 64, 72, 8, 65, 30, 76, 7, 47]. For example, cross-modal misalignment can result in "hallucination" in multimodal models [64, 35, 74]. It also provides weak, noisy, and even misleading supervision for multimodal pre-training [72, 82]. On the other hand, an alternative viewpoint suggests that multimodal representations may actually benefit from cross-modal misalignment [71, 50, 6, 29]. For instance, fine-tuning the representations learned by CLIP through random text augmentation, which deliberately introduces misalignment in style-related information, can lead to more robust representations for zero-shot learning, few-shot learning, and even adversarial attacks [6]. This contrast raises a crucial question:

*How can we theoretically reconcile these two opposing views on misalignment, and, more importantly, determine which should guide practical applications?*

In light of this, we offer a theoretical perspective that not only facilitates the understanding of misalignment but also provides guidance for real-world applications. Specifically, we formulate the problem using a latent variable model (LVM), which depicts the underlying generative process of image-text data with misalignment, as shown in Figure 1. In it, the latent space consists of latent semantic variables representing factors common to both modalities (e.g., object shapes and colors), along with modality-specific subspaces that capture unique variations in images and text. To make the LVM more adaptable to diverse real-world scenarios, we allow for an arbitrary causal structure among the latent semantic variables, providing flexibility in multimodal contexts. To model misalignment, we introduce two mechanisms: selection bias and perturbation bias. Both act on latent semantic information but differ in their effects. Selection bias determines which semantic information is preserved in the text. For example, when describing an object, the text might preserve information about its color ("black") but omit its texture details or shape. On the other hand, perturbation bias introduces errors, such as misannotating "black" (correct color) to "red" (incorrect color). Moreover, given their heterogeneity, we model the two modalities with separate generative processes.

Building on the proposed LVM (i.e., the *generative model*), we present a theoretical identifiability analysis within the MMCL framework (i.e., the *inference model*). We show, under mild assumptions, that the subset of semantic variables unaffected by selection and perturbation biases remain block-identifiable (Defn. 4.1)—that is, only the unaffected subset of semantic variables in the proposed LVM admits an invertible mapping to the representations learned by MMCL. In contrast, the remaining misaligned semantic variables affected by either bias are inherently excluded from the learned representations, irrespective of the latent causal structure among all semantic variables. This result provides a unified perspective on the seemingly opposing views discussed above. While misalignment can be problematic in tasks that rely on fully preserving semantic information to maximize downstream utility, it may paradoxically become beneficial in scenarios where robustness to distribution shifts is desired. In such cases, cross-modal misalignment naturally acts as a regularizer, encouraging models to focus on stable, invariant semantic factors shared across modalities.

**Contributions.** Our main contributions are presented below, with an in-depth discussion of related work available in App. B.

- We propose an LVM for multimodal data generation that explicitly characterizes cross-modal misalignment through two mechanisms: selection bias and perturbation bias (§ 3).

- We establish a general identifiability result, showing that MMCL recovers the subset of semantic variables unaffected by these biases, independent of the underlying latent causal structure (§ 4.1).

- We extend this result to two practical scenarios, tasks requiring common representations and those targeting invariant representations, offering actionable insights into how misalignment should inform real-world applications (§ 4.2).

- We empirically validate our theoretical findings through extensive experiments on both real-world and synthetic image-text datasets under various cross-modal misalignment conditions. Additionally, we present a case study on pretrained CLIP models (§ 5).

---

[3]For the sake of brevity, we use"cross-modal misalignment" and "misalignment" interchangeably throughout the paper, as the context clearly indicates that we are referring to misalignment between different modalities.

## 2 Preliminaries: Multimodal Contrastive Learning

Multimodal contrastive learning (MMCL) [55, 27, 81] aims to learn joint representations by aligning paired samples from different modalities, i.e., $\mathbf{t} \in \mathcal{T}$ for text and $\mathbf{x} \in \mathcal{X}$ for images, while pushing apart unpaired (negative) samples. In practice, MMCL typically employs two modality-specific encoders, i.e., $f_t(\mathbf{t})$ for text and $f_x(\mathbf{x})$ for images which project observed paired data into a shared representation space. The learning objective is generally formulated as the following contrastive loss:

$$\mathcal{L}_{\text{MMCL}}(f_x, f_t) = -\frac{1}{2K} \left[ \sum_{i=1}^{K} \log \frac{e^{\kappa(f_x(\mathbf{x}_i), f_t(\mathbf{t}_i))/\tau}}{\sum_{j=1}^{K} e^{\kappa(f_x(\mathbf{x}_i), f_t(\mathbf{t}_j))/\tau}} + \sum_{i=1}^{K} \log \frac{e^{\kappa(f_x(\mathbf{x}_i), f_t(\mathbf{t}_i))/\tau}}{\sum_{j=1}^{K} e^{\kappa(f_x(\mathbf{x}_j), f_t(\mathbf{t}_i))/\tau}} \right],$$
(1)

where $\{\mathbf{x}_i, \mathbf{t}_i\}_{i=1}^{K}$ are sampled paired data, $K$ denotes the number of training pairs, $\tau$ is a temperature hyperparameter controlling the concentration of the similarity distribution, and $\kappa(\cdot, \cdot)$ denotes a similarity measure. Asymptotically, when $K$ approaches infinity, and with $\tau = 1$ and the similarity function defined as the negative squared Euclidean distance, the objective in Eq. (1) reduces to [13]:

$$\mathcal{L}_{\text{SymAlignMaxEnt}}(f_x, f_t) = \mathbb{E}_{(\mathbf{x}, \mathbf{t}) \sim p_{\mathbf{x}, \mathbf{t}}} \left[ \|f_x(\mathbf{x}) - f_t(\mathbf{t})\|_2 \right] - \frac{1}{2} \Big( H\big(f_x(\mathbf{x})\big) + H\big(f_t(\mathbf{t})\big) \Big),$$
(2)

where $H(\cdot)$ denotes differential entropy [69, 68]. One of the main advantages of the asymptotic objective in Eq. (2) is its suitability for theoretical analysis [48, 13, 78]. At a high level, Eq. (2) naturally decomposes into two intuitive terms [69]: the first term encourages minimizing the distance between paired samples, while the second term promotes maximizing the entropy of learned representations. Following prior works [48, 13, 78], we also adopt this objective in our theoretical analysis.

A key feature of our work is its focus on the impact of *cross-modal misalignment*, in contrast to prior studies that assume perfectly aligned paired data. To model this, we introduce a novel latent variable model (§ 3) that defines a fundamentally different problem setting. Consequently, our theoretical analysis (§ 4) yields insights that diverge significantly from existing results, highlighting how misalignment influences the learned representations in multimodal contrastive learning.

## 3 Problem Formulation via a Generative Perspective

In this section, we introduce a novel latent variable model (LVM) to formalize the underlying generative processes of image-text data, explicitly characterizing cross-modal misalignment (§ 3.1). Building on this LVM, we introduce the technical assumptions underlying image-text pairs under misalignment for subsequent analysis (§ 3.2). See App. A for a complete notation table.

### 3.1 A Latent Variable Model Characterizing Cross-Modal Misalignment

Figure 1 illustrates the proposed LVM. In the following, we provide a detailed explanation of the model from three aspects: the latent space, and the image and text generation processes.

**Latent space.** We partition the entire latent space $\mathcal{Z}$ into three simply connected, open subspaces, i.e., $\mathcal{Z} = \mathcal{S} \times \mathcal{M}_x \times \mathcal{M}_t$, where each defines the support of a distinct group of latent variables. We denote the latent variables in $\mathcal{Z}$ as $\mathbf{z} = (\mathbf{s}, \mathbf{m}_x, \mathbf{m}_t)$, where $\mathbf{s}$, $\mathbf{m}_x$, and $\mathbf{m}_t$ lie in $\mathcal{S}$, $\mathcal{M}_x$, and $\mathcal{M}_t$, respectively. Below, we describe the characteristics of the latent variables $\mathbf{s}$, $\mathbf{m}_x$, and $\mathbf{m}_t$:

- $\mathbf{s} \in \mathcal{S} \subseteq \mathbb{R}^{n_s}$ (*Semantic variables*): Latent variables capturing the semantic content of the data, i.e., information that is interpretable or describable through human knowledge (e.g., object shape, color). We denote the index set of semantic variables as $\mathbb{I}_{\mathbf{s}} := [n_s]$[4] for future reference.

- $\mathbf{m}_x \in \mathcal{M}_x$[5] (*Image-specific variables*): Latent variables capturing non-semantic, image-specific factors that are independent of semantic variables $\mathbf{s}$ (e.g., camera noise or background artifacts).

- $\mathbf{m}_t \in \mathcal{M}_t$ (*Text-specific variables*): Latent variables capturing non-semantic, text-specific factors, independent of both semantic variables $\mathbf{s}$ and image-specific variables $\mathbf{m}_x$ (e.g., writing style or tone).

---

[4]Throughout this paper, we use the notation $[d]$ to represent the set $\{1, \ldots, d\}$ for any integer $d > 1$.
[5]For simplicity of notation, we omit the dimensions of certain variables throughout this work, such as $\mathbf{m}_x$.

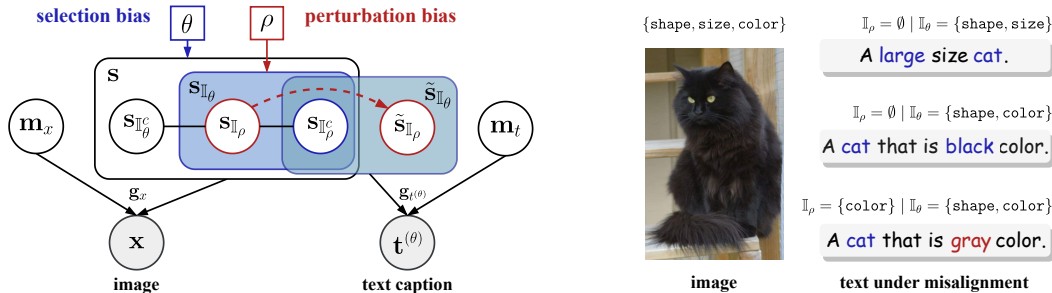

Figure 1: Illustration of the proposed latent variable model (left), with misalignment across modalities modeled via selection and perturbation bias. Image $\mathbf{x}$ is generated from semantic variables $\mathbf{s}$ and image-specific variables $\mathbf{m}_x$ via the generative process $g_x$. The corresponding text $\mathbf{t}^{(\theta)}$ is generated by $g_{t^{(\theta)}}$ which acts on a biased subset of semantic variables $\tilde{\mathbf{s}}_{\mathbb{I}_\theta} = (\mathbf{s}_{\mathbb{I}_\rho^c}, \tilde{\mathbf{s}}_{\mathbb{I}_\rho})$, influenced by selection bias $\theta$ and perturbation bias $\rho$, along with text-specific variables $\mathbf{m}_t$. Selection bias omits $\mathbf{s}_{\mathbb{I}_\theta^c}$, while perturbation bias marks a subset $\mathbb{I}_\rho \subseteq \mathbb{I}_\theta$ whose components may be randomly replaced to form $\tilde{\mathbf{s}}_{\mathbb{I}_\rho}$. Example image-text pairs (right) illustrate the misalignment induced by these two biases.

A key advantage of the proposed latent space structure is its flexibility and applicability to real-world scenarios. To this end, we depart from prior works that impose somewhat restrictive assumptions on the latent structure. Specifically, unlike approaches that enforce certain fixed graphical structures among semantic variables (e.g., assuming content causally determines style) [13], or methods based on nonlinear ICA that assume complete (or conditional on an auxiliary variable) independence among latent variables [28, 63], we allow for arbitrary dependency structures among semantic variables $\mathbf{s}$, since the true latent graph structure among semantic variables is often unknown in practice.

**Image generation.** An image $\mathbf{x} \in \mathcal{X}$ is generated from latent variables $\mathbf{z}_x = (\mathbf{s}, \mathbf{m}_x)$ via a diffeomorphism (i.e., a bijection with smooth inverse function) $g_x : \mathcal{S} \times \mathcal{M}_x \to \mathcal{X}$:

$$\mathbf{s} \sim p_{\mathbf{s}}, \quad \mathbf{m}_x \sim p_{\mathbf{m}_x}, \quad \mathbf{z}_x = (\mathbf{s}, \mathbf{m}_x), \quad \mathbf{x} = g_x(\mathbf{z}_x). \tag{3}$$

Here, $p_{\mathbf{s}}$ and $p_{\mathbf{m}_x}$ are prior distributions over subspaces $\mathcal{S}$ and $\mathcal{M}_x$, respectively; $\mathcal{X}$ defines an observation space that contains all generated images. This generative process formalizes that images fully encapsulate semantics, reflecting the fact that they are both informative and semantically rich.

**Text generation.** Unlike other multimodal data (e.g., camera-LiDAR [37]) acquired via sensors, given an image, its corresponding text inherently exhibits flexibility in semantic richness and may also include distortions introduced by humans or captioning models [36, 75], leading to cross-modal misalignment. Here, we formalize the root of such misalignment by introducing two types of biases: selection bias and perturbation bias. Accordingly, before formulating the text generation process, we first provide definitions of these two types of biases.

**Definition 3.1** (Selection bias $\theta$). Let $\mathcal{P}^+(\mathbb{I}_{\mathbf{s}})$ denote the set of all non-empty subsets[6] of the index set $\mathbb{I}_{\mathbf{s}}$, defined as $\mathcal{P}^+(\mathbb{I}_{\mathbf{s}}) := \mathcal{P}(\mathbb{I}_{\mathbf{s}}) \setminus \{\emptyset\}$, where $\mathcal{P}(\cdot)$ denotes the power set. The *selection bias* $\theta$ is defined as an integer index in the range $[2^{n_s} - 1]$, corresponding to a specific non-empty semantic subset $\mathbb{I}_\theta \in \mathcal{P}^+(\mathbb{I}_{\mathbf{s}})$. The complement $\mathbb{I}_\theta^c = \mathbb{I}_{\mathbf{s}} \setminus \mathbb{I}_\theta$ denotes the omitted semantic subset.

Note that each $\theta$ uniquely determines a non-empty semantic subset $\mathbb{I}_\theta \in \mathcal{P}^+(\mathbb{I}_{\mathbf{s}})$, which defines the semantic information to be expressed in the generated text. It also specifies a text generation mapping $g_{t^{(\theta)}} : \mathcal{S}_{\mathbb{I}_\theta} \times \mathcal{M}_t \to \mathcal{T}^{(\theta)}$, selected from a class of diffeomorphisms $\mathcal{G}_t$. Here, $\mathcal{T}^{(\theta)}$ denotes an observation space that contains all the observed text $\mathbf{t}^{(\theta)}$ under selection bias $\theta$.

**Definition 3.2** (Perturbation bias $\rho$). Let $\mathcal{P}_{\text{prop}}(\mathbb{I}_\theta) := \mathcal{P}(\mathbb{I}_\theta) \setminus \{\mathbb{I}_\theta\}$ denote the set of all proper subsets[7] of the selected index subset $\mathbb{I}_\theta$. The *perturbation bias* $\rho$ is defined as an integer index in the range $\rho \in [2^{|\mathbb{I}_\theta|} - 1]$, corresponding to a unique subset $\mathbb{I}_\rho \in \mathcal{P}_{\text{prop}}(\mathbb{I}_\theta)$ subject to perturbation. The complement $\mathbb{I}_\rho^c := \mathbb{I}_\theta \setminus \mathbb{I}_\rho$ denotes the semantic subset that remains unperturbed across modalities.

---

[6] Without loss of generality, we fix a graded lexicographic order (see Defn. C.1) over $\mathcal{P}^+(\mathbb{I}_{\mathbf{s}})$. See Clar. C.2 for a clarification for this choice of order.

[7] Again, we fix a graded lexicographic order over the proper subsets in $\mathcal{P}_{\text{prop}}(\mathbb{I}_\theta)$ throughout the paper.

**Example 3.1.** Let the full semantic index set be $\mathbb{I}_{\mathbf{s}} = \{\texttt{shape}, \texttt{size}, \texttt{color}\}$, so that the set of non-empty semantic subsets is $\mathcal{P}^+(\mathbb{I}_{\mathbf{s}}) = \{\{\texttt{shape}\}, \{\texttt{size}\}, \dots, \{\texttt{shape}, \texttt{size}, \texttt{color}\}\}$. Then, a selection bias $\theta = 5$ corresponds to the fifth subset, i.e., $\mathbb{I}_\theta = \{\texttt{shape}, \texttt{color}\}$, with the omitted semantics given by $\mathbb{I}_\theta^c = \{\texttt{size}\}$. The corresponding text-generation mapping $g_{t^{(\theta)}}$ uses only the selected semantic variables in $\mathbb{I}_\theta$ to generate text $\mathbf{t}^{(\theta)}$. The set of proper subsets of $\mathbb{I}_\theta$ is $\mathcal{P}_{\text{prop}}(\mathbb{I}_\theta) = \{\emptyset, \{\texttt{shape}\}, \{\texttt{color}\}\}$. Then, a perturbation bias $\rho = 3$ corresponds to the subset $\mathbb{I}_\rho = \{\texttt{color}\}$, and its complement within $\mathbb{I}_\theta$ is $\mathbb{I}_\rho^c = \{\texttt{shape}\}$. Under the combined biases $\theta = 5$ and $\rho = 3$, only shape is unbiasedly preserved, while color is subject to perturbation. A resulting text might be *"A cat that is red color"*, even though the image shows a large-sized black cat.

Building upon the previously defined selection bias $\theta$ and perturbation bias $\rho$, we now formalize the text generation process, explicitly characterizing the misalignment induced by these two biases. Consider an image $\mathbf{x}$ generated by Eq. (3), with associated semantic variables $\mathbf{s} = (\mathbf{s}_{\mathbb{I}_\theta}, \mathbf{s}_{\mathbb{I}_\theta^c})$, where the index set $\mathbb{I}_\theta$ is determined by $\theta$. For the corresponding text $\mathbf{t}^{(\theta)}$, we define the latent variables as $\mathbf{z}_{t^{(\theta)}} = (\tilde{\mathbf{s}}_{\mathbb{I}_\theta}, \mathbf{m}_t)$, where $\tilde{\mathbf{s}}_{\mathbb{I}_\theta}$ represents the perturbed semantic variables under perturbation bias $\rho$, and $\mathbf{m}_t$ denotes the text-specific latent variables. The text generation process is then formalized as:

$$\tilde{\mathbf{s}}_{\mathbb{I}_\theta} \sim p_{\tilde{\mathbf{s}}_{\mathbb{I}_\theta} | \mathbf{s}, \theta, \rho}, \quad \mathbf{m}_t \sim p_{\mathbf{m}_t}, \quad \mathbf{z}_{t^{(\theta)}} = (\tilde{\mathbf{s}}_{\mathbb{I}_\theta}, \mathbf{m}_t), \quad \mathbf{t}^{(\theta)} = g_{t^{(\theta)}}(\mathbf{z}_{t^{(\theta)}}), \tag{4}$$

where $p_{\mathbf{m}_t}$ denotes the prior distribution over the latent subspace $\mathcal{M}_t$, and $g_{t^{(\theta)}}$ is the diffeomorphic mapping specified by the selection bias $\theta$. The conditional $p_{\tilde{\mathbf{s}}_{\mathbb{I}_\theta} | \mathbf{s}, \theta, \rho}$ explicitly characterizes misalignment in text generation arises through perturbations within selected semantic dimensions.

## 3.2 Model Assumptions for Theoretical Analysis

Based on the proposed LVM, we now present the assumptions underpinning our theoretical analysis:

**Assumption 3.1** (Continuous positive densities). *The latent variables $\mathbf{s}$, $\mathbf{m}_x$, and $\mathbf{m}_t$ are continuous and admit strictly positive densities, i.e., $p_{\mathbf{s}} > 0$, $p_{\mathbf{m}_x} > 0$, and $p_{\mathbf{m}_t} > 0$, almost everywhere (a.e.) on their respective supports $\mathcal{S}$, $\mathcal{M}_x$, and $\mathcal{M}_t$.*

**Assumption 3.2** (Random perturbations). *Given a selection bias $\theta$ and a perturbation bias $\rho$, consider a specific image-text pair $(\mathbf{x}, \mathbf{t}^{(\theta)})$ generated by Eq. (3) and Eq. (4), respectively. The conditional distribution $p_{\tilde{\mathbf{s}}_{\mathbb{I}_\theta} | \mathbf{s}, \theta, \rho}$ is defined via a randomly sampled perturbation subset $A \subseteq \mathbb{I}_\rho$, such that:*

$$A \sim p_A, \quad p_{\tilde{\mathbf{s}}_{\mathbb{I}_\theta} | \mathbf{s}, \theta, \rho}(\tilde{\mathbf{s}}_{\mathbb{I}_\theta} \mid \mathbf{s}, A) = \delta(\tilde{\mathbf{s}}_{\mathbb{I}_\theta \setminus A} - \mathbf{s}_{\mathbb{I}_\theta \setminus A}) \cdot p_{\tilde{\mathbf{s}}_A | \mathbf{s}_A}(\tilde{\mathbf{s}}_A \mid \mathbf{s}_A). \tag{5}$$

*Here: (i) $A$ is the random subset of semantic indices to be perturbed, with $p_A$ defined over $\mathcal{P}(\mathbb{I}_\rho)$. For every $l \in \mathbb{I}_\rho$, there exists at least one subset $A \subseteq \mathbb{I}_\rho$ such that $l \in A$ and $p_A(A) > 0$; (ii) $\delta(\cdot)$ denotes the Dirac delta function, enforcing that variables outside $A$ remain unchanged; (iii) $p_{\tilde{\mathbf{s}}_A | \mathbf{s}_A}$ is a smooth, strictly positive conditional density over $\mathcal{S}_A \times \mathcal{S}_A$, where $\mathcal{S}_A$ is the domain of $\mathbf{s}_A$, and for each $\mathbf{s}_A$, the support of $p_{\tilde{\mathbf{s}}_A | \mathbf{s}_A}$ includes a non-empty open subset $\mathcal{O}_A \subseteq \mathcal{S}_A$.*

**Remark 3.1.** Note that Eq. (5) essentially implies that, in each $(\mathbf{x}, \mathbf{t}^{(\theta)})$, only a random subset $A$ of $\mathbb{I}_\rho$ undergoes perturbations, regardless of the underlying causal structure among latent semantic variables. The rationale is that latent semantic variables can only be modified indirectly by altering observations, rather than through direct intervention, unless the latent causal structure is fully identified. In the text modality, it occurs through the misassignment of certain content words to specific image semantics during the captioning process. Unlike a direct intervention, this cross-modal misalignment does not propagate to descendant semantic variables, as illustrated in Figure 2.

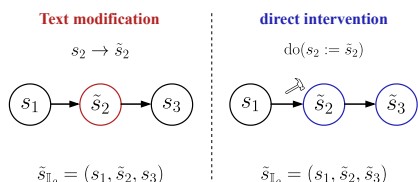

Figure 2: Text modifications affect only the semantics where content words are altered, while direct interventions act on latent semantic variables thereby propagating structural changes.

## 4 Identifiability Results for MMCL under Misalignment

In this section, we theoretically analyze how cross-modal misalignment impacts the identifiability of latent semantic variables in the proposed LVM, within the MMCL framework (§ 4.1). Based on

these results, we further provide practical insights into how misalignment should be addressed in real-world applications (§ 4.2). Detailed proofs of the theoretical results are provided in App. C. To begin, we restate the definition of *block-identifiability* [68] in the context of our problem setting:

**Definition 4.1** (Block-Identifiability). A subset of latent semantic variables $\mathbf{s}_{\mathbb{I}^*} \in \mathcal{S}_{\mathbb{I}^*}$, with $\mathbb{I}^* \subseteq \mathbb{I}_{\mathbf{s}}$, is said to be *block-identified* by functions $f_x : \mathcal{X} \to \mathbb{R}^{|\mathbb{I}^*|}$ and $f_t : \mathcal{T}^{(\theta)} \to \mathbb{R}^{|\mathbb{I}^*|}$ if the learned representations $\hat{\mathbf{z}}_x \in \mathbb{R}^{|\mathbb{I}^*|}$ and $\hat{\mathbf{z}}_t \in \mathbb{R}^{|\mathbb{I}^*|}$ retain *all* and *only* the information contained in $\mathbf{s}_{\mathbb{I}^*}$. Formally, there exist invertible mappings $h_x, h_t : \mathcal{S}_{\mathbb{I}^*} \to \mathbb{R}^{|\mathbb{I}^*|}$ s.t. $\hat{\mathbf{z}}_x = h_x(\mathbf{s}_{\mathbb{I}^*})$ and $\hat{\mathbf{z}}_t = h_t(\mathbf{s}_{\mathbb{I}^*})$.

## 4.1 Identifiability of Latent Semantic Variables under Cross-Modal Misalignment

Building on the definition above, the formalization of the proposed LVM, and assumptions outlined in § 3, we provide the following identifiability result:

**Theorem 4.1** (Identifiability of latent semantic variables). *Let $(\mathbf{x}, \mathbf{t}^{(\theta)})$ be image-text pairs drawn from the data-generating process described in § 3, where $\mathbf{x}$ is generated according to Eq. (3) and $\mathbf{t}^{(\theta)}$ is generated by Eq. (4). Further, suppose that Asms. 3.1 and 3.2 hold. Denote by $\mathbf{s}_{\mathbb{I}_\rho^c}$ the subset of semantic variables that annotated without bias in the text, and define its dimension as $n = |\mathbb{I}_\theta| - |\mathbb{I}_\rho|$. Let $f_x : \mathcal{X} \to (0,1)^n$ and $f_t : \mathcal{T}^{(\theta)} \to (0,1)^n$ be sufficiently flexible, smooth functions. Then, minimizing the loss $\mathcal{L}_{SymAlignMaxEnt}$ in Eq. (2) over samples $(\mathbf{x}, \mathbf{t}^{(\theta)})$ guarantees that $f_x$ and $f_t$ block-identify the semantic variables $\mathbf{s}_{\mathbb{I}_\rho^c}$ in the sense of Defn. 4.1.*

**Remark 4.1.** Thm. 4.1 formally establishes that, in the presence of cross-modal misalignment, the unbiased semantic variables $\mathbf{s}_{\mathbb{I}_\rho^c}$ that are shared across modalities can be effectively recovered, up to a block-wise indeterminacy, by minimizing the MMCL objective. In contrast, components that are misaligned, specifically $\mathbf{s}_{\mathbb{I}_\rho}$ and $\mathbf{s}_{\mathbb{I}_\theta^c}$, are entirely excluded from the learned representations. We emphasize that this result holds regardless of any underlying latent graph structure among semantic variables. At its core, this result highlights the model's capacity to focus exclusively on the aligned semantic aspects of the data. Furthermore, modality-specific, non-semantic factors, i.e., $\mathbf{m}_x$ and $\mathbf{m}_t$, are consistently discarded throughout the learning process. This further underscores the model's ability to extract meaningful, cross-modal semantic information while filtering out semantically uninformative or noise-induced components. A detailed proof of Thm. 4.1 is provided in App. C.2.

## 4.2 Insights into Cross-Modal Misalignment for Practice

The above result is general and not limited to any specific problem context. We now consider two real-world application scenarios: **(i)** pretraining with large-scale data and **(ii)** invariant representation learning. The former aims to capture comprehensive semantic information to support a wide range of downstream tasks, while the latter focuses on learning robust representations for out-of-distribution (OOD) generalization. In what follows, we present corollaries and insights for both scenarios.

**Corollary 4.1** (Identifiability of full latent semantic variables). *Assume that Asms. 3.1 and 3.2 hold. Let the selection bias be $\theta = 2^{n_s} - 1$ and the perturbation bias be $\rho = 1$, such that the full set of semantic variables $\mathbb{I}_{\mathbf{s}}$ is selected, and the perturbable semantic subset is trivial, i.e., $\mathbb{I}_\rho = \emptyset$. Then, all semantic variables $\mathbf{s}$ are block-identified via smooth functions $f_x : \mathcal{X} \to (0,1)^{n_s}$ and $f_t : \mathcal{T}^{(\theta)} \to (0,1)^{n_s}$, when minimizing $\mathcal{L}_{SymAlignMaxEnt}$.*

**Insight 4.1** (MMCL pretraining on large-scale data). Large-scale multimodal datasets (e.g., COCO [38], Conceptual Captions [62], LAION-5B [61]) often exhibit varying caption quality. Our analysis indicates that omitted or perturbed semantics are irretrievably lost in the learned representations, regardless of dataset size, although scale may mitigate sporadic misalignment by averaging its effects. Preserving a breadth of relevant semantic details is therefore crucial when pretraining foundation models, whose primary goal is to support diverse downstream tasks. As noted in Cor. 4.1, achieving this requires detailed and consistent annotation of image semantics. Consequently, improved caption control [36, 16, 14] is essential to avoid blind spots in semantic coverage.

**Corollary 4.2** (Identifiability of invariant semantic variables). *Assume that Asms. 3.1 and 3.2 hold. Consider an OOD setting in which a subset of semantic variables, $\mathbb{I}_{inv} \subset \mathbb{I}_{\mathbf{s}}$, remains invariant between training and testing environments, while the remaining semantic variables, $\mathbb{I}_{var} = \mathbb{I}_{\mathbf{s}} \setminus \mathbb{I}_{inv}$, undergo distribution shifts. If the union of omitted and perturbable semantic variables under selection bias $\theta$ and perturbation bias $\rho$ coincides with the environment-sensitive subset, i.e., $\mathbb{I}_{var} = \mathbb{I}_\theta^c \cup \mathbb{I}_\rho$,*

then the invariant semantic variables $\mathbf{s}_{\mathbb{I}_{inv}}$ are block-identified via smooth functions $f_x : \mathcal{X} \to (0,1)^{|\mathbb{I}_{inv}|}$ and $f_t : \mathcal{T}^{(\theta)} \to (0,1)^{|\mathbb{I}_{inv}|}$, by minimizing $\mathcal{L}_{SymAlignMaxEnt}$.

**Insight 4.2** (Invariant representation learning). In tasks requiring robust OOD performance (e.g., domain generalization) [57, 2], semantic variables that are vulnerable to distribution shifts can undermine generalization. As noted in Cor. 4.2, misalignment may, counterintuitively, enhance robustness by selectively omitting or perturbing these vulnerable variables. This suggests that MMCL may offer a novel perspective on invariant representation learning [54, 32, 15], as auditing and curating text is more precise and interpretable, since language is distilled from human knowledge.

## 5 Experiments

We conduct extensive experiments to validate our theoretical results, including numerical simulations (§ 5.1), a real-world image-text dataset with independent semantic variables (§ 5.2), a synthetic dataset with dependent semantic variables. (§ 5.3), and a case study with OpenCLIP models (§ 5.4).

### 5.1 Numerical Simulation

**Experimental setup.** We simulate data following the generative process in § 3. Semantic variables $\mathbf{s} \sim \mathcal{N}(0, \Sigma_{\mathbf{s}})$ (dim. 10) and modality-specific variables $\mathbf{m}_x \sim \mathcal{N}(0, \Sigma_{\mathbf{m}_x})$, $\mathbf{m}_t \sim \mathcal{N}(0, \Sigma_{\mathbf{m}_t})$ (dim. 5) encode potential causal dependencies via their respective covariances $\Sigma_{(\cdot)}$. To study cross-modal misalignment, we independently vary: **(i)** Selection bias: controlling $\mathbb{I}_\theta = \{1\}, \ldots, [10]$, and **(ii)** Perturbation bias: defining $\mathbb{I}_\rho = \emptyset, \ldots, [9]$, with Gaussian noise added to each $i \in \mathbb{I}_\rho$ with probability 0.75. In isolation, we fix $\mathbb{I}_\rho = \emptyset$ when varying $\mathbb{I}_\theta$, and $\mathbb{I}_\theta = [10]$ when varying $\mathbb{I}_\rho$. Image and text data are generated using randomly initialized invertible MLPs $g_x$ and $g_{t^\theta}$. We train two modality-specific MLP encoders for 100,000 steps using the $\mathcal{L}_{SymAlignMaxEnt}$ loss in Eq. (2), with embedding dimension matching the number of unbiased semantic components. See App. D.1 for further details.

We conduct two experiments. **(i)** Semantic identification: A lightweight MLP regressor is trained to recover each ground-truth semantic component from MMCL learned representations; we report mean $R^2$ over three seeds. **(ii)** Downstream performance: We evaluate learned representations on four tasks with targets $y_1, y_2, y_3, y_4$, defined as nonlinear functions over semantic subsets [3], [5], [7], [9]. We additionally binarize $y_2$ for classification, and apply a heavy-tailed shift to dimensions $\{9, 10\}$ of semantic variables to evaluate OOD generalization. See App. D.1 for full task design.

**Identification of semantics.** The results in Figure 3 show that, under the independent latent variable scenario, unbiased semantic variables are clearly block-identified ($R^2 \approx 1$), whereas misaligned semantics due to selection bias are effectively discarded ($R^2 \approx 0$). In the dependent latent variable scenario, some misaligned semantics become partially predictable, reflecting inherent mutual predictability among strongly dependent variables [68, 78]. Modality-specific variables are consistently omitted from the representations across all settings. Similar effects are observed under perturbation bias, as illustrated in Figure 11. Notably, although our theoretical results hold only up to invertible mappings, simple linear regression already achieves high $R^2$ scores, as demonstrated in Figure 12. These findings consolidate the identifiability results in Thm. 4.1. Additional analyses on misassigned encoding dimensions and combined bias effects, are provided in App. D.2.

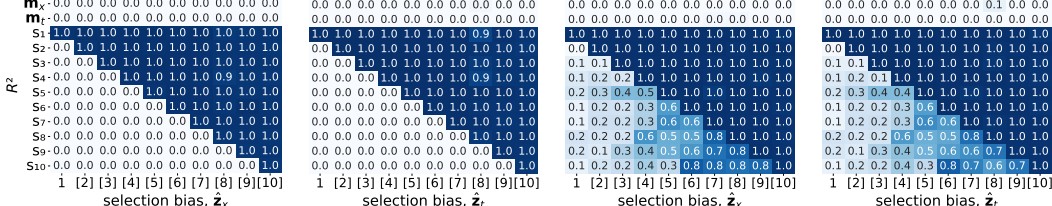

Figure 3: Mean $R^2$ scores under selection bias settings. From left to right: predictions based on $\hat{\mathbf{z}}_x$ and $\hat{\mathbf{z}}_t$ with **independent** latent semantics, followed by those with **dependent** latent semantics.

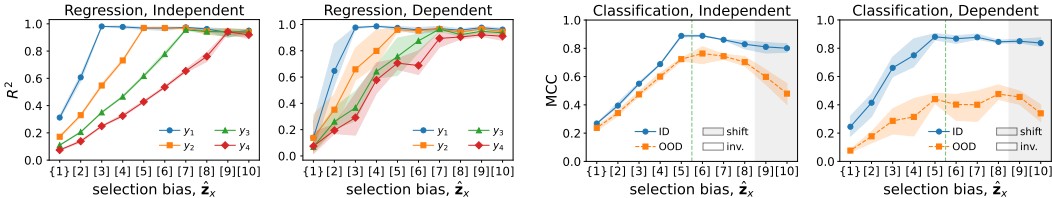

Figure 4: Downstream performance of pretrained $\hat{\mathbf{z}}_x$ under selection bias. Left: in-distribution (ID) regression performance. Right: ID classification and out-of-distribution (OOD) generalization.

**Downstream performance.** As shown in Figure 4, retaining more semantic information during pretraining significantly enhances in-distribution regression performance, consistent with Cor. 4.1. Conversely, under distribution shift scenarios, accurately identifying invariant semantic variables is essential for robust out-of-distribution generalization. In this setting, introducing appropriate selection or perturbation biases effectively removes variables sensitive to distribution shift, supporting the result in Cor. 4.2. Additional results on the effects of perturbation bias are provided in App. D.3.

## 5.2 MPI3D-Complex: Real-World Dataset with Factorized Latent Variables

**Experimental setup.** The MPI3D-Complex dataset [19] contains real-world images annotated with seven mutually independent, discrete ground-truth factors: object color (`color`), shape (`shape`), size (`size`), camera height (`cam.`), background color (`back.`), horizontal position (`hori.`), and vertical position (`vert.`). We treat `hori.` and `vert.` as image-specific factors, and the remaining five as semantic variables. See Figure 5 for example images from the MPI3D-Complex dataset. Text is generated from the ground-truth

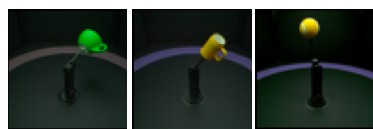

Figure 5: MPI3D-Comp. samples.

semantics using content-word mappings and three manually designed templates. To simulate misalignment, we introduce: **(i)** selection bias by progressively increasing the subset $\mathbb{I}_\theta$ of included semantic indices: ①: `color`, ②: {`color`, `shape`}, ③: {`color`, `shape`, `size`}, ④: {`color`, `shape`, `size`, `cam.`}, ⑤: all five; **(ii)** perturbation bias by fixing $\mathbb{I}_\theta = \{$`color`, `shape`, `size`, `cam.`, `back.`$\}$ and varying $\mathbb{I}_\rho$ from ①: $\emptyset$ to ⑤: {`shape`, `size`, `cam.`, `back.`} in reverse order, replacing selected values with alternatives at $90\%$ probability.

Training is performed using the loss $\mathcal{L}_{\text{MMCL}}$ defined in Eq. (1), following the formulation in [13]. Performance is evaluated using the average Matthews Correlation Coefficient (MCC) across three random seeds, based on the prediction of all image latent variables from the learned image and text representations using linear and nonlinear MLP decoders, respectively. Full details on dataset attributes, text generation, bias configurations, and architectures are provided in App. E.1.

**Results.** Figure 6 presents the nonlinear MCC results with learned representations. The findings demonstrate that, even with discrete latent variables, misaligned semantic variables across modalities, whether due to selection or perturbation biases, are consistently excluded from the representations (MCC = 0). In contrast, unbiased semantics are well recovered, with MCC scores predominantly approaching 1 and all values $\geq 0.8$, reinforcing our theoretical findings. Further investigations into MCC using linear classifiers and ablation studies on encoder dimensionality are provided in App. E.2.

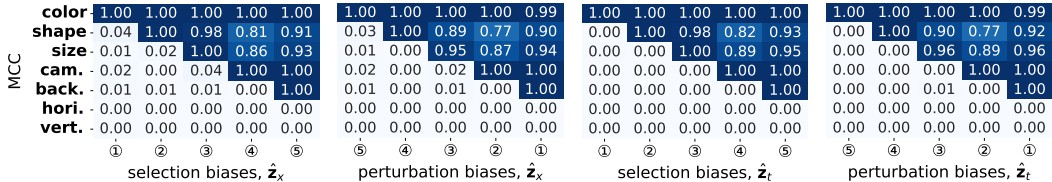

Figure 6: Mean MCC scores under misalignment settings. Left to right: Image features $\hat{\mathbf{z}}_x$ with selection and perturbation bias settings, text features $\hat{\mathbf{z}}_t$ under the same bias settings.

## 5.3 Causal3DIdent: Semi-Synthetic Dataset with Structured Causal Latent Variables

**Experimental setup.** We further conduct experiments on the *Causal3DIdent* dataset [83, 68, 13, 78], a semi-synthetic dataset allowing explicitly defined causal relations among latent variables. Images are generated from 10 latent variables: 3 discrete (shape, x_pos, y_pos) and 7 continuous (color, s_pos, s_color, b_color, alpha, beta, gamma). We treat the rotation angles (alpha, beta, gamma) as image-specific,

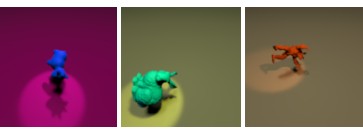

Figure 7: Causal3DIdent samples.

and the remaining as semantic variables following a causal graph among latent variables (Figure 17). See Figure 7 for example images from the dataset. For text, we discretize color-based attributes (color, s_color, b_color), retain continuous s_pos and simulate partial loss in spotlight information when generating text. Text descriptions are rendered using five templates, based on latent semantic variables under selection and perturbation bias. Specifically, selection bias settings ($\mathbb{I}_\theta$) vary from {shape} to the set of all seven semantic indices, with $\mathbb{I}_\rho = \emptyset$; perturbation bias settings progressively perturb subsets $\mathbb{I}_\rho$ from $\emptyset$ to {x_pos, y_pos, s_pos, color, s_color, b_color} in reverse order, with full semantic selection.

Training paradigm mirrors that of MPI3D-Complex. We evaluate using $R^2$ for continuous and MCC for discrete latent variables, averaged over three seeds. See App. F.1 for full experimental details.

**Results.** Figure 8 presents the prediction performance of a nonlinear MLP classifier or regressor trained on learned image representations. We observe that unbiased semantic variables shared across modalities, whether continuous (e.g., s_pos) or discrete (e.g., shape, x_pos, y_pos), are reliably recovered by MMCL across all settings, with predictive performance approaching perfect ($R^2 \approx 1$). For semantic variables that are continuous in the image modality but discretized in the text modality, prediction performance shows some degradation (e.g., s_color in selection setting ⑥ or perturbation setting ②), yet still achieves relatively high $R^2$ scores. Image-specific variables are consistently excluded from the learned representations, as indicated by $R^2 = 0$. Likewise, semantic variables omitted due to selection or perturbation bias are generally discarded. For instance, in selection setting ① or perturbation setting ⑦, only shape remains predictable. When factors such as x_pos or y_pos are included, other dependent semantic variables, such as color, become partially predictable (e.g., in selection setting ②). Similarly, identification of s_pos enhances predictability of s_color and b_color, reflecting the latent causal structure. Overall, the results on the Causal3DIdent dataset further support our theoretical findings. Analyses of the text representations are provided in App. F.2.

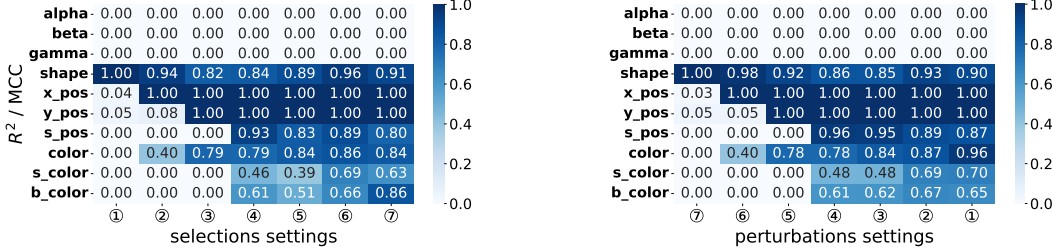

Figure 8: Predicting semantic variables under misalignment using image features. $R^2$ is reported for continuous factors and MCC for discrete factors. The predictions pattern align with our theoretical results, though extra dimensions may be predictable due to statistical correlations.

### 5.4 Case Study: Zero-Shot Evaluation of OpenCLIP Model Representations

We further validate our theoretical findings through a comprehensive zero-shot evaluation case study on OpenCLIP models [24], a foundation MMCL model pretrained on the LAION-400M dataset [61]. To achieve this, we develop a structured taxonomy comprising 146 visual concepts, organized into 15 distinct concept groups with corresponding abbreviations for clarity in plots and analyses: Animal, Clothing, Color, Food, Object, Role (i.e., human roles), Scene, Vehicle, Weather, Texture, POV (i.e., point of view of the photo), Emot. (i.e., emotion or mental state), Postproc. (i.e., post-processing), Trait (i.e., trait judgment by appearance), and Stere. (i.e., stereotypes). For each concept, we collect a dataset of up to 200 images from Flickr[8] by prompting with concept names, available under CC licenses. See the concept taxonomy and data collection details in App. G.1.

---

[8] https://www.flickr.com/

**Concept coverage in LAION-400M captions.** We begin by analyzing the concept coverage of LAION-400M captions, measuring the average coverage rate, defined as the percentage of captions that explicitly mention each concept or its synonyms, averaged by concept group. As shown in Figure 9, concept groups display substantial variation in average coverage: common concepts exhibit relatively high coverage (e.g., `Object`: 1.63%, `Color`: 2.16%), while some concepts are rarely mentioned (e.g., `Stere.`: 0.0003%, `Postproc.`: 0.0118%). Importantly, due to the large scale of the dataset, no concept group exhibited 0% coverage, indicating that complete omission by selection bias is unlikely. Thus, the coverage rate serves as an approximate indicator of selection bias across concept groups.

**Zero-shot evaluation of OpenCLIP models.** Figure 10 shows the zero-shot F1 performance of OpenCLIP across concept groups. Consistent with our theoretical findings, OpenCLIP models, regardless of scale, perform well on frequently captioned groups (e.g., `Animal`, `Object`), but their performance declines sharply for under-captioned groups (e.g., `Trait`, `Emot.`). Concept-level confusion matrices further reveal systematic misclassifications, particularly within low-coverage groups (see App. G.3). Notably, omitting captions for sensitive concepts such as `Stere.` may be intentional to avoid biased knowledge. In contrast, under-captioned yet valuable concepts like `Texture` and `Emot.` should be learned, suggesting a need for more targeted captioning strategies.

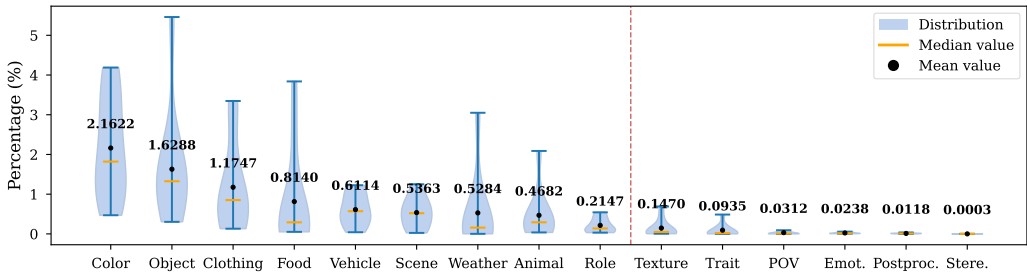

Figure 9: Concept coverage by group in LAION-400M captions. See App. G.2 for intra-group details.

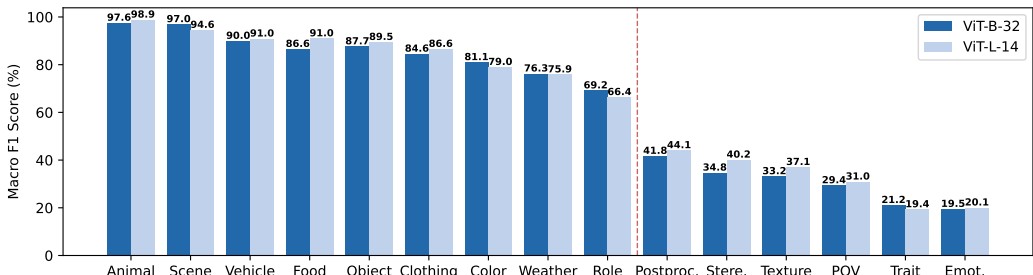

Figure 10: Zero-shot F1 score of OpenCLIP model trained on LAION-400M in each concept group.

## 6   Conclusion

In this work, we present a formal analysis of cross-modal misalignment in multimodal data pairs within the MMCL framework, examining its impact on the learned representations. We demonstrate that contrastive multimodal encoders retain only the unbiased shared semantic variables, while discarding misaligned latent variables. When image-text pairs exhibit misalignment due to selection or perturbation biases, the joint embedding prioritizes consistent content, while omitting altered or missing aspects. This trade-off is fundamental: perfectly aligned text captions preserve rich semantic detail, whereas selective or biased text can enhance domain invariance by filtering out distribution-sensitive factors. Our experiments, conducted across simulations and image-text datasets, empirically validate these theoretical findings. These insights highlight the need for multimodal learning frameworks that mitigate misalignment or leverage beneficial biases to enhance representation learning in real-world settings. More discussion of these implications is provided in App. I.

## Acknowledgments and Disclosure of Funding

The authors gratefully acknowledge support from the Responsible AI Research (RAIR) Centre (J. Q. Shi and Z. Zhang), the Centre for Augmented Reasoning (CAR) (Y. Liu and E. Gao), and the Commonwealth Bank of Australia through the CommonBank Centre for Foundational AI Research (A. van den Hengel). This support was essential to the completion of the research presented in this publication. The authors also thank the anonymous reviewers for their constructive feedback.

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

# On the Value of Cross-Modal Misalignment in Multimodal Representation Learning (Appendix)

## Contents

# A  Notation and Terminology

Table 1 provides a summary of the notations and terminologies used throughout the paper.

Table 1: Notations and terminologies used throughout the paper.

| | |
|---|---|
| **Observation and Latent Spaces** | |
| $\mathcal{X}$ | Image observation space ($\subseteq \mathbb{R}^{d_x}$) |
| $\mathcal{T}^{(\theta)}$ | Text observation subspace under selection bias $\theta$ |
| $\mathcal{S}$ | Latent semantic space ($\subseteq \mathbb{R}^{n_s}$) |
| $\mathcal{M}_x$ | Image-specific non-semantic latent space |
| $\mathcal{M}_t$ | Text-specific non-semantic latent space |
| $\mathbb{I}_{\mathbf{s}}$ | Index set of semantic variables: $[n_s]$ |
| $\mathbb{I}_{\text{inv}}$ | Index subset of semantic variables that remain invariant under distribution shift |
| $\mathbb{I}_{\text{var}}$ | Index subset of semantic variables that vary under distribution shift |
| **Mappings and Functions** | |
| $g_x$ | Differeomorphic generative mapping for images: $\mathcal{S} \times \mathcal{M}_x \to \mathcal{X}$ |
| $g_{t^{(\theta)}}$ | Differeomorphic generative mapping for text under selection $\theta$: $\mathcal{S}_{\mathbb{I}_\theta} \times \mathcal{M}_t \to \mathcal{T}^{(\theta)}$ |
| $\mathcal{G}_t$ | A diffeomorphism function class where $g_{t^{(\theta)}}$ is selected from |
| $f_x$ | Image encoder: $\mathcal{X} \to (0,1)^n$ with specified $n$ |
| $f_t$ | Text encoder: $\mathcal{T}^{(\theta)} \to (0,1)^n$ with specified $n$ |
| **Loss Functions** | |
| $\mathcal{L}_{\text{MMCL}}(f_x, f_t)$ | Symmetric InfoNCE loss for MMCL (Eq. (1)) |
| $\mathcal{L}_{\text{SymAlignMaxEnt}}(f_x, f_t)$ | Alignment and entropy maximization loss (Eq. (2)) |
| **Notations for Cross-Modal Misalignment** | |
| $\theta$ | Selection bias, an integer realization in the range $[2^{n_s} - 1]$ |
| $\mathcal{P}^+(\mathbb{I}_{\mathbf{s}})$ | The set of all non-empty subsets of $\mathbb{I}_{\mathbf{s}}$ |
| $\mathbb{I}_\theta$ | A selected semantic subset indexed by $\theta$, $\mathbb{I}_\theta \in \mathcal{P}^+(\mathbb{I}_{\mathbf{s}})$ |
| $\mathbb{I}_\theta^c$ | Omitted semantic subset under $\theta$, $\mathbb{I}_\theta^c = \mathbb{I}_{\mathbf{s}} \setminus \mathbb{I}_\theta$ |
| $\rho$ | Perturbation bias, an integer realization in the range $[2^{|\mathbb{I}_\theta|} - 1]$ |
| $\mathcal{P}_{\text{prop}}(\mathbb{I}_\theta)$ | The set of all proper subsets of $\mathbb{I}_\theta$ |
| $\mathbb{I}_\rho$ | A subset of $\mathbb{I}_\theta$ subject to perturbation indexed by $\rho$, $\mathbb{I}_\rho \in \mathcal{P}_{\text{prop}}(\mathbb{I}_\theta)$ |
| $\mathbb{I}_\rho^c$ | Subset of $\mathbb{I}_\theta$ that always be unbiased under $\rho$, $\mathbb{I}_\rho^c = \mathbb{I}_\theta \setminus \mathbb{I}_\rho$ |
| $p_{\mathbf{s}_{\mathbb{I}_\theta} \mid \mathbf{s}, \theta, \rho}$ | Perturbation conditional distribution, reflecting misalignment (Eq. (5)) |
| $A \subseteq \mathbb{I}_\rho$ | A random subset of semantic variables subject to perturbation, drawn from $p_A$ |
| **Distributions and Operators** | |
| $p_{\mathbf{s}}$, $p_{\mathbf{m}_x}$, $p_{\mathbf{m}_t}$ | Distributions over semantic, image-specific and text-specific variables, respectively |
| $H(\cdot)$ | Differential entropy |
| $\delta(\cdot)$ | Dirac delta function |
| **Random Variables** | |
| $\mathbf{x}$ | Image observation sampled from $\mathcal{X}$ |
| $\mathbf{t}^{(\theta)}$ | Text observation under selection view $\theta$, sampled from $\mathcal{T}^{(\theta)}$ |
| $\mathbf{s}$ | Latent semantic variables in $\mathcal{S}$ |
| $\mathbf{s}_{\mathbb{I}_\theta}$ | Selected latent semantic variables for generating text |
| $\mathbf{s}_{\mathbb{I}_\theta^c}$ | Omitted latent semantic variables when generating text |
| $\mathbf{s}_{\mathbb{I}_\rho}$ | Perturbable latent semantic variables for generating text |
| $\mathbf{s}_{\mathbb{I}_\rho^c}$ | Unbiased latent semantic variables within the selected part |
| $\mathbf{m}_x$ | Image-specific non-semantic variable in $\mathcal{M}_x$ |
| $\mathbf{m}_t$ | Text-specific non-semantic variable in $\mathcal{M}_t$ |
| $\mathbf{z}_x$ | Combined latent variables for images: $(\mathbf{s}, \mathbf{m}_x)$ |
| $\mathbf{z}_{t^{(\theta)}}$ | Combined latent variables for text under $\theta$: $(\mathbf{s}_{\mathbb{I}_\theta}, \mathbf{m}_t)$ |

# B  Related Work

**Theoretical multimodal (multi-view) contrastive learning.**   Recent work has sought to formalize the theoretical foundations of multimodal and multi-view representation learning, particularly under contrastive objectives. Wang et al. [69] decompose the InfoNCE loss [51] into an alignment term, which pulls positive pairs together, and a uniformity term, which encourages dispersion over a hypersphere—laying the groundwork for subsequent analysis. Zimmermann et al. [83] show that contrastive objectives can invert the data-generating process, while Liu et al. [45] extend this result to multimodal settings. However, these approaches often rely on strong assumptions about latent distributions or manifold structures, which limits their practical applicability. A complementary line of work, such as [68], demonstrates that contrastive learning with data augmentation can recover shared content without requiring such strong parametric assumptions. This has been extended to MMCL by [48, 13], and Yao et al. [78] further investigate identifiability under partial observability in multi-view settings. Distinct from prior work, we do not assume fixed content-style decompositions, causal directions, or perfectly aligned pairs. Instead, we analyze MMCL with image-text pairs under cross-modal misalignment and systematically examine its impact on representation learning.

**Vision-language models and perspectives on misalignment.**   Multimodal contrastive learning (MMCL) has achieved significant empirical success, particularly in aligning visual and textual modalities using models such as CLIP [55] and ALIGN [27]. These successes are partly attributed to the use of massive training corpora, e.g., LAION-5B [61], which are substantially larger than those used for vision-only foundation models [22, 52]. However, real-world multimodal datasets are often imperfectly aligned and noisy [49]. Existing empirical methods [1, 34] typically treat such misalignment as label noise, employing strategies such as multiple-instance learning or dataset refinement [1] to mitigate its impact. While some recent work suggests that contrastive models are robust to certain forms of structured misalignment [50], Others suggest augmenting text as a proxy for changes in visual content [6]. Our work suggests that cross-modal misalignment can act as either a barrier or a bridge, depending on the application. Unlike [50], which assumes linear representations without modeling the generative process, our analysis is grounded in a realistic latent variable model that provides a deeper understanding of misalignment from a data-generating perspective.

**Identifiability in latent variable models.**   Identifiability analysis addresses the fundamental question of whether the learning process can uniquely recover the latent generative structure or distribution underlying the observed data. This problem has been extensively studied in the context of nonlinear independent component analysis (ICA) [23, 46, 28, 63] and causal representation learning (CRL) [41, 43, 80, 77, 44]. In either setting, full identifiability, typically up to permutation, is rarely achievable in practice without strong assumptions, such as access to interventional data. Consequently, recent works have focused on partial identifiability [20, 31, 18] or relaxed equivalence classes, such as identifiability up to linear transformations [83, 45] or up to group-wise/block-wise indeterminacy [68, 78], which can offer sufficient guarantees for specific tasks or applications. In the context of multimodal representation learning, several recent studies have explored identifiability results [48, 13, 45], but largely neglect the presence of cross-modal misalignment. In contrast, our work explicitly models misalignment and adopts a block-identifiability definition to characterize the extent to which semantic factors can be recovered up to an invertible mapping.

**Invariant representation learning.**   Invariant representation learning (IRL) seeks to learn representations that remain robust under distributional shifts between environments [2, 15, 18], particularly in settings where empirical risk minimization (ERM) [66] fails to generalize out of distribution. In the absence of such challenge, ERM is sufficient for in-distribution prediction, rendering the objective of IRL unnecessary. From a causal perspective, learning invariant representations, or more ambitiously, autonomous mechanisms [60], requires observing sufficient variations in the latent factors underlying data [41]. Such variability can be introduced through interventional data [39, 33], exchangeability assumptions [56], or the use of auxiliary variables such as domain indices [80, 42]. However, direct interventions on latent variables are typically infeasible in real-world observational data, and auxiliary variable methods often require access to a large number of diverse environments to ensure identifiability—an assumption that is often challenging to satisfy in practice. Our work sheds light on an alternative approach: by leveraging the inherent flexibility of text supervision, we show that manipulating biases, specifically through selective omission or semantic perturbation in text, can serve as a controllable proxy for environmental variation.

## C  Proofs

### C.1  Lemmas

Before proceeding with the proof, we first establish the following lemmas.

**Lemma C.1** (Global Minimum of $\mathcal{L}_{\text{SymAlignMaxEnt}}$). *In the context of the proposed LVM as outlined in § 3.1, and Asms. 3.1 and 3.2, the global minimum of*

$$\mathcal{L}_{\text{SymAlignMaxEnt}}(f_x, f_t) = \underset{(\mathbf{x},\mathbf{t}^{(\theta)})\sim p_{\mathbf{x},\mathbf{t}(\theta)}}{\mathbb{E}} [\|f_x(\mathbf{x}) - f_t(\mathbf{t}^{(\theta)})\|_2] - \frac{1}{2}\Big(H\big(f_x(\mathbf{x})\big) + H\big(f_t(\mathbf{t}^{(\theta)})\big)\Big), \quad (6)$$

*is 0. This minimum can be attained by the following pair of smooth functions:*

$$f_x^* = \mathbf{d} \circ (g_x^{-1})_{\mathbb{I}_\rho^c} : \mathcal{X} \to (0,1)^n, \quad (7)$$

$$f_t^* = \mathbf{d} \circ (g_{t(\theta)}^{-1})_{\mathbb{I}_\rho^c} : \mathcal{T}^{(\theta)} \to (0,1)^n, \quad (8)$$

*where:*

- *$g_x$ and $g_{t(\theta)}$ denote the true underlying generative mappings for images and paired text, respectively, as described in § 3.*

- *The operator $(\cdot)_{\mathbb{I}_\rho^c}$ extracts the components corresponding to the unbiased semantic variables (i.e., unaffected by the selection bias nor the perturbation bias), with $n = \big|\mathbb{I}_\rho^c\big|$ being their dimensionality.*

- *$\mathbf{d} = (d_1, \ldots, d_n)$ is defined via the Darmois construction [12, 23, 68], where for each $i \in [n]$,*

$$d_i(\mathbf{s}_{\mathbb{I}_\rho^c}) = \text{CDF}_i\big(s_{\mathbb{I}_\rho^c,i} \mid \mathbf{s}_{\mathbb{I}_\rho^c,[i-1]}\big) = \mathbb{P}\Big(S_{\mathbb{I}_\rho^c,i} \leq s_{\mathbb{I}_\rho^c,i} \,\Big|\, \mathbf{s}_{\mathbb{I}_\rho^c,[i-1]}\Big),$$

*with $\text{CDF}_i$ denoting the conditional cumulative distribution of $s_{\mathbb{I}_\rho^c,i}$ given $\mathbf{s}_{\mathbb{I}_\rho^c,[i-1]}$.*

**Proof of Lem. C.1.**  We prove that the candidate functions $f_x^*$ and $f_t^*$ in Equations (7) and (8) yield $\mathcal{L}_{\text{SymAlignMaxEnt}}(f_x^*, f_t^*) = 0$. Substituting these candidate functions into the loss in Eq. (6), we have

$$\mathcal{L}_{\text{SymAlignMaxEnt}}(f_x^*, f_t^*) = \underset{(\mathbf{x},\mathbf{t}^{(\theta)})\sim p_{\mathbf{x},\mathbf{t}(\theta)}}{\mathbb{E}} \Big[\|f_x^*(\mathbf{x}) - f_t^*(\mathbf{t}^{(\theta)})\|_2\Big]$$

$$- \frac{1}{2}\Big(H\big(f_x^*(\mathbf{x})\big) + H\big(f_t^*(\mathbf{t}^{(\theta)})\big)\Big).$$

By the invertibility of the generative processes $g_x$ and $g_{t(\theta)}$ (see § 3), we may change variables to express the expectation over the latent variables:

$$\mathcal{L}_{\text{SymAlignMaxEnt}}(f_x^*, f_t^*) = \underset{(\mathbf{z}_x,\mathbf{z}_{t(\theta)})\sim p_{\mathbf{z}_x,\mathbf{z}_{t(\theta)}}}{\mathbb{E}} \Big[\|\mathbf{d}(\mathbf{s}_{\mathbb{I}_\rho^c}) - \mathbf{d}(\tilde{\mathbf{s}}_{\mathbb{I}_\rho^c})\|_2\Big]$$

$$- \frac{1}{2}\Big(H\big(\mathbf{d}(\mathbf{s}_{\mathbb{I}_\rho^c})\big) + H\big(\mathbf{d}(\tilde{\mathbf{s}}_{\mathbb{I}_\rho^c})\big)\Big),$$

where $\mathbf{s}_{\mathbb{I}_\rho^c}$ and $\tilde{\mathbf{s}}_{\mathbb{I}_\rho^c}$ denote the preserved unbiased components of the semantic variables across image-text pairs, respectively.

We now show that these unbiased semantic components are identical across modalities almost everywhere (a.e.). By Asm. 3.2, for any image-text pair the text is generated via a random perturbation process that modifies only a subset $A \subseteq \mathbb{I}_\rho$ of the activated semantic variables. Specifically, recall Eq. (5), the perturbation density is defined as

$$p_{\tilde{\mathbf{s}}_{\mathbb{I}_\theta}|\mathbf{s},\theta,\rho}\big(\tilde{\mathbf{s}}_{\mathbb{I}_\theta} \mid \mathbf{s}, A\big) = \delta\big(\tilde{\mathbf{s}}_{\mathbb{I}_\theta \setminus A} - \mathbf{s}_{\mathbb{I}_\theta \setminus A}\big) \, p_{\tilde{\mathbf{s}}_A|\mathbf{s}_A}\big(\tilde{\mathbf{s}}_A \mid \mathbf{s}_A\big).$$

Since $A \subseteq \mathbb{I}_\rho$, it follows that the indices in $\mathbb{I}_\rho^c$ are a subset of those in $\mathbb{I}_\theta \setminus A$; that is, $\mathbb{I}_\rho^c \subseteq \mathbb{I}_\theta \setminus A$. Thus, the Dirac delta in the above expression enforces that

$$\tilde{\mathbf{s}}_{\mathbb{I}_\rho^c} = \mathbf{s}_{\mathbb{I}_\rho^c} \quad \text{almost surely (a.s.)} \quad \forall\, \mathbf{s} \sim p_{\mathbf{s}}, \tilde{\mathbf{s}}_{\mathbb{I}_\theta} \sim p_{\tilde{\mathbf{s}}_{\mathbb{I}_\theta}|\mathbf{s},\theta,\rho},$$

under selection bias $\theta$ and perturbation $\rho$, regardless of the particular perturbation set $A$ at each time.

Further, by the properties of the Darmois construction [12], the mapping $\mathbf{d}$ transforms $\mathbf{s}_{\mathbb{I}_\rho^c}$ into a uniform distribution over $(0,1)^n$ (with $n = |\mathbb{I}_\rho^c|$). Since the uniform distribution is the unique maximum entropy (i.e., zero) distribution on a bounded domain (under no further moment constraints) [26, 10], the entropy terms in the loss are maximized. In the formulation of $\mathcal{L}_{\text{SymAlignMaxEnt}}$, this maximal entropy precisely cancels any potential reduction in the loss, ensuring that

$$\mathcal{L}_{\text{SymAlignMaxEnt}}(f_x^*, f_t^*) = 0.$$

Therefore, the global minimum of $\mathcal{L}_{\text{SymAlignMaxEnt}}$ is achieved at 0 by the given function pairs $f_x^*$ and $f_t^*$, completing the proof. $\qquad\square$

**Lemma C.2** (Uniformizing Mapping Preserves All Information). *Let $h : \mathcal{U} \to \mathcal{V}$ be a smooth map between simply connected, open $\mathcal{C}^1$ manifolds $\mathcal{U}, \mathcal{V} \subseteq \mathbb{R}^n$. Suppose that $\mathbf{u}$ is a random variable taking values in $\mathcal{U}$ with a smooth probability density that is strictly positive a.e.. If the pushforward $\mathbf{v} = h(\mathbf{u})$ is uniformly distributed on $\mathcal{V}$, then $h$ is a global diffeomorphism; in other words, $h$ is bijective and depends on every component of $\mathbf{u}$.*[9]

***Proof of Lem. C.2.*** Let $p_{\mathbf{u}} : \mathcal{U} \to \mathbb{R}$ and $p_{\mathbf{v}} : \mathcal{V} \to \mathbb{R}$ denote the probability density functions of $\mathbf{u}$ and $\mathbf{v}$, respectively. Since $\mathbf{v} = h(\mathbf{u})$, the change-of-variables formula (refer to, e.g., [5]) yields

$$p_{\mathbf{v}}(\mathbf{v}) = p_{\mathbf{u}}(\mathbf{u}) \cdot \left| \det J(h)(\mathbf{u}) \right|^{-1},$$

where $\mathbf{u}$ is any preimage of $\mathbf{v}$ under $h$. By assumption, $\mathbf{v}$ is uniformly distributed on $\mathcal{V}$; that is, there exists a constant $C > 0$ such that

$$p_{\mathbf{v}}(\mathbf{v}) = C \quad \text{for all } \mathbf{v} \in \mathcal{V}.$$

Thus, for any $\mathbf{u}$ with $\mathbf{v} = h(\mathbf{u})$ we obtain

$$\left| \det J(h)(\mathbf{u}) \right|^{-1} = \frac{C}{p_{\mathbf{u}}(\mathbf{u})}.$$

Since $p_{\mathbf{u}}(\mathbf{u})$ is strictly positive a.e. on $\mathcal{U}$, it follows that

$$\left| \det J(h)(\mathbf{u}) \right|^{-1} > 0 \quad \text{a.s.} \quad \forall \mathbf{u} \sim p_{\mathbf{u}},$$

or equivalently, $\det J(h)(\mathbf{u}) \neq 0$ a.e.. By the Inverse Function Theorem (see, e.g., [58]), this implies that $h$ is a local diffeomorphism.

Moreover, since $\mathcal{U}$ and $\mathcal{V}$ are simply connected, open $\mathcal{C}^1$ manifolds, standard covering space theory (refer to, e.g., the discussion around Theorem 1.38 in [21]) implies that $h$ is a covering map. The uniformity of $p_{\mathbf{v}}$ forces $h$ to be surjective (otherwise, some points in $\mathcal{V}$ would have zero density, contradicting uniformity). Since any covering map from a simply connected space is trivial, $h$ must be equivalent to the identity covering. In other words, $h$ is a homeomorphism onto $\mathcal{V}$ and hence both injective and surjective (i.e., a bijection).

Finally, the fact that the Jacobian determinant is nonzero a.e. guarantees that $h$ depends on all components of $\mathbf{u}$; if any component were omitted, the rank of the Jacobian would drop, contradicting non-singularity. Furthermore, by the Global Inverse Function Theorem (refer to, e.g., [59]), the inverse of $h$ is smooth.

In summary, $h$ is a global diffeomorphism from $\mathcal{U}$ onto $\mathcal{V}$. Consequently, it preserves all information of $\mathbf{u}$: every variation in $\mathbf{u}$ is reflected in $\mathbf{v} = h(\mathbf{u})$, and $\mathbf{u}$ can be uniquely and smoothly recovered from $\mathbf{v}$. This completes the proof. $\qquad\square$

---

[9]We do not claim originality for this result due to its fundamental nature in topology and measure theory; rather, we detail it here as a tool for our subsequent arguments.

## C.2 Proof of Theorem 4.1

We now proceed to prove Thm. 4.1. To begin, we restate the theorem for clarity:

**Theorem 4.1** (Identifiability of latent semantic variables). *Let $(\mathbf{x}, \mathbf{t}^{(\theta)})$ be image-text pairs drawn from the data-generating process described in § 3, where $\mathbf{x}$ is generated according to Eq. (3) and $\mathbf{t}^{(\theta)}$ is generated by Eq. (4). Further, suppose that Asms. 3.1 and 3.2 hold. Denote by $\mathbf{s}_{\mathbb{I}_\rho^c}$ the subset of semantic variables that annotated without bias in the text, and define its dimension as $n = |\mathbb{I}_\theta| - |\mathbb{I}_\rho|$. Let $f_x : \mathcal{X} \to (0,1)^n$ and $f_t : \mathcal{T}^{(\theta)} \to (0,1)^n$ be sufficiently flexible, smooth functions. Then, minimizing the loss $\mathcal{L}_{SymAlignMaxEnt}$ in Eq. (2) over samples $(\mathbf{x}, \mathbf{t}^{(\theta)})$ guarantees that $f_x$ and $f_t$ block-identify the semantic variables $\mathbf{s}_{\mathbb{I}_\rho^c}$ in the sense of Defn. 4.1.*

*Proof of Thm. 4.1.* The proof is organized into the following five steps:

- First, we show that the objective function $\mathcal{L}_{SymAlignMaxEnt}(f_x, f_t)$ achieves a global minimum value of 0. At this minimum, any pair of smooth functions $f_x$ and $f_t$ satisfying this condition must exhibit invariance across modalities. This invariance condition ensures that the learned image representations and text representations must align across all positive $\mathbf{x}$ and $\mathbf{t}^{(\theta)}$ pairs.

- Next, we prove that minimizing $\mathcal{L}_{SymAlignMaxEnt}$ inherently eliminates any dependence of the learned representations on modality-specific variables $\mathbf{m}_x$ or $\mathbf{m}_t$. This ensures that the representations are restricted to the dependence on latent semantic variables.

- By contradiction, we further establish that any contribution from the omitted semantic variables induced by selection bias $\theta$, i.e., $\mathbf{s}_{\mathbb{I}_\theta^c}$, would violate the invariance condition established in Step 1. This guarantees that the representations exclude the dependence on omitted semantic variables.

- We then establish the exclusion of perturbed semantic variables influenced by perturbation bias $\rho$, i.e., $\mathbf{s}_{\mathbb{I}_\rho}$, from the learned representations, also by contradiction.

- Finally, we demonstrate that the optimized mappings, which compose with the underlying generative functions, are invertible with respect to the learned representations and the true unbiased semantic variables $\mathbf{s}_{\mathbb{I}_\rho^c}$. This ensures that the representations block-identify the preserved unbiased semantic variables, thereby concluding the proof.

**Step 1** (Global minimum and invariance condition). Let $f_x : \mathcal{X} \to (0,1)^n$ and $f_t : \mathcal{T}^{(\theta)} \to (0,1)^n$ be any smooth functions attaining the global minimum. Define the smooth mappings:

$$h_x = f_x \circ g_x, \quad h_t = f_t \circ g_{t^{(\theta)}}.$$

Since all terms in $\mathcal{L}_{SymAlignMaxEnt}$ are non-negative, and its global minimum is 0 by Lem. C.1, each term in $\mathcal{L}_{SymAlignMaxEnt}$ must vanish a.s. for any pairing $(\mathbf{x}, \mathbf{t}^{(\theta)})$. Thus, minimizing $\mathcal{L}_{SymAlignMaxEnt}$ leads to:

$$\mathbb{E}_{(\mathbf{x}, \mathbf{t}^{(\theta)}) \sim p_\mathbf{x} p_{\mathbf{t}^{(\theta)} | \mathbf{x}}} [\| f_x(\mathbf{x}) - f_t(\mathbf{t}^{(\theta)}) \|_2] = \mathbb{E}_{(\mathbf{z}_x, \mathbf{z}_{t(\theta)}) \sim p_{\mathbf{z}_x} p_{\mathbf{z}_{t(\theta)} | \mathbf{z}_x}} [\| h_x(\mathbf{z}_x) - h_t(\mathbf{z}_{t(\theta)}) \|_2] = 0, \quad (9)$$

$$H\big(f_x(\mathbf{x})\big) = H\big(h_x(\mathbf{z}_x)\big) = 0, \tag{10}$$

$$H\big(f_t(\mathbf{t}^{(\theta)})\big) = H\big(h_t(\mathbf{z}_{t(\theta)})\big) = 0. \tag{11}$$

From Eq. (9), it follows that

$$h_t(\mathbf{z}_{t(\theta)}) = h_x(\mathbf{z}_x) \quad \text{a.s.} \quad \forall \mathbf{z}_x \sim p_\mathbf{s} p_{\mathbf{m}_x}, \ \mathbf{z}_{t(\theta)} \sim p_{\mathbf{s}_{\mathbb{I}_\theta} | \mathbf{s}, \theta, \rho} p_{\mathbf{m}_t}, \tag{12}$$

which ensures alignment of representations a.s. for any pair $(\mathbf{x}, \mathbf{t}^{(\theta)})$. The substitution of expectations for the image modality in Eq. (9) is valid because $\mathbf{x}$ follows the pushforward distribution of $\mathbf{z}_x$ under the deterministic diffeomorphism $g_x$ (Eq. (3)); a similar argument applies to the text (Eq. (4)).

Equations (10) and (11) imply that $h_x$ and $h_t$ map the latent variables $\mathbf{z}_x$ and $\mathbf{z}_{t(\theta)}$ onto uniform distributions over $(0,1)^n$ (with $n = |\mathbb{I}_\rho^c|$), since their differential entropy equals to zero.

**Step 2** (Exclusion of modality-specific variables) **.** We now show that the smooth functions $h_x$ and $h_t$ depend only on latent semantic variables $\mathbf{s}$ (with further exclusion of components in later steps) and not on modality-specific variables $\mathbf{m}_x$ or $\mathbf{m}_t$.

Since $\mathbf{z}_x = (\mathbf{s}, \mathbf{m}_x)$ and $\mathbf{z}_{t^{(\theta)}} = (\tilde{\mathbf{s}}_{\mathbb{I}_\theta}, \mathbf{m}_t)$ are the latent variables generating images $\mathbf{x}$ and paired text $\mathbf{t}^{(\theta)}$, respectively, the data-generating process in § 3 implies the following independence properties:

(**c1**) $\mathbf{m}_x$ is independent of $\mathbf{z}_{t^{(\theta)}}$: This means changes in $\mathbf{m}_x$ do not influence $\mathbf{z}_{t^{(\theta)}}$. Moreover, since the text generation process $g_{t^{(\theta)}}$ is independent of $\mathbf{m}_x$, it follows that:

$$h_t(\tilde{\mathbf{s}}_{\mathbb{I}_\theta}, \mathbf{m}_t, \mathbf{m}_x) = h_t(\tilde{\mathbf{s}}_{\mathbb{I}_\theta}, \mathbf{m}_t). \tag{13}$$

(**c2**) $\mathbf{m}_t$ is independent of $\mathbf{z}_x$: This means changes in $\mathbf{m}_t$ do not influence $\mathbf{z}_x$. Similarly, since the image generation process $g_x$ is independent of $\mathbf{m}_t$, it follows that:

$$h_x(\mathbf{s}, \mathbf{m}_x, \mathbf{m}_t) = h_x(\mathbf{s}, \mathbf{m}_x). \tag{14}$$

Combining Equations (12) and (13), we have:

$$h_x(\mathbf{s}, \mathbf{m}_x) = h_t(\tilde{\mathbf{s}}_{\mathbb{I}_\theta}, \mathbf{m}_t, \mathbf{m}_x), \quad \text{a.s.} \quad \forall \mathbf{z}_x \sim p_\mathbf{s}\, p_{\mathbf{m}_x}, \ \mathbf{z}_{t^{(\theta)}} \sim p_{\tilde{\mathbf{s}}_{\mathbb{I}_\theta} | \mathbf{s}, \theta, \rho}\, p_{\mathbf{m}_t}. \tag{15}$$

Consider a small perturbation $\mathbf{m}_x \mapsto \mathbf{m}_x + \varsigma$, where $\|\varsigma\|$ is arbitrarily small but remains within the open space $\mathcal{M}_x$. Since changes in $\mathbf{m}_x$ do not influence $h_t$ by statement (**c1**), we obtain:

$$h_t(\tilde{\mathbf{s}}_{\mathbb{I}_\theta}, \mathbf{m}_t, \mathbf{m}_x + \varsigma) = h_t(\tilde{\mathbf{s}}_{\mathbb{I}_\theta}, \mathbf{m}_t, \mathbf{m}_x). \tag{16}$$

Since $\mathbf{m}_x$ is independent of $\mathbf{s}$, perturbations in $\mathbf{m}_x$ does not alter the semantic variables, and $p_{\mathbf{m}_x} > 0$ a.e. over $\mathcal{M}_x$ by Asm. 3.1. Thus, substituting

$$(\mathbf{s}, \mathbf{m}_x) \mapsto (\mathbf{s}, \mathbf{m}_x + \varsigma)$$

in Eq. (15) and combining with Eq. (16), we get:

$$h_x(\mathbf{s}, \mathbf{m}_x + \varsigma) = h_x(\mathbf{s}, \mathbf{m}_x).$$

By the smoothness of $h_x$ (inherited from the smoothness of $g_x$ and $f_x$), taking $\varsigma \to \mathbf{0}$ gives:

$$\frac{\partial h_x}{\partial \mathbf{m}_x} = \lim_{\varsigma \to \mathbf{0}} \frac{h_x(\mathbf{s}, \mathbf{m}_x + \varsigma) - h_x(\mathbf{s}, \mathbf{m}_x)}{\varsigma} = 0.$$

Thus, $h_x$ is independent of $\mathbf{m}_x$.

A symmetric argument applies to $\mathbf{m}_t$. If $\mathbf{m}_t \mapsto \mathbf{m}_t + \varsigma$ with $\varsigma$ a small enough perturbation, then by Eq. (14) in statement (**c2**), $h_x(\mathbf{s}, \mathbf{m}_x, \mathbf{m}_t + \varsigma)$ remains unchanged. The invariance condition in Eq. (12) then forces $h_t(\tilde{\mathbf{s}}_{\mathbb{I}_\theta}, \mathbf{m}_t + \varsigma)$ to remain constant w.r.t. $\varsigma$, showing that $h_t$ is independent of $\mathbf{m}_t$. Therefore, the learned representations satisfy:

$$h_x(\mathbf{z}_x) = h_x(\mathbf{s}), \quad \text{a.s.} \quad \forall \mathbf{z}_x \sim p_\mathbf{s}\, p_{\mathbf{m}_x}, \tag{17}$$

$$h_t(\mathbf{z}_{t^{(\theta)}}) = h_t(\tilde{\mathbf{s}}_{\mathbb{I}_\theta}), \quad \text{a.s.} \quad \forall \mathbf{z}_{t^{(\theta)}} \sim p_{\tilde{\mathbf{s}}_{\mathbb{I}_\theta} | \mathbf{s}, \theta, \rho}\, p_{\mathbf{m}_t}. \tag{18}$$

**Step 3** (Exclusion of omitted semantic variables)**.** We now establish that the function $h_x$ is independent of $\mathbf{s}_{\mathbb{I}_\theta^c}$, where $\mathbb{I}_\theta^c = \mathbb{I}_\mathbf{s} \setminus \mathbb{I}_\theta$. In other words, the learned representations do not contain information about the omitted semantic variables that are absent in the corresponding text.

Using the invariance condition in Eq. (12), together with the independence of modality-specific non-semantic variables in Equations (17) and (18), we have the following updated invariance condition:

$$h_x(\mathbf{s}) = h_t(\tilde{\mathbf{s}}_{\mathbb{I}_\theta}), \quad \text{a.s.} \quad \forall \mathbf{s} \sim p_\mathbf{s}, \ \tilde{\mathbf{s}}_{\mathbb{I}_\theta} \sim p_{\tilde{\mathbf{s}}_{\mathbb{I}_\theta} | \mathbf{s}, \theta, \rho}. \tag{19}$$

Next, we show by contradiction that $h_x$ is independent of $\mathbf{s}_{\mathbb{I}_\theta^c}$. Suppose, for the sake of contradiction, that there exists a function $h_x^c = f_x^c \circ g_x$ which depends on at least one component of the omitted semantic variables $\mathbf{s}_{\mathbb{I}_\theta^c}$. Formally,

$$\exists\, l \in \mathbb{I}_\theta^c, \ (\mathbf{s}_{\mathbb{I}_\theta}^*, \mathbf{s}_{\mathbb{I}_\theta^c}^*) \in \mathcal{S}, \quad \text{such that} \quad \frac{\partial\, h_x^c(\mathbf{s}_{\mathbb{I}_\theta}^*, \mathbf{s}_{\mathbb{I}_\theta^c}^*)}{\partial\, s_l^*} \neq 0.$$

By the $C^1$ continuity of $h_x^c$, guaranteed by the smoothness of both $g_x$ and $f_x^c$, and that $\mathcal{S}$ is an open space, it follows that

$$\exists \eta > 0 : s_l \mapsto h_x^c\big(\mathbf{s}_{\mathbb{I}_\theta}^*, (s_l, \mathbf{s}_{\mathbb{I}_\theta^c \setminus \{l\}}^*)\big) \quad \text{is strictly monotonic on} \quad (s_l^* - \eta,\ s_l^* + \eta),$$

where $\mathbf{s}_{\mathbb{I}_\theta^c \setminus \{l\}}^*$ denotes all components of $\mathbf{s}_{\mathbb{I}_\theta^c}^*$ except $s_l$.

Since $p_\mathbf{s} > 0$ a.e. on $\mathcal{S}$, we can find two distinct realizations of latent semantic variables

$$\big(\mathbf{s}_{\mathbb{I}_\theta}^*, (s_l^-, \mathbf{s}_{\mathbb{I}_\theta^c \setminus \{l\}}^*)\big),\ \big(\mathbf{s}_{\mathbb{I}_\theta}^*, (s_l^+, \mathbf{s}_{\mathbb{I}_\theta^c \setminus \{l\}}^*)\big) \quad \text{with } s_l^- \in (s_l^* - \eta,\ s_l^*),\ s_l^+ \in (s_l^*,\ s_l^* + \eta) \tag{20}$$

that correspond to two different image observations, such that

$$h_x^c\big(\mathbf{s}_{\mathbb{I}_\theta}^*, (s_l^-, \mathbf{s}_{\mathbb{I}_\theta^c \setminus \{l\}}^*)\big) \ \neq\ h_x^c\big(\mathbf{s}_{\mathbb{I}_\theta}^*, (s_l^+, \mathbf{s}_{\mathbb{I}_\theta^c \setminus \{l\}}^*)\big). \tag{21}$$

However, combining Eq. (19), we have

$$h_x^c\big(\mathbf{s}_{\mathbb{I}_\theta}^*, (s_l^-, \mathbf{s}_{\mathbb{I}_\theta^c \setminus \{l\}}^*)\big) \ =\ h_x^c\big(\mathbf{s}_{\mathbb{I}_\theta}^*, (s_l^+, \mathbf{s}_{\mathbb{I}_\theta^c \setminus \{l\}}^*)\big) \ =\ h_t\big(\tilde{\mathbf{s}}_{\mathbb{I}_\theta}^*\big), \tag{22}$$

where $\tilde{\mathbf{s}}_{\mathbb{I}_\theta}^*$ represents the perturbed semantic variables of $\mathbf{s}_{\mathbb{I}_\theta}^*$ introduced by selection bias (with the exclusion of perturbed components further addressed in the next step).

Equations (21) and (22) thus contradict each other. Hence, such a function $h_x^c$ cannot exist. Consequently, $h_x$ must be independent of $\mathbf{s}_{\mathbb{I}_\theta^c}$. Formally,

$$h_x(\mathbf{z}_x) \ =\ h_x(\mathbf{s}_{\mathbb{I}_\theta}), \quad \text{a.s.} \quad \forall \mathbf{z}_x \sim p_\mathbf{s}\, p_{\mathbf{m}_x}. \tag{23}$$

---

**Clarification C.1** (Causal Interpretations). **(i)** *Justification for the existence of distinct points in Eq.* (20). This follows from the assumption $p_\mathbf{s} > 0$ a.e. on $\mathcal{S}$ by Asm. 3.1. From a latent SCM perspective [53, 67], even if a specific semantic component $s_l$ in $\mathbf{s}_{\mathbb{I}_\theta^c}$ is the ancestor node of some other semantic components in $\mathbf{s}_{\mathbb{I}_\theta}$, the strict positivity of $p_\mathbf{s}$ ensures that the exogenous noise variables are well-defined. Thus, for different values of $s_l$, there exist corresponding noise values that keep $\mathbf{s}_{\mathbb{I}_\theta}^*$ remaining fixed. **(ii)** *What if the unknown causal structure is* $\mathbf{s}_{\mathbb{I}_\theta^c} \to \mathbf{s}_{\mathbb{I}_\theta}$? The potential causal influence from $\mathbf{s}_{\mathbb{I}_\theta^c}$ to $\mathbf{s}_{\mathbb{I}_\theta}$ does not resolve the contradiction. Independence here means that, once $\mathbf{s}_{\mathbb{I}_\theta}$ is set, there is no direct functional path from $\mathbf{s}_{\mathbb{I}_\theta^c}$ to the representations $h_x(\mathbf{s}_{\mathbb{I}_\theta})$, i.e., the causal influence among them is fully accounted for by the realized value of $\mathbf{s}_{\mathbb{I}_\theta}$.

---

In summary, these arguments show that $h_x$ is genuinely independent of $\mathbf{s}_{\mathbb{I}_\theta^c}$, even allowing for arbitrary unknown causal interactions among the latent semantic variables.

**Step 4** (Exclusion of perturbed semantic variables). We now demonstrate that both representations are independent of $\mathbf{s}_{\mathbb{I}_\rho}$ and $\tilde{\mathbf{s}}_{\mathbb{I}_\rho}$ respectively, as a consequence of the contradiction between the invariance condition and the random perturbations introduced by perturbation bias.

First, we refine the invariance condition by excluding omitted semantic variables as established above. Combining Equations (12), (18) and (23), we obtain:

$$h_x(\mathbf{s}_{\mathbb{I}_\theta}) \ =\ h_t\big(\tilde{\mathbf{s}}_{\mathbb{I}_\theta}\big), \quad \text{a.s.} \quad \forall \mathbf{s} \sim p_\mathbf{s},\ \tilde{\mathbf{s}}_{\mathbb{I}_\theta} \sim p_{\tilde{\mathbf{s}}_{\mathbb{I}_\theta} | \mathbf{s}, \theta, \rho}. \tag{24}$$

Next, we show that $h_t$ must be independent of $\tilde{\mathbf{s}}_{\mathbb{I}_\rho}$ by contradiction. Suppose, for a contradiction, that there exist some function $h_t^c = f_t^c \circ g_{t(\theta)}$ which depends on at least on component of the perturbed semantic variables $\tilde{\mathbf{s}}_{\mathbb{I}_\rho}$. Formally,

$$\exists l \in \mathbb{I}_\rho,\ (\tilde{\mathbf{s}}_{\mathbb{I}_\rho}^*, \tilde{\mathbf{s}}_{\mathbb{I}_\rho^c}^*), \quad \text{such that} \quad \frac{\partial h_t^c(\tilde{\mathbf{s}}_{\mathbb{I}_\rho}^*, \tilde{\mathbf{s}}_{\mathbb{I}_\rho^c}^*)}{\partial \tilde{s}_l^*} \neq 0.$$

By the $C^1$ continuity of $h_t^c$ guaranteed by the smoothness of $f_t^c$ and $g_{t(\theta)}$, for some sufficiently small $\eta > 0$, we have the following inequality:

$$h_t^c\big((s_l^-, \tilde{\mathbf{s}}_{\mathbb{I}_\rho \setminus \{l\}}^*), \tilde{\mathbf{s}}_{\mathbb{I}_\rho^c}^*\big) \neq h_t^c\big((s_l^+, \tilde{\mathbf{s}}_{\mathbb{I}_\rho \setminus \{l\}}^*), \tilde{\mathbf{s}}_{\mathbb{I}_\rho^c}^*\big),\ \forall s_l^- \in (\tilde{s}_l^* - \eta, \tilde{s}_l^*),\ s_l^+ \in (\tilde{s}_l^*, \tilde{s}_l^* + \eta). \tag{25}$$

On the other hand, by the pairing conditions in Asm. 3.2, there exists at least one subset $A \subseteq \mathbb{I}_\rho$ of perturbed semantic variables such that $l \in A$ and $p_A(A) > 0$. Pick one such set and call it $A$. Define

the realization of latent semantic variables corresponding to the image of this pair as $(\mathbf{s}_A^*, \mathbf{s}_{\mathbb{I}_\theta \setminus A}^*)$ (here, we omit the latent semantic components that have already been excluded in previous steps).

By Asm. 3.2, we know that $\mathbf{s}_{\mathbb{I}_\theta \setminus A}^* = \tilde{\mathbf{s}}_{\mathbb{I}_\theta \setminus A}^*$ a.e., that is, $(\mathbf{s}_{\mathbb{I}_\rho \setminus A}, \mathbf{s}_{\mathbb{I}_\rho^c}^*) = (\tilde{\mathbf{s}}_{\mathbb{I}_\rho \setminus A}^*, \tilde{\mathbf{s}}_{\mathbb{I}_\rho^c}^*)$ a.e.. Thus, we can rewrite the corresponding realization of textual semantic variables $\tilde{\mathbf{s}}_{\mathbb{I}_\theta}^* = (\tilde{\mathbf{s}}_{\mathbb{I}_\rho}^*, \tilde{\mathbf{s}}_{\mathbb{I}_\rho^c}^*)$ as $(\tilde{\mathbf{s}}_A^*, \mathbf{s}_{\mathbb{I}_\theta \setminus A}^*)$.

Further, also by Asm. 3.2, there exists a non-empty open subspace $\mathcal{O}_A \subseteq \mathcal{S}_A$ such that $p_{\tilde{\mathbf{s}}_A | \mathbf{s}_A}(\cdot | \mathbf{s}_A^*)$ is strictly positive on $\mathcal{O}_A$. Since the perturbed random variable $\tilde{\mathbf{s}}_A^*$ is a realization within this open subspace, we know it lies in $\mathcal{O}_A$ and $\mathcal{O}_A$ is non-empty. Moreover, because $p_{\tilde{\mathbf{s}}_A | \mathbf{s}_A}(\cdot | \mathbf{s}_A^*)$ is smooth and strictly positive on $\mathcal{O}_A$, there exists a sufficiently small $\eta_1 > 0$ such that

$$p_{\tilde{\mathbf{s}}_A | \mathbf{s}_A}(\tilde{\mathbf{s}}_A | \mathbf{s}_A^*) > 0, \ \forall \tilde{\mathbf{s}}_A \in \{\tilde{\mathbf{s}}_{A \setminus \{l\}}^*\} \times (\tilde{s}_l^* - \eta_1, \tilde{s}_l^* + \eta_1), \ \text{with} \ \{\tilde{\mathbf{s}}_{A \setminus \{l\}}^*\} \times (\tilde{s}_l^* - \eta_1, \tilde{s}_l^* + \eta_1) \subseteq \mathcal{O}_A.$$

Thus, with a positive probability guaranteed by the above conditional, for the realization of image semantic variables $(\mathbf{s}_A^*, \mathbf{s}_{\mathbb{I}_\theta \setminus A}^*)$, we can construct two distinct realizations of perturbed semantic variables for generating different text (due to $p_{\mathbf{s}} > 0$ a.e.):

$$\big((s_l', \tilde{\mathbf{s}}_{A \setminus \{l\}}^*), \mathbf{s}_{\mathbb{I}_\theta \setminus A}^*\big), \ \ \big((s_l'', \tilde{\mathbf{s}}_{A \setminus \{l\}}^*), \mathbf{s}_{\mathbb{I}_\theta \setminus A}^*\big),$$

where

$$s_l' \in (\tilde{s}_l^* - \eta_2, \tilde{s}_l^*), \quad s_l'' \in (\tilde{s}_l^*, \tilde{s}_l^* + \eta_2) \quad \text{with} \quad \eta_2 = \min(\eta, \eta_1).$$

Based on the invariance condition established in Eq. (24), we have the following equalities:

$$h_t^c\big((s_l', \tilde{\mathbf{s}}_{A \setminus \{l\}}^*), \mathbf{s}_{\mathbb{I}_\theta \setminus A}^*\big) = h_t^c\big((s_l'', \tilde{\mathbf{s}}_{A \setminus \{l\}}^*), \mathbf{s}_{\mathbb{I}_\theta \setminus A}^*\big) = h_x(\mathbf{s}_A^*, \mathbf{s}_{\mathbb{I}_\theta \setminus A}^*).$$

This is contradicted by the inequality established in Eq. (25), which implies that such a $h_t^c$ cannot exist. Consequently, any $h_t$ minimizing the loss must be independent of the perturbed semantic variables $\tilde{\mathbf{s}}_{\mathbb{I}_\rho}$, i.e.,

$$h_t(\tilde{\mathbf{s}}_{\mathbb{I}_\theta}) = h_t(\mathbf{s}_{\mathbb{I}_\rho^c}), \quad \text{a.s.} \quad \forall \tilde{\mathbf{s}}_{\mathbb{I}_\theta} \sim p_{\tilde{\mathbf{s}}_{\mathbb{I}_\theta} | \mathbf{s}, \theta, \rho}. \tag{26}$$

Updating the invariance condition, and combining Equations (24) and (26), we obtain:

$$h_x(\mathbf{s}_{\mathbb{I}_\theta}) = h_t(\mathbf{s}_{\mathbb{I}_\rho^c}), \quad \text{a.s.} \quad \forall \mathbf{s} \sim p_{\mathbf{s}}. \tag{27}$$

The exclusion of $\mathbf{s}_{\mathbb{I}_\rho}$ from image representations $h_x(\mathbf{s}_{\mathbb{I}_\theta})$ can be demonstrated by a similar procedure to **Step 3**, namely, that excluded semantic components from one modality cannot exist in another view, regardless of the latent causal structure among $\mathbf{s}_{\mathbb{I}_\theta}$. Specifically, fixing the value of $\mathbf{s}_{\mathbb{I}_\rho^c}$ and varying $\mathbf{s}_{\mathbb{I}_\rho}$ within a small region, we can sample distinct semantic variables (due to $p_{\mathbf{s}} > 0$ a.e. over $\mathcal{S}$ by Asm. 3.1). The smoothness of $h_x$ then leads to an inequality if it depends on any component in $\mathbb{I}_\rho$. This inequality contradicts the alignment condition established in Eq. (27). Thus, $h_x$ must also be independent of $\mathbf{s}_{\mathbb{I}_\rho}$.

Overall, due to the exclusion of modality-specific variables ($\mathbf{m}_x$ and $\mathbf{m}_t$), omitted semantic variables ($\mathbf{s}_{\mathbb{I}_\theta^c}$) and perturbed semantic variables ($\mathbf{s}_{\mathbb{I}_\rho}$) introduced by selection and perturbation biases for generating text, we now have the following equalities:

$$h_x(\mathbf{z}_x) = h_x(\mathbf{s}_{\mathbb{I}_\rho^c}), \quad \text{a.s.} \quad \forall \mathbf{z}_x \sim p_{\mathbf{s}} \, p_{\mathbf{m}_x}, \tag{28}$$

$$h_t(\mathbf{z}_{t(\theta)}) = h_t(\mathbf{s}_{\mathbb{I}_\rho^c}), \quad \text{a.s.} \quad \forall \mathbf{z}_{t(\theta)} \sim p_{\tilde{\mathbf{s}}_{\mathbb{I}_\theta} | \mathbf{s}, \theta, \rho} \, p_{\mathbf{m}_t}. \tag{29}$$

**Step 5** (Preservation of all unbiased semantic variables)**.** Based on Equations (28) and (29), we define the learned image and textual representations as

$$\hat{\mathbf{z}}_x = h_x(\mathbf{s}_{\mathbb{I}_\rho^c}), \ \hat{\mathbf{z}}_t = h_t(\mathbf{s}_{\mathbb{I}_\rho^c}), \quad \text{with} \quad \hat{\mathbf{z}}_x \in (0,1)^n, \ \hat{\mathbf{z}}_t \in (0,1)^n.$$

According to Equations (10) and (11), we know that the learned representations in both modalities are uniformly distributed over $(0,1)^n$. By directly applying Lem. C.2, it follows that the learned representations $\hat{\mathbf{z}}_x$ (and also $\hat{\mathbf{z}}_t$) include *all* and *only* the information of the unaltered semantic components $\mathbf{s}_{\mathbb{I}_\rho^c}$ a.s., and that $h_x$ (and similarly $h_t$) is invertible. Consequently, the true modality-shared semantic variables $\mathbf{s}_{\mathbb{I}_\rho^c}$ are block-identified by $f_x$ and $f_t$ in the sense of Defn. 4.1.

Thereupon, the proof of Thm. 4.1 is complete. $\qquad\qquad\qquad\qquad\qquad\qquad\qquad\qquad\square$

## C.3 Proof of Corollary 4.1

We now proceed to prove Cor. 4.1. To begin, we restate the corollary for clarity:

**Corollary 4.1** (Identifiability of full latent semantic variables)**.** *Assume that Asms. 3.1 and 3.2 hold. Let the selection bias be $\theta = 2^{n_s} - 1$ and the perturbation bias be $\rho = 1$, such that the full set of semantic variables $\mathbb{I}_\mathbf{s}$ is selected, and the perturbable semantic subset is trivial, i.e., $\mathbb{I}_\rho = \emptyset$. Then, all semantic variables $\mathbf{s}$ are block-identified via smooth functions $f_x : \mathcal{X} \to (0,1)^{n_s}$ and $f_t : \mathcal{T}^{(\theta)} \to (0,1)^{n_s}$, when minimizing $\mathcal{L}_{SymAlignMaxEnt}$.*

*Proof.* As we have fixed a graded lexicographic order over the range of $\theta$, that is, over $\mathcal{P}^+(\mathbb{I}_\mathbf{s})$ as defined in Defn. 3.1. Then, setting $\theta = 2^{n_s} - 1$ corresponds to selecting the full set of semantic variables for text generation, i.e., $\mathbb{I}_\theta = \mathbb{I}_\mathbf{s}$.

Furthermore, we have similarly fixed a graded lexicographic order over the range of $\rho$, i.e., over $\mathcal{P}_{\mathrm{prop}}(\mathbb{I}_\mathbf{s})$, as defined in Defn. 3.2. Given that $\mathbb{I}_\theta = \mathbb{I}_\mathbf{s}$, setting $\rho = 1$ implies that all semantic variables $\mathbf{s}$ are preserved without perturbation during the generation of the corresponding text $\mathbf{t}^{(\theta)}$.

Under these assumptions, and by Asm. 3.2, the perturbing subset $A$ is always trivial because $\mathbb{I}_\rho$ is trivial. Consequently, we have

$$p_{\tilde{\mathbf{s}}_{\mathbb{I}_\theta}|\mathbf{s},\theta,\rho}(\tilde{\mathbf{s}}_{\mathbb{I}_\theta}|\mathbf{s}) = \delta(\mathbf{s} - \mathbf{s}) \quad \text{with} \quad \mathbb{I}_\theta = \mathbb{I}_\mathbf{s}, \ \mathbb{I}_\rho = \emptyset \quad \text{a.s.} \quad \forall \mathbf{s} \sim p_\mathbf{s},$$

which indicates that $\tilde{\mathbf{s}} = \mathbf{s}$ almost surely.

Now consider any pair of smooth functions $f_x : \mathcal{X} \to (0,1)^{n_s}$ and $f_{t^{(\theta)}} : \mathcal{T}^{(\theta)} \to (0,1)^{n_s}$ that achieve the global minimum of the loss in Eq. (2), i.e., yield a value of zero. From **Step 1** and **Step 2** of the proof of Thm. 4.1, it follows that:

$$h_x(\mathbf{z}_x) = h_t(\mathbf{z}_{t^{(\theta)}}) = h_x(\mathbf{s}) = h_t(\mathbf{s}) \quad \text{a.s.} \quad \forall \mathbf{z}_x \sim p_\mathbf{s} \, p_{\mathbf{m}_x}, \ \mathbf{z}_{t^{(\theta)}} \sim p_{\tilde{\mathbf{s}}_{\mathbb{I}_\theta}|\mathbf{s},\theta,\rho} \, p_{\mathbf{m}_t}.$$

Since both the omitted and perturbable index subsets are trivial, we may directly apply Lem. C.2, which implies that the full semantic vector $\mathbf{s}$ is block-identified by both $f_x$ and $f_t$.

This completes the proof. $\qquad\square$

**Definition C.1** (Graded lexicographic order)**.** Let $S$ be a finite set with a total order (e.g., $S = \{1, 2, \ldots, n\}$). The *graded lexicographic order* on the non-empty subsets of $S$ is defined as follows: **(i)** Subsets are ordered first by increasing cardinality (i.e., all subsets of size $k$ precede those of size $k + 1$). **(ii)** Within each group of equal cardinality, subsets are ordered lexicographically: that is, $\{i_1, i_2, \ldots, i_k\} < \{j_1, j_2, \ldots, j_k\}$ if there exists an $\ell$ such that $i_m = j_m$ for all $m < \ell$ and $i_\ell < j_\ell$.

**Clarification C.2** (Fixed graded lexicographic order)**.** The *graded lexicographic order* over subsets of semantic indices is fixed solely for notational consistency and clarity. For example, given semantic variables $\mathbf{s} = (s_1, s_2, s_3)$ with index set $\mathbb{I}_\mathbf{s} = \{1, 2, 3\}$, the graded lexicographic order over the non-empty powerset $\mathcal{P}^+(\mathbb{I}_\mathbf{s})$ yields:

$$\mathcal{P}^+(\mathbb{I}_\mathbf{s}) = \{\{1\}, \{2\}, \{3\}, \{1, 2\}, \{1, 3\}, \{2, 3\}, \{1, 2, 3\}\}.$$

This ordering imposes no constraints on the underlying latent structure and is adopted without loss of generality. Since the true permutation of latent variables is unidentifiable, any consistent order can be used to index subset-valued parameters such as $\theta$ (selection) and $\rho$ (perturbation). Importantly, both $\theta$ and $\rho$ act indirectly through text observations rather than directly operating on the latent space. The fixed order simply determines how each value of $\theta$ or $\rho$ maps to a subset of semantic indices. For instance, omitting color in a caption corresponds to removing the associated latent variable—*which* index of $\theta$ encodes this depends on the fixed ordering. Thus, adopting a graded lexicographic order ensures a reproducible indexing scheme without limiting model generality.

## C.4 Proof of Corollary 4.2

We now proceed to prove Cor. 4.2. For clarity, we restate the corollary below:

**Corollary 4.2** (Identifiability of invariant semantic variables). *Assume that Asms. 3.1 and 3.2 hold. Consider an OOD setting in which a subset of semantic variables, $\mathbb{I}_{inv} \subset \mathbb{I}_{\mathbf{s}}$, remains invariant between training and testing environments, while the remaining semantic variables, $\mathbb{I}_{var} = \mathbb{I}_{\mathbf{s}} \setminus \mathbb{I}_{inv}$, undergo distribution shifts. If the union of omitted and perturbable semantic variables under selection bias $\theta$ and perturbation bias $\rho$ coincides with the environment-sensitive subset, i.e., $\mathbb{I}_{var} = \mathbb{I}_{\theta}^c \cup \mathbb{I}_{\rho}$, then the invariant semantic variables $\mathbf{s}_{\mathbb{I}_{inv}}$ are block-identified via smooth functions $f_x : \mathcal{X} \to (0,1)^{|\mathbb{I}_{inv}|}$ and $f_t : \mathcal{T}^{(\theta)} \to (0,1)^{|\mathbb{I}_{inv}|}$, by minimizing $\mathcal{L}_{SymAlignMaxEnt}$.*

*Proof.* Under the OOD setting, and without loss of generality, let the subset of semantic variables susceptible to distribution shift be denoted by $\mathbb{I}_{var} = \mathbb{I}_{var}^1 \cup \mathbb{I}_{var}^2$. Suppose the index set associated with selection bias is given by $\mathbb{I}_{\theta} = \mathbb{I}_{var}^1 \cup \mathbb{I}_{inv}$, and that the perturbation bias acts on $\mathbb{I}_{\rho} = \mathbb{I}_{var}^1$. That is, the subset $\mathbb{I}_{var}^2$ is entirely omitted by the selection mechanism (i.e., excluded from $\mathbb{I}_{\theta}$), while the subset $\mathbb{I}_{var}^1$ is included but remains vulnerable to perturbation, as determined by $\rho$.

Given this structure, we directly apply the argument from the proof of Thm. 4.1. The omission of variables in $\mathbb{I}_{var}^2$, together with the perturbation of variables in $\mathbb{I}_{var}^1$, ensures that only the invariant subset $\mathbb{I}_{inv}$ is both selected and unperturbed. Therefore, the invariant semantic components $\mathbf{s}_{\mathbb{I}_{inv}}$ are block-identified via smooth functions

$$f_x : \mathcal{X} \to (0,1)^{|\mathbb{I}_{inv}|} \quad \text{and} \quad f_t : \mathcal{T}^{(\theta)} \to (0,1)^{|\mathbb{I}_{inv}|},$$

which attain the global minimum of the alignment objective.

This concludes the proof. $\qquad\square$

# D  Numerical Simulation Details

We provide additional details on the numerical simulations that are not fully covered in § 5.1. Specifically:

- In App. D.1, we outline the experimental setup, including hyperparameters, model architectures, and the construction of downstream tasks.
- In App. D.2, we present additional experiments that further validate our theoretical findings.
- In App. D.3, we analyze downstream performance under various perturbation bias settings.

## D.1  Detailed Experimental Setup

**Latent space construction.**  We define a latent space comprising a semantic subspace $\mathcal{S} \subseteq \mathbb{R}^{10}$ and two modality-specific subspaces $\mathcal{M}_x, \mathcal{M}_t \subseteq \mathbb{R}^5$. Variables are sampled from Gaussian priors:

$$\mathbf{s} \sim \mathcal{N}(0, \Sigma_{\mathbf{s}}), \quad \mathbf{m}_x \sim \mathcal{N}(0, \Sigma_{\mathbf{m}_x}), \quad \mathbf{m}_t \sim \mathcal{N}(0, \Sigma_{\mathbf{m}_t}).$$

For *independent* semantic variables, $\Sigma_{\mathbf{s}} = I_{10}$ yields a factorized standard Gaussian, aligning with nonlinear ICA assumptions [28, 63]. For *dependent* semantics, we sample $\Sigma_{\mathbf{s}}$ from Wishart distribution $\mathcal{W}_{10}(I, 10)$ (identity scale, 10 degrees of freedom), introducing latent dependencies as in [68, 13, 78]. Modality-specific covariances $\Sigma_{\mathbf{m}_x}$ and $\Sigma_{\mathbf{m}_t}$ are sampled similarly from $\mathcal{W}_5(I, 5)$, the Wishart distribution with identity scale and 5 degrees of freedom.

**Selection and perturbation biases.**  Given the 10-dimensional semantic space, the total number of nonempty subsets is $2^{10} - 1 = 1023$, defining the full range of *selection bias* configurations $\theta$. For tractability, we choose 10 representative selection subsets $\mathbb{I}_\theta$, detailed in Table 2.

To isolate *perturbation bias*, we fix full selection ($\theta = 1023$) and evaluate 10 representative perturbation subsets $\mathbb{I}_\rho$, as listed in Table 3. Perturbations are introduced by independently modifying each semantic variable $s_i$ in $\mathbb{I}_\rho$ with probability 0.75 using additive Gaussian noise:

$$\tilde{s}_i = s_i + \mathcal{N}(0, \Sigma_{\epsilon, i}),$$

where the joint noise covariance $\Sigma_\epsilon$ is sampled from $\mathcal{W}_{10}(I, 10)$. Perturbations are applied solely to the text modality, aligning the proposed LVM and assumptions stated in § 3.

In addition to the isolated bias settings, we consider a *joint bias* scenario to investigate compounding effects. Specifically, we select $\mathbb{I}_\theta = [8]$ (corresponding $\theta = 968$, excluding the last two semantic indices), and apply perturbations to the first two indices ($\mathbb{I}_\rho = [2]$, correspnding to $\rho = 12$).

Results across these 20 representative configurations (10 selection settings, 10 perturbation settings) and the joint bias case serve to validate the theoretical findings under a range of misalignment settings.

Table 2: Selection bias settings and selected semantic indices.

| $\theta, \rho = 1$ | 1 | 11 | 56 | 176 | 386 | 638 | 848 | 968 | 1013 | 1023 |
|---|---|---|---|---|---|---|---|---|---|---|
| $\mathbb{I}_\theta$ | {1} | [2] | [3] | [4] | [5] | [6] | [7] | [8] | [9] | [10] |

Table 3: Perturbation bias settings and perturbable semantic indices.

| $\rho \mid \theta = 1023$ | 1 | 2 | 12 | 57 | 177 | 387 | 639 | 849 | 969 | 1014 |
|---|---|---|---|---|---|---|---|---|---|---|
| $\mathbb{I}_\rho$ | $\emptyset$ | {1} | [2] | [3] | [4] | [5] | [6] | [7] | [8] | [9] |

**Parameter settings.**  We parameterize the generative functions $g_x$ and $g_{t(\theta)}$ using randomly initialized 3-layer invertible MLPs, following prior work [13]. Invertibility is enforced by maintaining a condition number threshold of $1e{-}3$ for each layer.

The encoding functions $f_x$ and $f_t$ are implemented as 7-layer MLPs and optimized using the Adam optimizer. For MMCL training, we use a batch size of 6144, a learning rate of $1e{-}4$, and train

for 100,000 iterations. The loss function is given by Eq. (2), with Euclidean distance used as the similarity metric and a temperature parameter set to 1.0. To ensure training stability, gradients are clipped using a maximum 2-norm of 2.

All experiments are run with three distinct random seeds, and we report averaged $R^2$ values, clipped to the interval $[0, 1]$ for interpretability.

**Downstream tasks design.** To evaluate pretrained representations under various bias conditions, we construct several downstream tasks. Specifically, four regression tasks are created by generating labels using complex nonlinear functions $f_{y_i}$, each applied to different subsets of the true semantic variables:

$$y_1 = f_{y_1}(\mathbf{s}_{[3]}), \quad y_2 = f_{y_2}(\mathbf{s}_{[5]}), \quad y_3 = f_{y_3}(\mathbf{s}_{[7]}), \quad y_4 = f_{y_4}(\mathbf{s}_{[9]}).$$

Each function $f_{y_{(\cdot)}} : \mathbb{R}^d \to \mathbb{R}$ includes quadratic, cubic, pairwise, and triple-wise interaction terms, as well as sinusoidal, logarithmic, and exponential transformations. The full formulation for a semantic vector $\mathbf{s}_{[d]}$ is:

$$f_{y_{(\cdot)}}(\mathbf{s}_{[d]}) = \sum_{i=1}^{d} s_i^2 + 0.3 \sum_{i=1}^{d} s_i^3 + 0.5 \sum_{1 \leq i < j \leq d} s_i s_j + 0.2 \sum_{1 \leq i < j < k \leq d} s_i s_j s_k$$

$$+ 0.7 \sum_{i=1}^{d} \left( \sin(s_i) + \cos(s_i) \right) + 0.4 \sum_{i=1}^{d} \log(1 + |s_i|) + 0.4 \sum_{i=1}^{d} e^{-|s_i|}.$$

For the classification task, labels are obtained by binarizing $y_2$ at its median value, which serves as the decision boundary. To simulate distribution shifts in the observations $\mathbf{x}$, we apply a skewed, heavy-tailed transformation to semantic dimensions 9 and 10:

$$s_i^{\text{ood}} = 2 \, \text{sign}(s_i^{\text{id}}) \cdot |s_i^{\text{id}}|^2, \quad \text{for } i \in \{9, 10\}.$$

For both downstream tasks, we *fix* the pretrained encoders and evaluate the quality of the learned representations using a two-layer MLP as a probing model. We generate 20,480 samples as the evaluation set for training the regressors and classifiers, along with an additional 20,480 samples as the in-distribution test set. To assess OOD generalization, we generate another 20,480 samples from the shifted latent space as the OOD test set.

The regressors are trained using Mean Squared Error (MSE) loss, and the classifiers use Cross-Entropy loss, both with a learning rate of $10^{-3}$ and trained for 10,000 steps. We report classification performance using the average Matthews Correlation Coefficient (MCC). Importantly, in the OOD setting, we perform *no adaptation or fine-tuning*; the classifier is evaluated directly to test the generalization capability of the pretrained representations.

## D.2 Additional Identification Results

**Results under perturbation bias.** As shown in Figure 11, the results of predicting true latent semantic variables under various perturbation bias settings exhibit similar trends to those observed under selection bias. Modality-specific variables are consistently discarded, unbiased semantic variables are faithfully block-identified, and misaligned semantic variables become partially identifiable in scenarios with dependent latent variables—demonstrating a consistent pattern across both image and text modalities. These findings further support and reinforce our theoretical analysis.

**Linearity of learned representations.** We further analyze the linearity of the identified semantic representations by reporting the $R^2$ scores obtained from linear regression applied to the learned features for predicting the true latent variables. As shown in Figure 12 (selection bias) and Figure 13 (perturbation bias), the performance of linear regression closely mirrors that of nonlinear regression reported in Figure 3. This strong correspondence suggests that the relationship between the learned representations and the true latent semantic variables is approximately linear, indicating that the identified representation subspace is nearly linear.

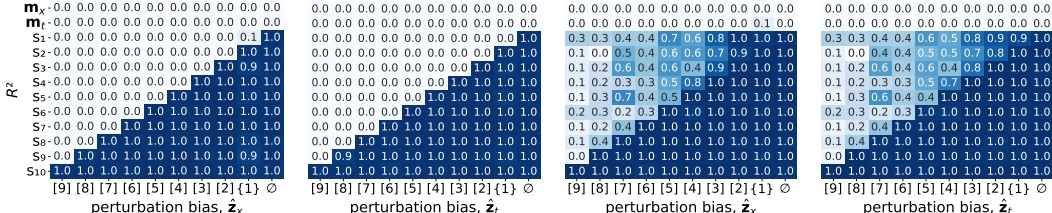

Figure 11: Mean $R^2$ scores under perturbation bias settings. Left to right: predictions using representations $\hat{\mathbf{z}}_x$ and $\hat{\mathbf{z}}_t$ under independent latent semantic variables, followed by those under dependent latent semantic variables.

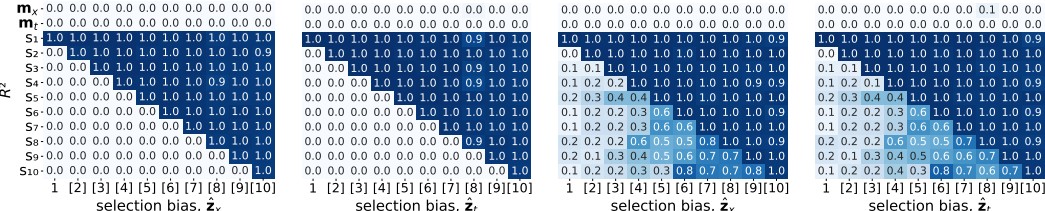

Figure 12: Evaluating linearity of learned representations under selection bias. Left to right: predictions using representations $\hat{\mathbf{z}}_x$ and $\hat{\mathbf{z}}_t$ under independent latent semantic variables, followed by those under dependent latent semantic variables.

**Ablation studies.** We perform two ablation studies to further examine the robustness of our theoretical findings.

First, we investigate the impact of assigning an incorrect representation dimension. Specifically, we consider a scenario in which the true dimension of the selected semantic variables is 3 (with selection $\mathbb{I}_\theta = [3]$), but we intentionally set the representation dimension to 5. As shown in Table 4, in the independent case, all selected semantic variables are successfully preserved, while omitted semantic variables are effectively discarded. In contrast, the dependent scenario yields significantly different patterns of $R^2$ scores compared to those obtained using the correct representation dimension (see Figures 3 and 12). These results suggest that redundant representation dimensions tend to encode exogenous noise, potentially introducing unnecessary complexity into the learned representations.

Second, we explore the joint effect of selection and perturbation biases by defining a scenario with selection bias $\mathbb{I}_\theta = [8]$ and perturbation bias $\mathbb{I}_\rho = [2]$. Results presented in Table 5 demonstrate that when both biases coexist, their effects on semantic identification remain consistent: semantic variables that are either omitted or perturbed are discarded, while unbiased semantic variables—those that are selected and yet unperturbed—are reliably preserved in the learned representations.

Together with previous results, these findings further reinforce our theoretical conclusions in Thm. 4.1.

Table 4: The effect of using an incorrect encoding size. The representation size is set to 5, whereas the true dimension should be 3. The biases are defined as $\mathbb{I}_\theta = [3]$ and $\mathbb{I}_\rho = \emptyset$.

| Setting | | Reps. | $R^2$ of Predicting Latent Semantic Variables under $\mathbb{I}_\theta = [3]$ and $\mathbb{I}_\rho = \emptyset$ | | | | | | | | | | | |
|---|---|---|---|---|---|---|---|---|---|---|---|---|---|---|
| | | | $s_1$ | $s_2$ | $s_3$ | $s_4$ | $s_5$ | $s_6$ | $s_7$ | $s_8$ | $s_9$ | $s_{10}$ | $\mathbf{m}_x$ | $\mathbf{m}_t$ |
| independ. | linear | $\hat{\mathbf{z}}_x$ | 0.98 | 0.96 | 0.98 | 0.01 | 0.03 | 0.03 | 0.02 | 0.04 | 0.01 | 0.02 | 0.02 | 0.00 |
| | | $\hat{\mathbf{z}}_t$ | 1.00 | 1.00 | 1.00 | 0.00 | 0.00 | 0.00 | 0.00 | 0.00 | 0.00 | 0.00 | 0.00 | 0.04 |
| | non-lin. | $\hat{\mathbf{z}}_x$ | 0.98 | 0.99 | 0.99 | 0.00 | 0.02 | 0.02 | 0.03 | 0.04 | 0.00 | 0.00 | 0.00 | 0.00 |
| | | $\hat{\mathbf{z}}_t$ | 1.00 | 1.00 | 1.00 | 0.00 | 0.00 | 0.00 | 0.00 | 0.00 | 0.00 | 0.00 | 0.00 | 0.06 |
| dependent | linear | $\hat{\mathbf{z}}_x$ | 0.98 | 0.99 | 0.99 | 0.14 | 0.23 | 0.16 | 0.10 | 0.22 | 0.42 | 0.36 | 0.01 | 0.00 |
| | | $\hat{\mathbf{z}}_t$ | 1.00 | 1.00 | 1.00 | 0.13 | 0.20 | 0.13 | 0.09 | 0.20 | 0.41 | 0.34 | 0.00 | 0.01 |
| | non-lin. | $\hat{\mathbf{z}}_x$ | 0.99 | 1.00 | 0.99 | 0.13 | 0.22 | 0.16 | 0.15 | 0.20 | 0.43 | 0.36 | 0.02 | 0.00 |
| | | $\hat{\mathbf{z}}_t$ | 1.00 | 1.00 | 1.00 | 0.13 | 0.20 | 0.13 | 0.09 | 0.20 | 0.42 | 0.35 | 0.00 | 0.05 |

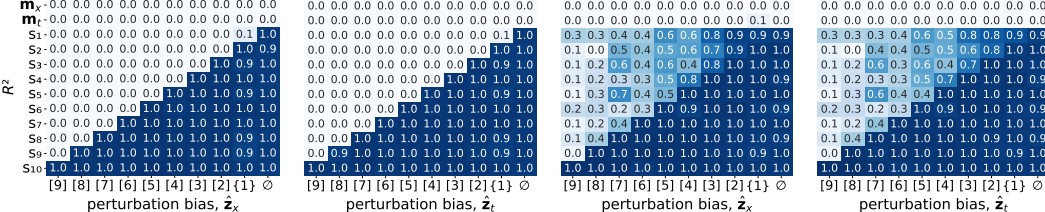

Figure 13: Linearity of learned representations under perturbation bias. Left to right: predictions using representations $\hat{\mathbf{z}}_x$ and $\hat{\mathbf{z}}_t$ under independent latent semantic variables, followed by those under dependent latent semantic variables.

Table 5: Coexistence of both selection and perturbation biases. The biases are defined as $\mathbb{I}_\theta = [8]$ and $\mathbb{I}_\rho = [2]$.

| Setting | | Reps. | $R^2$ **of Predicting Latent Semantic Variables** ($\mathbb{I}_\theta = [8]$, $\mathbb{I}_\rho = [2]$) | | | | | | | | | | | |
|---|---|---|---|---|---|---|---|---|---|---|---|---|---|---|
| | | | $s_1$ | $s_2$ | $s_3$ | $s_4$ | $s_5$ | $s_6$ | $s_7$ | $s_8$ | $s_9$ | $s_{10}$ | $\mathbf{m}_x$ | $\mathbf{m}_t$ |
| independ. | linear | $\hat{\mathbf{z}}_x$ | 0.00 | 0.01 | 0.97 | 0.98 | 0.97 | 0.95 | 0.99 | 0.98 | 0.00 | 0.00 | 0.00 | 0.00 |
| | | $\hat{\mathbf{z}}_t$ | 0.00 | 0.00 | 0.98 | 0.98 | 0.98 | 0.98 | 0.99 | 0.98 | 0.00 | 0.00 | 0.00 | 0.00 |
| | non-lin. | $\hat{\mathbf{z}}_x$ | 0.00 | 0.01 | 0.98 | 0.98 | 0.98 | 0.98 | 0.96 | 0.99 | 0.98 | 0.00 | 0.00 | 0.00 |
| | | $\hat{\mathbf{z}}_t$ | 0.00 | 0.00 | 0.99 | 0.99 | 0.99 | 0.99 | 0.99 | 0.99 | 0.99 | 0.00 | 0.00 | 0.00 |
| dependent | linear | $\hat{\mathbf{z}}_x$ | 0.67 | 0.57 | 0.97 | 0.99 | 0.99 | 0.98 | 0.99 | 0.97 | 0.63 | 0.63 | 0.00 | 0.00 |
| | | $\hat{\mathbf{z}}_t$ | 0.64 | 0.53 | 0.98 | 0.97 | 0.99 | 0.98 | 0.99 | 0.98 | 0.61 | 0.60 | 0.00 | 0.00 |
| | non-lin. | $\hat{\mathbf{z}}_x$ | 0.68 | 0.57 | 0.99 | 0.99 | 0.99 | 0.99 | 0.99 | 0.99 | 0.64 | 0.64 | 0.00 | 0.00 |
| | | $\hat{\mathbf{z}}_t$ | 0.65 | 0.53 | 0.99 | 0.99 | 0.99 | 0.99 | 0.99 | 0.99 | 0.61 | 0.61 | 0.00 | 0.00 |

### D.3 Additional Downstream Results

We further report downstream task performance under varying perturbation bias settings. Specifically, the preserved semantic variables are sequentially reversed—starting from semantic index 10 and incrementally expanding until the full semantic set is included. The results shown in Figure 14 indicate that, in general, semantic variables critical to downstream tasks must be preserved in the learned representations to achieve high performance. This observation holds across both independent and dependent latent semantic settings.

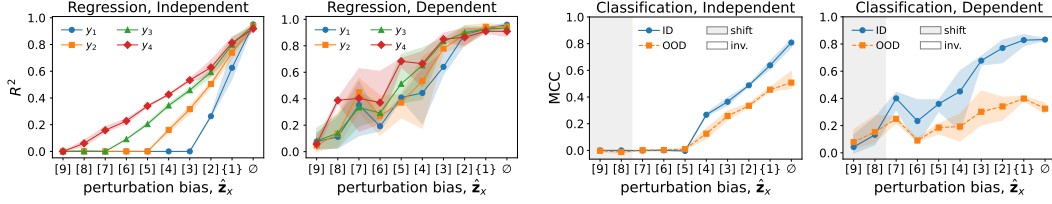

Figure 14: Downstream performance of pretrained representations $\hat{\mathbf{z}}_x$ under perturbation bias. Top: in-distribution (ID) regression performance. Bottom: ID classification and out-of-distribution (OOD) generalization.

## E    Experiment Details on MPI3D-Complex Dataset

We provide additional details on the MPI3D-Complex dataset that are not fully covered in § 5.2. In App. E.1, we comprehensively describe the experimental setup, including a dataset overview, the selection and perturbation bias configurations used to generate text, model architecture, and training parameters. In App. E.2, we present additional results, including the use of a linear classifier for predicting latent factors and an ablation study on encoder dimensionality.

### E.1 Detailed Experimental Setup

**MPI3D-Complex dataset.** MPI3D-Complex [19] contains 460,800 real-world images of resolution $64 \times 64 \times 3$, spanning all combinations of seven *mutually independent, discrete factors* (see Table 6). We designate horizontal and vertical positions (`hori.`, `vert.`) as image-specific due to their low-level visual nature, while treating the remaining five as semantic variables for representation learning and evaluation purposes.

**Text latent factors.** Text descriptions are generated from the ground-truth semantic attributes using content-word mappings (see Table 6). Under unbiased settings, all five semantic attributes are included. For text-specific variation, a generation template is chosen via a discrete latent variable $m_t \sim \mathrm{Uniform}([3])$, each corresponding to a different human-written sentence structure.

Table 6: Latent variables of MPI3D-Complex and corresponding content words in text.

| Factor Name | Distribution | Content Words |
|---|---|---|
| Object color (`color`) | $\mathrm{Uniform}(\{0,\ldots,3\})$ | `yellow`, `green`, `olive`, `red` |
| Object shape (`shape`) | $\mathrm{Uniform}(\{0,\ldots,3\})$ | `coffee-cup`, `tennis-ball`, `croissant`, `beer-cup` |
| Object size (`size`) | $\mathrm{Uniform}(\{0,1\})$ | `small`, `large` |
| Camera height (`cam.`) | $\mathrm{Uniform}(\{0,\ldots,2\})$ | `top`, `center`, `bottom` |
| Background color (`back.`) | $\mathrm{Uniform}(\{0,\ldots,2\})$ | `purple`, `sea-green`, `salmon` |
| Horizontal axis (`hori.`) | $\mathrm{Uniform}(\{0,\ldots,39\})$ | — (image-specific factor) |
| Vertical axis (`vert.`) | $\mathrm{Uniform}(\{0,\ldots,39\})$ | — (image-specific factor) |

**Text generation under misalignment settings.** We generate text for each image under various selection and perturbation bias settings to investigate the effects of misalignment. To ensure computational tractability—since exhaustively enumerating all possible configurations is both infeasible and unnecessary—we adopt a progressively incremental strategy for introducing biases into the latent semantic variables, as outlined in Table 7.

In selection bias settings (where the perturbable set $\mathbb{I}_\rho$ is empty), textual descriptions include only the content words corresponding to a subset of the true image semantic variables. We define five incremental settings: ①: {`color`}, ②: {`color`, `shape`}, ③: {`color`, `shape`, `size`}, ④: {`color`, `shape`, `size`, `cam.`}, ⑤: all five attributes. The specific text generation templates associated with $g_{t^{(\theta)}}$ under each selection setting are detailed in Table 8.

In perturbation bias settings (where all semantic indices are selected), we apply random substitutions to a subset $\mathbb{I}_\rho$. For each factor in $\mathbb{I}_\rho$, its text value is replaced with a randomly sampled alternative with probability 0.9. Five perturbation configuration are used: ①: $\emptyset$, ②: {`back.`}, ③: {`cam.`, `back.`}, ④: {`size`, `cam.`, `back.`}, ⑤: {`shape`, `size`, `cam.`, `back.`}.

Table 7: Misalignment settings for text generation of MPI3D-Complex dataset.

| Setting | Selection Bias, $\mathbb{I}_\theta$ | Perturbation Bias, $\mathbb{I}_\rho$ |
|---|---|---|
| ① | {`color`} | $\emptyset$ |
| ② | {`color`, `shape`} | {`back.`} |
| ③ | {`color`, `shape`, `size`} | {`cam.`, `back.`} |
| ④ | {`color`, `shape`, `size`, `cam.`} | {`size`, `cam.`, `back.`} |
| ⑤ | {`color`, `shape`, `size`, `cam.`, `back.`} | {`shape`, `size`, `cam.`, `back.`} |

**Training details.** For each setting, the dataset is partitioned into training, evaluation, and test subsets in a fixed ratio of 44,720 : 23,040 : 23,040. Across all configurations, we train the image and

Table 8: Text generation templates for MPI3D-Complex dataset for different selection settings.

| Setting | Text Generation Templates for Each Selection View |
|---------|---------------------------------------------------|
| ① | "An object colored {color}." 
 "It has a {color} appearance." 
 "Something with {color}." |
| ② | "A {shape} that is {color}." 
 "The {color} {shape}." 
 "An object shaped like a {shape}, colored {color}." |
| ③ | "A {size} {shape} in {color}." 
 "{color}, {size}, {shape}." 
 "The object is {size}, shaped as a {shape}, and colored {color}." |
| ④ | "A {size}{shape} in {color}, seen from {cam.}." 
 "Viewed from {cam.}, a {color}, {size} {shape}." 
 "A {size} {shape} with {color}, perspective: {cam.}." |
| ⑤ | "A {size} {shape} in {color}, viewed from {cam.}, {back.}." 
 "From {cam.}, you see a {color}, {size} {shape}, {back.}." 
 "A {size} {shape}, {color}, placed {back.}, observed from {cam.}." |

text encoders for 200,000 steps using the training subset, averaging results over three random seeds. The training objective is the multimodal contrastive loss $\mathcal{L}_{\text{MMCL}}$, defined in Eq. (1), and optimized using the Adam optimizer with an initial learning rate of $1 \times 10^{-5}$, a batch size of 256, and a temperature parameter $\tau = 1.0$.

For image encoding, we use a ResNet18 backbone followed by a fully connected layer with a fixed input dimensionality of 100. The output dimensionality is adjusted according to the number of unbiased semantic factors under each bias setting.

For text encoding, we tokenize text using the `nltk.PunktTokenizer`, following the procedure in [13]. Tokenized sequences are transformed into two-dimensional one-hot embeddings and processed using a convolutional neural network (CNN) with a variable number of layers, determined by the shape of the tokenized input. The output dimensionality of the CNN is configured to match that of the image encoder, ensuring compatibility in the joint representation space.

The encoding size is generally set to match the number of unbiased semantic dimensions, i.e., $\dim(\mathbb{I}_\rho^c)$. However, when this number is less than 3, we set the encoding size to 3 to ensure minimal representational capacity. An ablation study on this design choice is provided in the following section.

**Evaluation metrics.** After training the representations, we freeze the encoders and train both a linear classifier (logistic regression) and a nonlinear classifier (a two-layer MLP with ReLU activation) for each setting and each image latent factor. Classifiers are trained on the evaluation subset for 10,000 steps.

Performance is assessed on the test set using the Matthews Correlation Coefficient (MCC), computed separately for each latent factor. MCC is chosen for its robustness in evaluating binary classification performance under class imbalance.

### E.2 Additional Results

**Linearity of learned representations.** We evaluate the linear separability of the learned representations by reporting MCC scores for predicting each latent factor using a linear classifier. As shown in Figure 15, and in comparison to the nonlinear results in Figure 6, the findings reveal that certain latent semantic variables—most notably `size`—are not linearly embedded in the learned representation space when training image-text pairs with MMCL.

In contrast, factors such as `color`, `shape`, `cam.`, and `back.` exhibit strong linear separability, suggesting that these semantic variables are linearly represented by the learned image and text encoders.

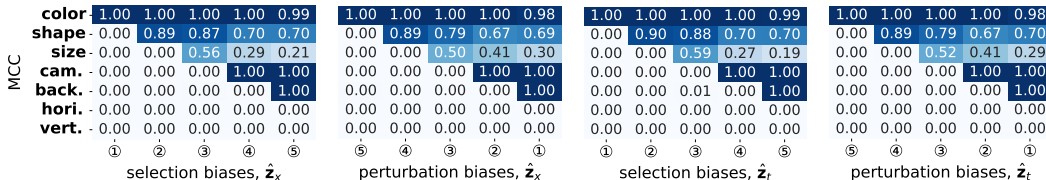

Figure 15: Evaluating linearity of learned representations under misalignment. Left to right: image features $\hat{\mathbf{z}}_x$ under selection bias, image features $\hat{\mathbf{z}}_x$ under perturbation bias, text features $\hat{\mathbf{z}}_t$ under selection bias, and text features $\hat{\mathbf{z}}_t$ under perturbation bias.

**Ablations on the encoding size.**    We conduct an ablation study on the encoding size by evaluating a selection bias setting in which only the semantic attribute {color} is selected for text generation. All other training parameters are kept consistent with those used in our main experiments, except for the encoding dimensionality.

As shown in Figure 16, in contrast to training with purely numerical data, learning from image-text data exhibits sensitivity to the choice of encoding size. Specifically, when the encoding size is set to 1—exactly matching the number of perfectly aligned semantic dimensions—the image encoder tends to be under-optimized, resulting in high variance across runs. However, increasing the encoding size leads to more stable and reliable performance.

Notably, in the presence of independent latent factors, the additional (redundant) encoding dimensions do not appear to capture misaligned semantic variables, suggesting that excess capacity does not harm identifiability in this setting.

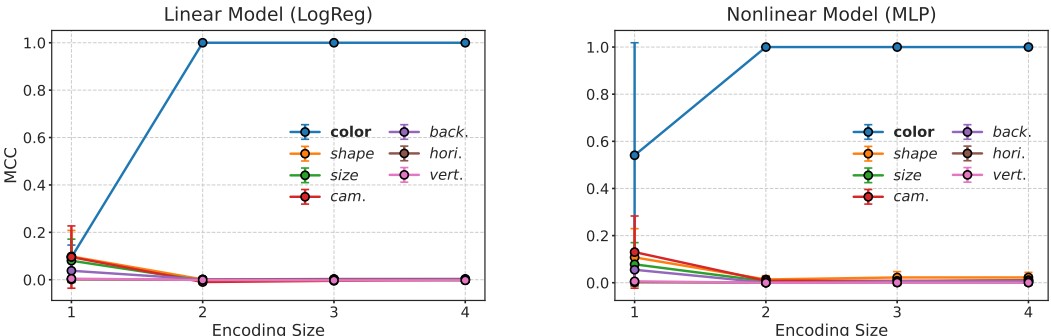

Figure 16: Ablation study on encoding size under a selection bias setting where only the {color} attribute is included in text generation. Left: average MCC over three runs using a linear classifier. Right: average MCC using a nonlinear classifier.

## F   Experiment Details on Causal3DIdent Dataset

We provide additional details on the Causal3DIdent dataset that are not fully covered in § 5.3. In App. F.1, we offer a comprehensive description of the experimental setup, including the image and text latent factors, the image generation process, the design of selection and perturbation bias settings for text generation, and training configurations. In App. F.2, we present supplementary results, including analyses of the learned text representations and assessments of the linearity of the learned representations.

### F.1   Detailed Experimental Setup

**Image latent factors and image generation.**    Following prior work [83, 68, 13, 78], we utilize the *Causal3DIdent* dataset to synthesize images from a predefined latent causal structure. Images are generated using the Blender renderer [9], which applies a complex rendering function parameterized by 11 input variables. In our configuration, the object's $z$-position is fixed, leaving 10 latent factors that govern image generation.

These include 3 discrete variables—object shape (shape), and object positions along the horizontal (x_pos) and vertical (y_pos) axes—and 7 continuous variables: object color (color), spotlight position (s_pos) and color (s_color), background color (b_color), and the three object rotation angles (alpha, beta, gamma). We treat the rotation angles (alpha, beta, gamma) as image-specific latent variables, while the remaining factors are considered semantic latent variables, structured according to the causal graph shown in Figure 17.

We synthesize 80,000 samples for MMCL training, 10,000 samples for classifier or regressor training, and another 10,000 samples for test-time evaluation. Images are rendered at a resolution of $128 \times 128 \times 3$.

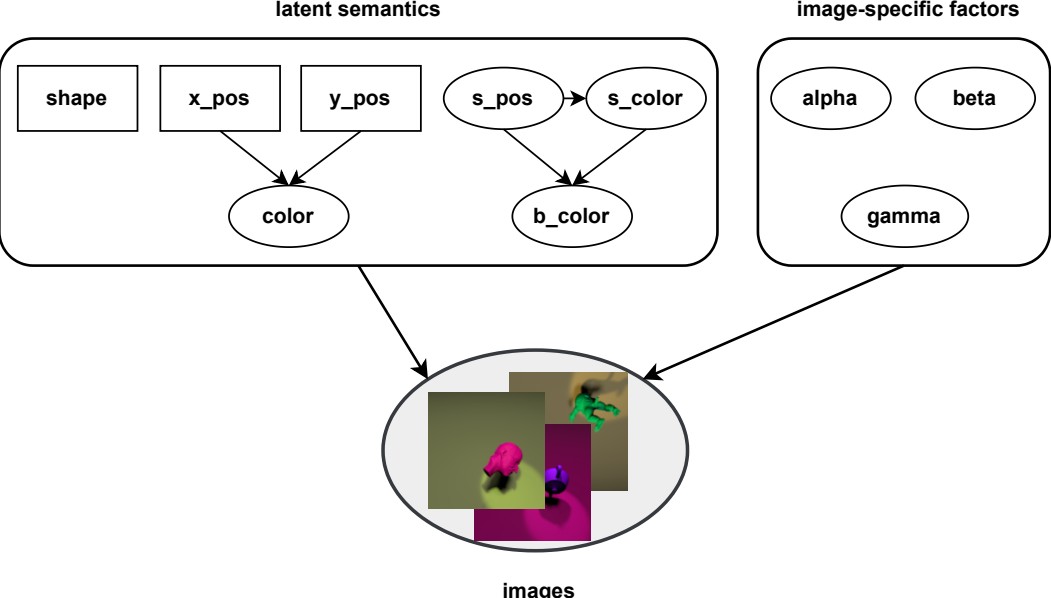

Figure 17: Latent causal model governing image generation in the Causal3DIdent dataset. Rectangular nodes represent discrete latent random variables, while elliptical nodes denote continuous ones. Object shape (shape), horizontal and vertical position (x_pos, y_pos), object color (color), spotlight position and color (s_pos, s_color), and background color (b_color) are treated as latent semantic variables shared across modalities and potentially subject to misalignment. In contrast, the rotation angles—alpha, beta, and gamma—are considered image-specific latent factors.

**Text latent factors.** We discretize the continuous variables color, s_color, and b_color using sampled image semantic variables mapped to distinct color palettes: TABLEAU_COLORS for color, CSS4_COLORS for s_color, and XKCD_COLORS for b_color. While s_color remains continuous in the underlying latent representation, we simulate partial information loss for the spotlight position (s_pos) during the generating mapping to form text. As a result, the generated textual descriptions of s_pos do not constitute an information-preserving transformation.

To introduce text-specific variation, we employ five manually designed templates to generate text from the latent factors under each bias setting, adapting the text rendering pipeline from [13]. A complete list of latent factors and their types is provided in Table 9.

**Text generation under different misalignment settings.** Following the MPI3D-Complex experiments, we explore a series of incrementally increasing selection and perturbation bias configurations to introduce varying degrees and types of cross-modal misalignment. These settings enable a systematic investigation of how different forms of alignment impact representation learning. Each configuration is indexed using circled numerals and summarized in Table 10.

For the perturbable semantic variables in each perturbation setting, we randomly sample the corresponding text semantic values. Specifically, for discrete variables such as x_pos and y_pos, we

Table 9: Latent factors of Causal3DIdent and corresponding content words in text.

| Factor Name | Image Modality | Text Modality | Content Words |
|---|---|---|---|
| shape | $\text{Uniform}(\{0, \ldots, 6\})$ | $\text{Uniform}(\{0, \ldots, 6\})$ | `teapot`, `hare`, `dragon`, `cow`, `armadillo`, `horse`, `head` |
| x_pos | $\text{Uniform}(\{0, 1, 2\})$ | $\text{Uniform}(\{0, 1, 2\})$ | `left`, `center`, `right` |
| y_pos | $\text{Uniform}(\{0, 1, 2\})$ | $\text{Uniform}(\{0, 1, 2\})$ | `top`, `mid`, `bottom` |
| s_pos | $\text{Uniform}([0, 1])$ | $\text{Uniform}([0, 1])$ | `northwest`, `northeast`, `center`, `southwest`, `southeast` |
| color | $\frac{1}{6}(\texttt{x\_pos} + \texttt{y\_pos}) + \frac{1}{3}\text{Uniform}([0, 1])$ | Up to 10 colors | Color names in `TABLEAU_COLORS` |
| s_color | $\frac{1}{2}(\texttt{s\_pos} + \text{Uniform}([0, 1]))$ | Up to 147 colors | Color names in `CSS4_COLORS` |
| b_color | $\frac{1}{3}(\texttt{s\_pos} + \texttt{s\_color} + \text{Uniform}([0, 1]))$ | Up to 954 colors | Color names in `XKCD_COLORS` |
| alpha | $\text{Uniform}([0, 1]))$ | — | — |
| beta | $\text{Uniform}([0, 1]))$ | — | — |
| gamma | $\text{Uniform}([0, 1]))$ | — | — |
| phrase | — | $\text{Uniform}(\{0, \ldots, 5\}))$ | — |

sample uniformly from the set $\{0, 1, 2\}$; for continuous variables, we sample uniformly from the interval $[0, 1]$.

Table 11 provides the text generation templates associated with each selection setting. Representative image–text pairs generated under different selection and perturbation bias configurations are shown in Figure 18.

Table 10: Misalignment settings for text generation of Causal3DIdent dataset.

| Setting | Selection Bias, $\mathbb{I}_\theta$ | Perturbation Bias, $\mathbb{I}_\rho$ |
|---|---|---|
| ① | {shape} | {x_pos, y_pos, s_pos, color, s_color, b_color} |
| ② | {shape, x_pos} | {y_pos, s_pos, color, s_color, b_color} |
| ③ | {shape, x_pos, y_pos} | {s_pos, color, s_color, b_color} |
| ④ | {shape, x_pos, y_pos, s_pos} | {color, s_color, b_color} |
| ⑤ | {shape, x_pos, y_pos, s_pos, color} | {s_color, b_color} |
| ⑥ | {shape, x_pos, y_pos, s_pos, color, s_color} | {b_color} |
| ⑦ | {shape, x_pos, y_pos, s_pos, color, s_color, b_color} | $\emptyset$ |

**Training details.** Across all experimental settings, we train the image and text encoders for 100,000 steps on the training subset, using three different random seeds to ensure robustness. The training objective is the multimodal contrastive loss $\mathcal{L}_{\text{MMCL}}$, defined in Eq. (1), and optimized using the Adam optimizer with an initial learning rate of $1 \times 10^{-5}$, a batch size of 256, and a temperature parameter $\tau = 1.0$.

For both the image and text encoders, we adopt the same architectures used in the MPI3D-Complex experiments. The encoding dimensionality is adjusted according to the bias setting: for selection settings ① through ⑦, the encoding sizes are set to 3, 3, 4, 5, 5, 6, and 7, respectively; for perturbation settings ① through ⑦, the encoding sizes are assigned in reverse order: 7, 6, 5, 5, 4, 3, and 3.

**Evaluation metrics.** After training the representations, we freeze the encoders and, for each bias setting, train both a linear classifier (logistic regression) and a nonlinear classifier (a two-layer MLP with ReLU activation) for each discrete latent factor. Similarly, for continuous latent factors, we train both a linear regressor and a nonlinear regressor (a two-layer MLP with ReLU activation). All classifiers and regressors are trained for 10,000 steps using the evaluation subset.

We assess the predictive performance of the learned representations by evaluating their ability to recover the ground-truth latent factors corresponding to their respective modalities. This evaluation accounts for the fact that some semantic variables may appear in discrete or continuous form, depending on the modality and rendering process.

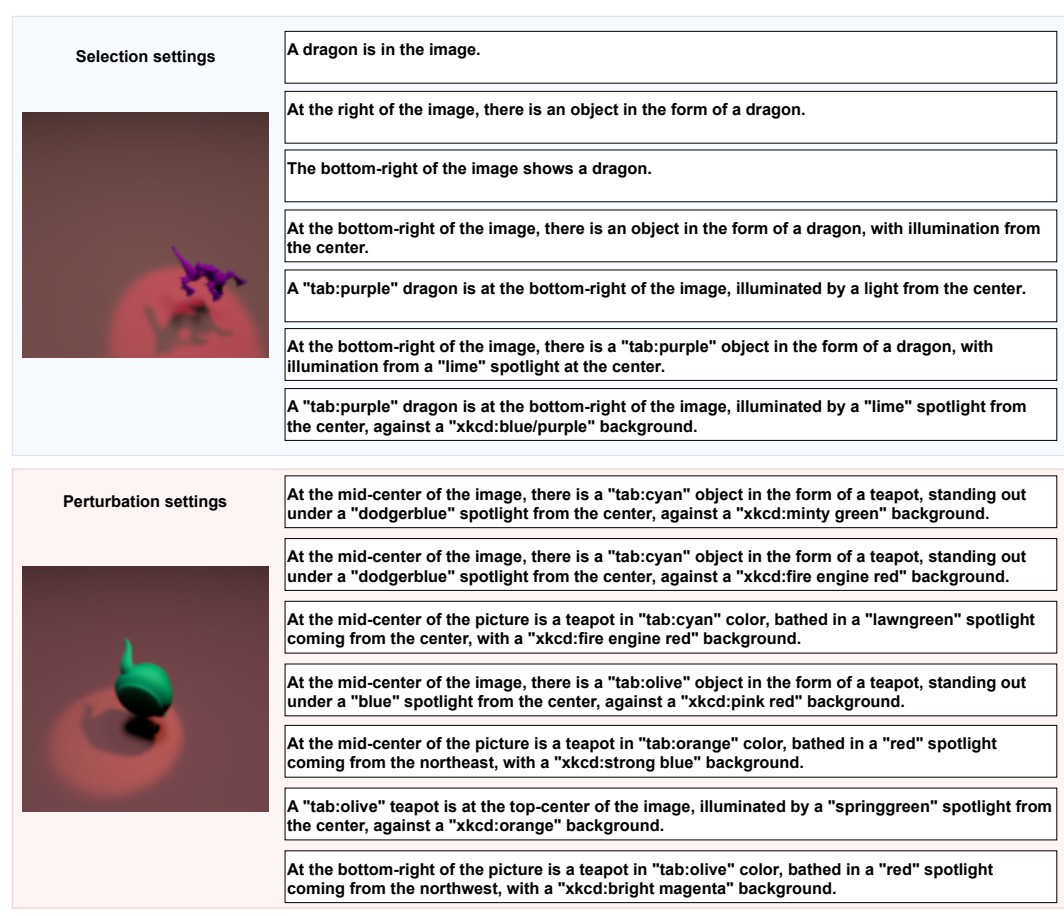

| Selection settings | A dragon is in the image. |
| | At the right of the image, there is an object in the form of a dragon. |
| | The bottom-right of the image shows a dragon. |
| | At the bottom-right of the image, there is an object in the form of a dragon, with illumination from the center. |
| | A "tab:purple" dragon is at the bottom-right of the image, illuminated by a light from the center. |
| | At the bottom-right of the image, there is a "tab:purple" object in the form of a dragon, with illumination from a "lime" spotlight at the center. |
| | A "tab:purple" dragon is at the bottom-right of the image, illuminated by a "lime" spotlight from the center, against a "xkcd:blue/purple" background. |

| Perturbation settings | At the mid-center of the image, there is a "tab:cyan" object in the form of a teapot, standing out under a "dodgerblue" spotlight from the center, against a "xkcd:minty green" background. |
| | At the mid-center of the image, there is a "tab:cyan" object in the form of a teapot, standing out under a "dodgerblue" spotlight from the center, against a "xkcd:fire engine red" background. |
| | At the mid-center of the picture is a teapot in "tab:cyan" color, bathed in a "lawngreen" spotlight coming from the center, with a "xkcd:fire engine red" background. |
| | At the mid-center of the image, there is a "tab:olive" object in the form of a teapot, standing out under a "blue" spotlight from the center, against a "xkcd:pink red" background. |
| | At the mid-center of the picture is a teapot in "tab:orange" color, bathed in a "red" spotlight coming from the northeast, with a "xkcd:strong blue" background. |
| | A "tab:olive" teapot is at the top-center of the image, illuminated by a "springgreen" spotlight from the center, against a "xkcd:orange" background. |
| | At the bottom-right of the picture is a teapot in "tab:olive" color, bathed in a "red" spotlight coming from the northwest, with a "xkcd:bright magenta" background. |

Figure 18: Example image-text pairs from Causal3DIdent under different selection bias settings. The left panel shows randomly selected images; the right panel presents the corresponding text from top to bottom, each generated under selection settings ① to ⑦ with no perturbations, and perturbation bias ⑦ to ① with full selections.

Prediction performance is measured on the test set using the Matthews Correlation Coefficient (MCC) for discrete factors and the coefficient of determination ($R^2$) for continuous factors. Metrics are computed separately for each latent factor.

### F.2 Additional Results

**Results of text representations.**    We now turn to the analysis of the text representations learned by MMCL. As shown in Figure 19, we observe patterns similar to those found in the image modality. In particular, discrete latent semantic variables—such as shape, x_pos, and y_pos—are reliably identified, provided they are unbiased and consistently aligned across modalities. A notable case is s_pos, which is a continuous latent factor in both modalities but is mapped to only five discrete tokens in the text observations, rendering the text generation process non-invertible. Despite this lossy transformation, the model achieves a relatively high $R^2$, suggesting that the learned text representations remain strongly influenced by the alignment objective, even when semantic information is partially lost.

For other latent semantic variables that are continuous in the image modality but discretized in the text modality—such as color, s_color, and b_color—we observe varying degrees of performance degradation. This drop in performance is likely attributable not only to the quantization of continuous values but also to semantic ambiguity introduced during text generation. Notably, all three attributes correspond to different aspects of color, yet they may be described using overlapping vocabulary drawn from distinct color palettes. For instance, tab:cyan from TABLEAU_COLORS refers to object

Table 11: Text generation templates for Causal3DIdent dataset under different selection settings.

| Setting | Text Generation Templates |
|---|---|
| ① | "A {shape} is visible."
"A {shape} is in the image."
"The image shows a {shape}."
"The picture is a {shape}."
"There is an object in the form of a {shape}." |
| ② | "A {shape} is visible, positioned at the {x_pos} of the image."
"A {shape} is at the {x_pos} of the image."
"The {x_pos} of the image shows a {shape}."
"At the {x_pos} of the picture is a {shape}."
"At the {x_pos} of the image, there is an object in the form of a {shape}." |
| ③ | "A {shape} is visible, positioned at the {y_pos}-{x_pos} of the image."
"A {shape} is at the {y_pos}-{x_pos} of the image."
"The {y_pos}-{x_pos} of the image shows a {shape}."
"At the {y_pos}-{x_pos} of the picture is a {shape}."
"At the {y_pos}-{x_pos} of the image, there is an object in the form of a {shape}." |
| ④ | "A {shape} is visible, positioned at the {y_pos}-{x_pos}, with a spotlight shining from {s_pos}."
"A {shape} is at the {y_pos}-{x_pos}, illuminated by a light from {s_pos}."
"The {y_pos}-{x_pos} shows a {shape}, highlighted by a light from {s_pos}."
"At the {y_pos}-{x_pos} is a {shape}, under a light from {s_pos}."
"There is a {shape} at {y_pos}-{x_pos}, lit from {s_pos}." |
| ⑤ | "A {shape} of {color} color is visible at {y_pos}-{x_pos}, with a spotlight from {s_pos}."
"A {color} {shape} is at {y_pos}-{x_pos}, lit from {s_pos}."
"The area {y_pos}-{x_pos} shows a {color} {shape}, under a light from {s_pos}."
"A {color} {shape} is illuminated at {y_pos}-{x_pos} from {s_pos}."
"A {color} object shaped like a {shape} is lit from {s_pos}." |
| ⑥ | "A {color} {shape} is lit by a {s_color} spotlight from {s_pos}, at {y_pos}-{x_pos}."
"At {y_pos}-{x_pos}, a {color} {shape} is under a {s_color} light from {s_pos}."
"The {shape} is {color}, under a {s_color} light at {s_pos}."
"A {color} {shape} under a {s_color} spotlight at {s_pos}, located at {y_pos}-{x_pos}."
"A {color} {shape} stands under a {s_color} light from {s_pos}." |
| ⑦ | "A {color} {shape} under a {s_color} spotlight at {s_pos}, with a {b_color} background, at {y_pos}-{x_pos}."
"At {y_pos}-{x_pos}, a {color} {shape} is under a {s_color} light from {s_pos}, against a {b_color} background."
"A {color} {shape} appears at {y_pos}-{x_pos}, lit by {s_color} from {s_pos}, with {b_color} background."
"The scene shows a {color} {shape} under {s_color} lighting at {s_pos}, with a {b_color} backdrop."
"A {color} object shaped like a {shape}, under a {s_color} spotlight at {s_pos}, with a {b_color} background." |

color (color), cyan from CSS4_COLORS describes spotlight color (s_color), and xkcd:cyan from XKCD_COLORS indicates background color (b_color). Despite referencing different latent variables, these tokens all contain the word cyan, which is tokenized identically by nltk.PunktTokenizer, resulting in ambiguity in the text observations. These findings highlight the importance of using distinct and unambiguous content words when representing semantically different concepts in multi-modal learning—particularly when the text modality is not treated merely as an auxiliary input for visual representation learning.

Interestingly, the identification of x_pos and y_pos in the text representations does not lead to improved predictability of color, in contrast to what is often observed in the image modality. This aligns with our theoretical expectation that perturbation biases disrupt the underlying causal structure in the image latent space.

Regarding the text-specific factor `phrase`, we find it to be partially encoded in the learned representations. This contrasts with the image-specific continuous factors, which are consistently omitted. The partial identifiability of `phrase` is likely attributable to its discrete nature, which violates the conditions typically required for modality-specific factors to be excluded—consistent with findings reported in [13]. Moreover, this effect appears more pronounced under selection bias settings, particularly when the encoding dimensionality exceeds the true dimensionality of the unbiased semantic subspace.

Overall, the behavior of the learned text representations provides empirical support for our theoretical analysis, even under conditions where certain modeling assumptions are relaxed or violated.

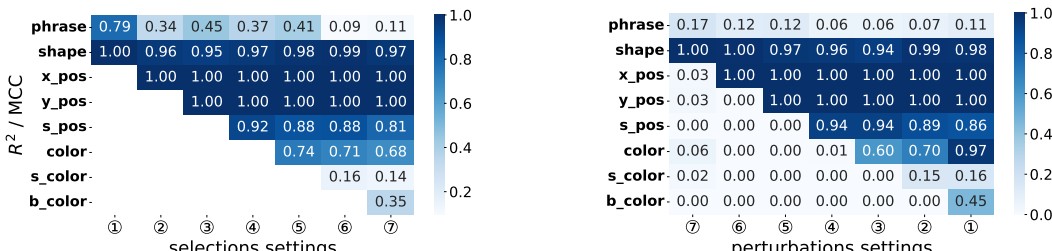

Figure 19: Prediction of latent semantic variables under misalignment with text features. $R^2$ for continuous and MCC for discrete factors. Left: Selection bias. Right: Perturbation bias.

**Linearity of learned representations.** We assess the linear separability of the learned image and text representations by evaluating the performance of linear classifiers and regressors trained to predict each latent factor. As shown in Figure 20 and Figure 21, and in comparison to the nonlinear prediction results, the findings indicate that not all latent semantic variables are linearly embedded in the representations learned through MMCL. In particular, the semantic factor `color` consistently demonstrates poor linear predictability, suggesting that it is encoded nonlinearly in both modalities. Conversely, factors such as `x_pos` and `y_pos` exhibit strong linear separability in certain settings, indicating that these semantic variables are more directly captured in the latent space of both the image and text encoders. Overall, these results suggest that the linearity of the learned representations is factor-dependent and shaped by both modality-specific encoding strategies and the underlying structure of the input data—highlighting an important direction for future investigation.

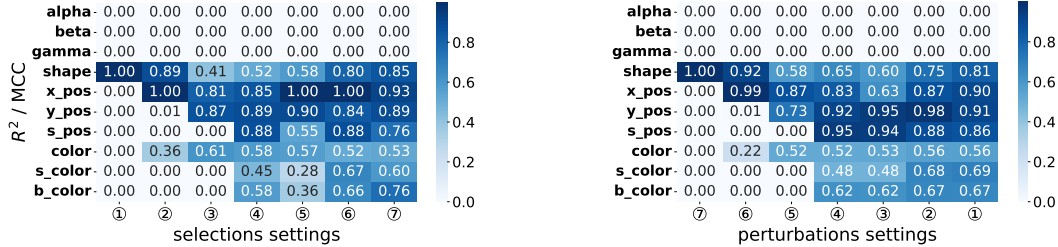

Figure 20: Evaluating linearity of image features under misalignment settings. $R^2$ for continuous and MCC for discrete factors. Left: Selection bias. Right: Perturbation bias.

# G Details on OpenCLIP Case Study

## G.1 Concepts Taxonomy and Data Collection

**Concepts taxonomy.** To validate the representations of pretrained OpenCLIP models, we curated a set of 146 concepts organized into 15 distinct concept groups. Each concept group is treated as a separate solution space, and we aim to ensure that the concepts within each group are mutually exclusive. Table 12 presents all the concepts used in the case study. For simplicity, abbreviations are used in the main text, with their full meanings provided in the concept group column.

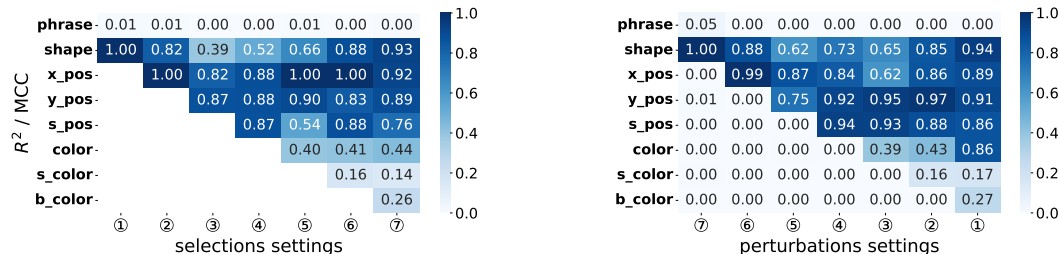

Figure 21: Evaluating linearity of text features under misalignment settings. $R^2$ for continuous and MCC for discrete factors. Left: Selection bias. Right: Perturbation bias.

Table 12: Taxonomy of concept categories with abbreviations, grouped by conceptual relevance.

| Group | Explanation | Concept Name |
|---|---|---|
| Animal | animals | dog, cat, horse, bird, elephant, giraffe, cow, zebra, rabbit, duck |
| Clothing | clothing | shirt, pants, dress, shoes, hat, jacket, skirt, tie, hoodie, socks |
| Color | colors | red, blue, green, yellow, black, white, orange, pink, gray, brown |
| Food | food | pizza, burger, sandwich, salad, cake, coffee, tea, beer, ice cream, noodles |
| Object | object | chair, table, phone, laptop, car, bottle, bag, cup, backpack, television |
| Role | roles | chef, teacher, athlete, doctor, engineer, artist, pilot, firefighter, police officer, lawyer |
| Scene | scenes | beach, kitchen, forest, street, park, office, bedroom, classroom, stadium, playground |
| Vehicle | vehicles | bus, truck, airplane, train, motorcycle, bicycle, boat, van, taxi, scooter |
| Weather | weathers | sandstorm, drought, rain, snow, fog, rainbow, storm, moonlight, overcast, frost |
| Texture | texture | glossy, matte, rough, smooth, fuzzy, silky, grainy, wrinkled, slippery, furry |
| POV | viewpoint | close-up, wide shot, top-down, low angle, high angle, aerial, shallow-depth, long exposure |
| Emot. | mental state | tired, focused, surprised, proud, shy, bored, confused, excited, thoughtful, nervous |
| Postproc. | post-procssing | hdr, desaturated, color-filtered, over-sharpened, saturated, sepia, heavily edited, bokeh, tilt-shift |
| Trait | moral bias | easy-going, cold-hearted, evil, criminal, immoral, pure, corrupt, hero, cheater, deserving |
| Stere. | stereotypes | exotic look, ethnic look, slacker, aggressive look, gangster look, terrorist look, illegal look, privileged, ghetto look |

**Data collection.** We collected up to 200 images per concept by using the concept names as search queries on Flickr, filtering for content licensed under Creative Commons. To ensure relevance, all images were manually reviewed, and those deemed unrelated to the queried concept were removed. The remaining images were retained for subsequent experiments. Table 13 summarizes the number of images retained for each concept. It is important to note that some concept groups, such as those related to stereotypes, include terms that may reflect societal biases. We do not endorse or perpetuate these biases; rather, these concepts are included solely to assess whether pretrained models encode or reproduce such biased associations in their representations.

### G.2 Details on Caption Frequency Statistics

To assess caption coverage frequency for each concept, we first enumerated plausible synonyms for every concept name. We then computed the percentage of captions in the LAION-400M dataset that included either the original concept term or one of its synonyms. Figure 9 in § 5.4 presents the group-wise average coverage rates.

Table 13: Concept-wise image counts grouped by concept category for zero-shot evaluation.

| Group | Concept | Count | Group | Concept | Count | Group | Concept | Count |
|---|---|---|---|---|---|---|---|---|
| Animal | dog | 124 | Animal | cat | 112 | Animal | horse | 109 |
| Animal | bird | 106 | Animal | elephant | 98 | Animal | giraffe | 90 |
| Animal | cow | 88 | Animal | zebra | 81 | Animal | rabbit | 77 |
| Animal | duck | 65 | Clothing | shirt | 113 | Clothing | pants | 104 |
| Clothing | dress | 101 | Clothing | shoes | 96 | Clothing | hat | 92 |
| Clothing | jacket | 87 | Clothing | skirt | 84 | Clothing | tie | 78 |
| Clothing | hoodie | 72 | Clothing | socks | 68 | Color | red | 130 |
| Color | blue | 127 | Color | green | 119 | Color | yellow | 114 |
| Color | black | 111 | Color | white | 109 | Color | orange | 97 |
| Color | pink | 91 | Color | gray | 88 | Color | brown | 82 |
| Food | pizza | 118 | Food | burger | 113 | Food | sandwich | 107 |
| Food | salad | 101 | Food | cake | 95 | Food | coffee | 93 |
| Food | tea | 90 | Food | beer | 85 | Food | ice cream | 82 |
| Food | noodles | 78 | Emot. | tired | 58 | Emot. | focused | 55 |
| Emot. | surprised | 51 | Emot. | proud | 49 | Emot. | shy | 45 |
| Emot. | bored | 44 | Emot. | confused | 42 | Emot. | excited | 40 |
| Emot. | thoughtful | 39 | Emot. | nervous | 36 | Trait | easy-going | 29 |
| Trait | cold-hearted | 27 | Trait | evil | 25 | Trait | criminal | 24 |
| Trait | immoral | 23 | Trait | pure | 22 | Trait | corrupt | 20 |
| Trait | hero | 18 | Trait | cheater | 17 | Trait | deserving | 15 |
| Object | chair | 120 | Object | table | 115 | Object | phone | 112 |
| Object | laptop | 110 | Object | car | 108 | Object | bottle | 105 |
| Object | bag | 100 | Object | cup | 97 | Object | backpack | 93 |
| Object | television | 90 | Postproc. | hdr | 33 | Postproc. | desaturated | 32 |
| Postproc. | color-filtered | 31 | Postproc. | over-sharpened | 30 | Postproc. | saturated | 28 |
| Postproc. | sepia | 27 | Postproc. | heavily edited | 26 | Postproc. | bokeh | 24 |
| Postproc. | tilt-shift | 22 | Role | chef | 88 | Role | teacher | 86 |
| Role | athlete | 85 | Role | doctor | 83 | Role | engineer | 80 |
| Role | artist | 78 | Role | pilot | 74 | Role | firefighter | 71 |
| Role | police officer | 68 | Role | lawyer | 66 | Scene | beach | 120 |
| Scene | kitchen | 115 | Scene | forest | 110 | Scene | street | 107 |
| Scene | park | 104 | Scene | office | 102 | Scene | bedroom | 98 |
| Scene | classroom | 95 | Scene | stadium | 92 | Scene | playground | 89 |
| Stere. | exotic look | 21 | Stere. | ethnic look | 20 | Stere. | slacker | 19 |
| Stere. | aggressive look | 18 | Stere. | gangster look | 17 | Stere. | terrorist look | 16 |
| Stere. | illegal look | 15 | Stere. | privileged | 14 | Stere. | ghetto look | 13 |
| Texture | glossy | 47 | Texture | matte | 45 | Texture | rough | 43 |
| Texture | smooth | 41 | Texture | fuzzy | 40 | Texture | silky | 38 |
| Texture | grainy | 36 | Texture | wrinkled | 34 | Texture | slippery | 33 |
| Texture | furry | 30 | Vehicle | bus | 106 | Vehicle | truck | 104 |
| Vehicle | airplane | 102 | Vehicle | train | 100 | Vehicle | motorcycle | 97 |
| Vehicle | bicycle | 93 | Vehicle | boat | 89 | Vehicle | van | 86 |
| Vehicle | taxi | 83 | Vehicle | scooter | 80 | POV | close-up | 55 |
| POV | wide shot | 52 | POV | top-down | 50 | POV | low angle | 48 |
| POV | high angle | 47 | POV | aerial | 44 | POV | shallow-depth | 42 |
| POV | long exposure | 39 | Weather | sandstorm | 66 | Weather | drought | 63 |
| Weather | rain | 61 | Weather | snow | 58 | Weather | fog | 56 |
| Weather | rainbow | 53 | Weather | storm | 50 | Weather | moonlight | 47 |
| Weather | overcast | 45 | Weather | frost | 42 | | | |

To explore intra-group variability, Figure 22 provides a more granular view of concept-level coverage. It reveals that even within a single concept group, the degree of caption coverage can vary substantially across individual concepts. As expected, commonly observable concepts, such as object types, colors, and food items, tend to appear more frequently in captions. In contrast, abstract or subjective concepts, such as mental states, post-processing artifacts, moral judgments, and social stereotypes, are significantly underrepresented. This discrepancy may arise from several factors: ethical concerns leading annotators to avoid certain terms, unconscious omission of nuanced details, or general challenges in describing abstract visual content.

### G.3 Zero-Shot Evaluation Details

To probe the representations of pretrained OpenCLIP models, we evaluate two variants trained on the LAION-400M dataset: ViT-B-32 and ViT-L-14. Using the collected image samples, we perform zero-shot analysis by employing the concept names as text prompts. Each concept group is treated as a distinct solution space. To mitigate the impact of sampling bias in the test dataset, we report macro-averaged F1 scores at the group level.

As shown in Figure 10, concept groups that are frequently captioned—defined here as those with coverage rates exceeding 0.15%, are generally well represented in the multimodal contrastive learning (MMCL) models, with F1 scores above 65%. In contrast, concept groups that are severely under-captioned, often due to selection bias or ethical omission, exhibit poor representation, with F1 scores falling below 40%. The results are consistent across different model scales, further reinforcing our theoretical findings, particularly the impact of selection bias on the learned representations.

Figure 23 presents the intra-group confusion matrix from the zero-shot evaluation using the OpenCLIP ViT-B-32 model trained on the LAION-400M dataset. As clearly shown, well-captioned concept groups exhibit strong diagonal patterns, indicating accurate predictions—whereas concept groups affected by selection bias suffer from substantial misclassification.

This misclassification has multifaceted implications. For certain under-captioned yet potentially valuable concepts, such as `Texture` (texture) and `Emot.` (mental state), the poor performance suggests a need for more comprehensive and semantically rich captioning pipelines in multimodal contrastive learning (MMCL) pretraining. Conversely, for sensitive concepts like `Trait` (moral judgement) and `Stere.` (stereotypes), which ideally should not be encoded, this under-representation may be desirable. In these cases, the inherent biases in the captioning process may function as epistemic filters that implicitly align model representations with human ethical standards.

## H   Computation Resources

All experiments were conducted on a high-performance computing cluster equipped with 4×NVIDIA A100 GPUs (40 GB each), running CUDA 12.2 and driver version 535.161.07. The system also included an AMD EPYC 7313 16-core processor and 503 GB of RAM. For the numerical simulations, we trained over 120 models in total, requiring approximately 70 GPU-hours across 4 GPUs. On the MPI3D-Complex dataset, we trained 36 models, consuming approximately 27 GPU-hours. For the Causal3DIdent dataset, we trained 42 models, which required roughly 25 GPU-hours across 4 GPUs. Additionally, we generated 100,000 synthetic images for the Causal3DIdent dataset using Blender. Rendering was performed over four days on a separate workstation equipped with an AMD Ryzen 7 7700X 8-core processor (4.50 GHz) and a single NVIDIA RTX 4090 GPU (24 GB).

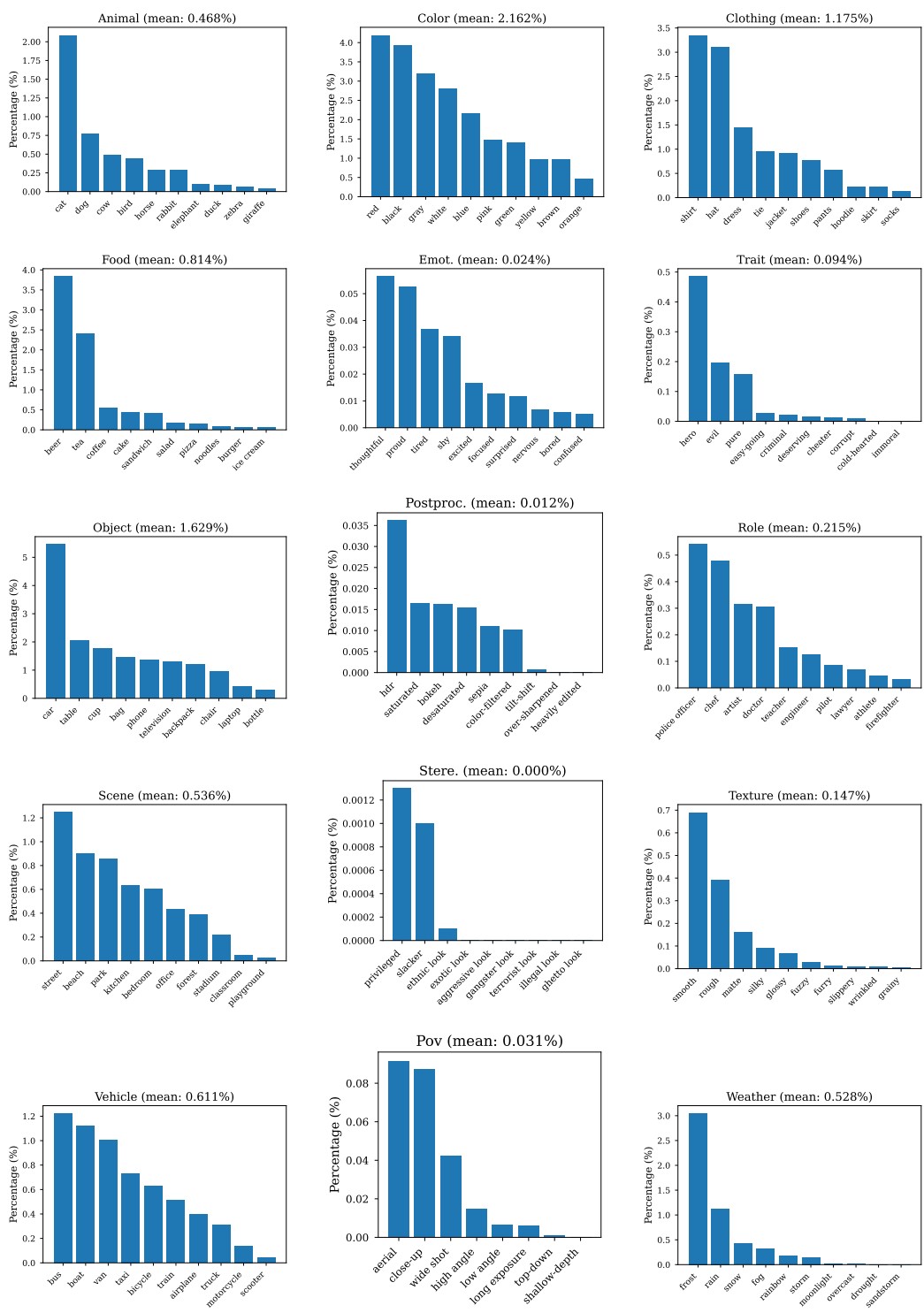

Figure 22: Statistics of concept-level coverage within each group in LAION-400M captions.

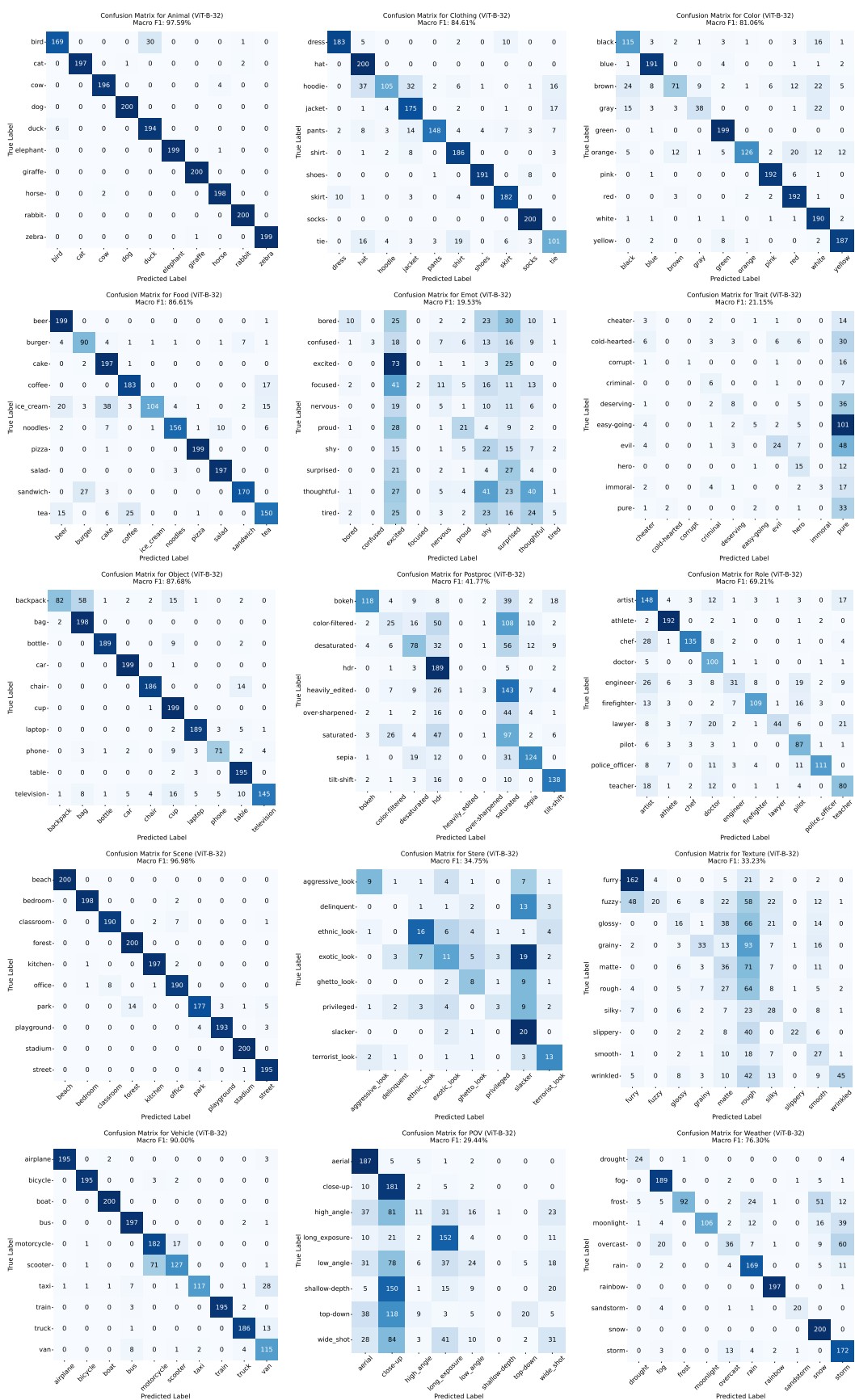

Figure 23: Confusion matrix for zero-shot prediction using OpenCLIP ViT-B-32 of each group.

# I Discussions

In this section, we reflect on the limitations of the current study, propose future research directions informed by our findings, and discuss broader implications.

## I.1 Limitations and Future Directions

**Estimating dataset biases.** Our study focuses on formalizing the effects of semantic misalignment, but does not directly address the empirical estimation of dataset-specific biases such as selection and perturbation bias. These biases are often latent and difficult to observe, particularly in large-scale, web-curated datasets. Yet, identifying and quantifying them is essential for improving interpretability, data quality, and downstream robustness. This task involves challenges in defining concept taxonomies, determining annotation granularity, and choosing evaluation metrics. Our framework suggests practical strategies: selection bias can be diagnosed via concept coverage statistics over a predefined vocabulary, while perturbation bias may be estimated by comparing dataset captions with those generated by strong image-to-text models, using semantic similarity or factual consistency metrics. We view the development of robust bias identification pipelines, such as those explored by [79], as a valuable direction for future work.

**Expanding the taxonomy of misalignment.** Multimodal data presents a wide range of potential misalignments across modalities. In this work, we deliberately focus on semantic misalignment in image-text pairs, a particularly prevalent and influential form of bias in vision-language learning. This type of misalignment is central to current multimodal research due to its relevance in large-scale pretraining paradigms and real-world applications (e.g., CLIP, ALIGN). By explicitly modeling selection and perturbation biases, our framework captures common semantic omissions and distortions that significantly affect representational quality. Extending the analysis to other forms of misalignment, such as temporal lag in video-language data, modality-specific dropout, or ambiguity arising from multi-entity co-occurrence, poses additional challenges and may require substantially different modeling assumptions and techniques. We view our proposed abstraction as a tractable and principled foundation for understanding semantic misalignment and believe it can work as a building-block for future explorations on this direction.

**Modeling cross-modal semantic synergy.** Multimodal representations often exhibit semantic synergies arising not only from the alignment of shared content but also from emergent meaning generated through cross-modal integration. At a theoretical level, this aligns with ideas in grounded cognition: cognition (and by extension, semantic representation) is rooted in the integration of perceptual, motor, and introspective modalities via *simulation* [3]. This parallels our formal distinction between two forms of synergy: i) Some semantics may be deterministically inferable across modalities—such as emotional tone from facial expression, which our latent-space alignment framework readily accommodates. 2) Other higher-order semantics (e.g., sarcasm, irony, humor) emerge through non-additive, nonlinear interactions between modalities and thus require modeling joint-modality latent variables. While our current theory does not explicitly model these emergent semantic phenomena, establishing semantic invariance under structured misalignment is a necessary precursor. Extending our framework to capture structured, emergent cross-modal semantics remains a promising avenue for future research.

**Formulating missing data in the latent space.** Our current framework addresses misalignment arising primarily from fixed selection and perturbation biases in textual annotations. However, real-world datasets, particularly large-scale, user-generated corpora such as LAION-5B [61], often display random and unstructured semantic omissions. Extending our latent variable model to accommodate such random missingness (e.g., missing completely at random or missing not at random) poses both theoretical and practical challenges. Future research should examine the identifiability consequences of randomly missing semantic variables and establish conditions that ensure partial or probabilistic recovery of latent factors remains feasible.

**Linearity of multimodal representations.** Although our theoretical analysis guarantees identifiability of unbiased semantic factors up to general invertible transformations, empirical observations suggest learned representations are often approximately linear with respect to the underlying semantics. This motivates an important open question: under what conditions can identifiability up to a

*linear* transformation be rigorously established, without imposing overly restrictive assumptions on data-generating processes or latent structures? Clarifying this issue is not only theoretically significant but also practically beneficial, as linear representations enhance interpretability and simplify downstream applications. Future studies might investigate training objectives, regularization strategies, or architectural inductive biases explicitly designed to encourage linearity. Bridging this empirical regularity with robust theoretical guarantees represents a promising research frontier.

## I.2 Broader Implications

**Linguistic relativity and epistemic constraints in multimodal AI.** Our findings offer computational support for a contemporary perspective on linguistic relativity [70], which posits that language shapes human perception and conceptualization. In multimodal AI, linguistic supervision implicitly determines which aspects of the visual world are foregrounded, and which are omitted. Selection and perturbation biases in textual annotations thus act as *epistemic filters*, constraining the conceptual space that models can represent and generalize over. This reframes dataset design as both an epistemic and normative act: inclusion or omission in annotations encodes a stance on salience, relevance, or appropriateness. Two practical insights follow. First, achieving generalizable representations requires supervision that faithfully captures the intended semantic scope without systematic omissions. Second, curating data to intentionally include or exclude ethically sensitive or socially salient content provides a mechanism for aligning learned representations with human values [4, 11]. When annotation is treated as an epistemic commitment, dataset design becomes a central tool for shaping both semantic fidelity and ethical alignment.

**Human annotation biases as signals of implicit value judgments.** Beyond explicit annotation content, human annotation errors, such as consistent omissions or mislabels, reveal subtle but powerful behavioral signals. These errors are not uniformly random; rather, they concentrate in dimensions that humans intuitively deprioritize under limited attention [25]. Core attributes like risk or intentionality are rarely mislabeled, while peripheral or low-salience details are more frequently neglected. Such patterns implicitly encode a *value hierarchy* over semantic factors. This perspective invites a reinterpretation: rather than discarding annotation errors as noise, we can study them as reflections of human value structure. Systematically neglected factors often carry little perceived cost when missed, suggesting lower social or cognitive importance. These regularities form a behavioral prior, informing models not only what remains invariant to environment changes, but what is *valued*. Incorporating such cues enables a richer form of alignment grounded in human preferences, priorities, and ethical sensibilities [17].

