# OpenReview forum: "On the Value of Cross-Modal Misalignment in Multimodal Representation Learning"
_NeurIPS.cc/2025/Conference — NeurIPS 2025 spotlight_

### Official Review · Reviewer_mroJ · 2025-06-27

**Clarity:** 3
**Significance:** 3
**Originality:** 3
**Rating:** 4
**Confidence:** 4

**Summary:**

This work proposes a Latent Variable Model (LVM) for resolving Cross-Modal Misalignment problem, which explicitly characterizes misalignment through two mechanisms: Selection bias and Perturbation bias. It makes the theoretical analysis is made to validate the  the representations learned by MMCL framwork can capture the semantic variables invariant to selection and perturbation bias. Experiments on synthetic and real-world image-text datasets (e.g., MPI3D-Complex, Causal3DIdent) demonstrate the efectiveness of the proposed method.

**Questions:**

See weakness part.

**Ethical Concerns:**

["NO or VERY MINOR ethics concerns only"]

**Final Justification:**

The additional experiments and discussion address my concerns, and I maintain a positive score.

**Limitations:**

Yes

**Paper Formatting Concerns:**

No Paper Formatting Concerns

**Quality:**

3

**Strengths And Weaknesses:**

Strengths:
1. A clear and reasonable framwork is proposed to disentangle the unbiased and biased factors for cross-modal representation learning, where sufficient theoretical analysis is given to surport the framwork.
2. The source code is attached that enhance the reproducibility and reliability of this work.

Weakness:
1. This study solely does not include experimental designs to apply this framework to more common downstream tasks (e.g., image-text matching), which undermines the work's persuasiveness and limits its broader impact.

2. This experiments of this paper only validates the efficetiveness of the proposed framework on distinguishing the aligned and misaligned factors of iamge-text pairs, whilst not designing experiments tp apply this framework to resolve more prevalent related downstream tasks, such image-text matching. It weakens the persuasiveness and broader impcat of this work.

---

> ### Author Rebuttal · Authors · 2025-07-29
>
> We thank you for your review and for recognizing the value of our theoretical framework and reproducibility. We address your comments and clarify our contributions below.
>
> ---
> > **W1** *"This study solely does not include experimental designs to apply this framework to more common downstream tasks (e.g., image-text matching), which undermines the work's persuasiveness and limits its broader impact."*
>
> We appreciate your concern regarding conventional downstream tasks such as image-text matching. However, we would like to clarify that the primary goal of this paper is not to introduce a new training method or optimize task performance, but to provide a principled theoretical framework for understanding how selection and perturbation biases shape semantic representations in multimodal contrastive learning. Specifically, our work offers a formal identifiability analysis of how multimodal misalignment filters semantic content during learning.
>
> To support this goal, our original experiments are intentionally designed around controlled, factorized datasets (e.g., MPI3D, Causal3DIdent) where ground-truth latent variables are known. This setting allows for direct empirical validation of the theoretical claims in Theorem 4.1 and its corollaries, without introducing additional confounding factors.
>
> Nonetheless, to address your concern regarding practical applicability, we conducted an additional illustrative analysis using OpenCLIP trained on LAION-400M. In this case study, we perform zero-shot image-to-text matching across 146 visual concepts grouped into 15 semantic categories (e.g., color, object, emotion). For each concept, we collected up to 200 CC-licensed Flickr images and measured the frequency of concept mentions in LAION-400M captions to estimate selection bias. We report the average F1 scores using OpenCLIP ViT-B/32 and ViT-L/14:
>
> ||Color|Object|Clothing|Food|Vehicle|Scene|Weather|Animal|Role|Texture|Trait|Viewpoint|Emotion|Postprocess|Stereotype|
> |-|-|-|-|-|-|-|-|-|-|-|-|-|-|-|-|
> |Average Coverage (%)|2.1622|1.6288|1.1747|0.8140|0.6114|0.5365|0.5284|0.4682|0.2147|0.1470|0.0935|0.0312|0.0238|0.0118|0.0003|
> |ViT-B/32 F1 Score (%)|81.1|87.7|84.6|86.6|90.0|97.0|76.3|97.6|69.2|33.2|21.2|29.4|19.5|41.8|34.8|
> |ViT-L/14 F1 Score (%)|79.0|89.5|86.6|91.0|90.1|94.6|75.9|98.9|66.4|37.1|19.4|31.0|20.1|44.1|40.2|
>
> These results reveal a clear trend: visual concepts with low caption coverage suffer from significantly lower retrieval performance, consistent with our theoretical predictions about the role of semantic fidelity and coverage in MMCL. This provides further evidence that our framework is applicable and predictive in real-world settings. We plan to include a brief version of this additional case study in the final version to strengthen practical relevance.
>
> ---
> > **W2** *"This experiments of this paper only validates the efficetiveness of the proposed framework on distinguishing the aligned and misaligned factors of iamge-text pairs, whilst not designing experiments tp apply this framework to resolve more prevalent related downstream tasks, such image-text matching. It weakens the persuasiveness and broader impcat of this work."*
>
> We appreciate your perspective and agree that further downstream evaluations can provide additional insight into practical relevance. However, we would like to clarify that the primary objective of this work is not to introduce a new training pipeline or method related to downstream tasks, but to develop a principled theoretical framework that explains how multimodal misalignment shapes learned representations in MMCL, with a focus on identifiability under selection and perturbation biases.
>
> Our contributions lie in the domain of representation learning theory—formulating and proving identifiability results under semantic misalignment. To connect this theory to practice, we derive implications via Corollaries 4.1 and 4.2, which illustrate how different forms of misalignment can help or hinder generalization depending on the semantics affected. These insights offer conceptual guidance for dataset and method design in real-world multimodal learning.
>
> While we conducted a case study in our previous response validating pretrained OpenCLIP representations, the design of downstream systems, such as image-text matching methods informed by our theoretical framework, is a valuable direction but remains outside the scope of this work. We see it as a promising avenue for future research building upon our theoretical foundation.
>
> ----
> Thank you again for your time and for motivating this additional clarification and analysis. We hope that our responses address your concerns.

---

> > ### Comment · Reviewer_mroJ · 2025-08-06
> > **Comment**
> >
> > Thank you for your detailed response. The additional experiments and discussion address my concerns, and I will maintain a positive score.

---

> > > ### Author Response · Authors · 2025-08-06
> > > **Thanks**
> > >
> > > Thank you for your timely feedback. We’re pleased that the additional experiments and discussion addressed your concerns. We again sincerely appreciate the time you dedicated to reviewing our work.

---

> ### Author Response · Authors · 2025-08-06
> **Inviting Further Discussion**
>
> Dear Reviewer mroJ,
>
> Thank you again for reviewing our work. As the discussion phase progresses, we wanted to kindly follow up in case you had any further thoughts or questions on our rebuttal.
>
> We would be happy to clarify any remaining concerns and would greatly appreciate your engagement in the discussion.
>
> Best regards,
> Authors of Submission 642

---

### Official Review · Reviewer_t7nT · 2025-07-02

**Clarity:** 4
**Significance:** 4
**Originality:** 3
**Rating:** 5
**Confidence:** 2

**Summary:**

Leveraging multimodal contrastive learning (MMCL) and a latent variable model (LVM), the authors investigate the problem of cross-modal misalignment and propose a unified perspective by introducing two key mechanisms: selection bias and perturbation bias. Through a thorough theoretical analysis, they demonstrate that MMCL learns representations that retain only the semantic information invariant to these biases. The work provides both theoretical foundations and practical insights into how misalignment impacts multimodal representation learning.

**Questions:**

1. On page 3, two loss functions are introduced: L_MMCL and L_SymAlignMaxEnt. However, in the experimental setup (page 7, section 5.1, line 262), L_SymAlignMaxEnt is used for the first experiment, while L_MMCL is used in the second experiment (page 8, section 5.2, line 298). It would be helpful to clarify the rationale behind choosing different loss functions for these experiments, and to specify any differences in their expected outcomes or suitability for each setting.

2. In Figure 4 (page 7), the X-axis is somewhat unclear. Adding a more detailed explanation in the figure description on page 8 would greatly improve clarity for readers trying to interpret the results.

3. According to the experiment setups (lines 261, 301, and 324), it appears that all models are based on simple MLP architectures. I’m curious whether the proposed theoretical insights hold under more complex models, such as transformer-based architectures. Exploring this would strengthen the practical relevance of the findings.

4. On page 3, in the section on Latent Space (lines 121–122), it is stated that text-specific variables are independent of semantic variables $s$, with “grammar” given as an example. However, some grammatical features—such as verb tense—can carry semantic meaning. This raises the question of whether certain text-specific variables are inherently entangled with semantic information. A brief discussion or clarification would be valuable here.

**Ethical Concerns:**

["NO or VERY MINOR ethics concerns only"]

**Final Justification:**

The concerns I raised have all been addressed. I believe the author has put significant effort into this work, and from my perspective, it is worthy of acceptance.

**Limitations:**

I suggest including a discussion on the limitations of the assumptions required by the proposed theorems.

**Quality:**

3

**Strengths And Weaknesses:**

Strengths:
1. The paper presents a comprehensive theoretical analysis of the cross-modal misalignment problem, offering a unified perspective that reconciles two contrasting views: (1) that misalignment can lead to hallucinations and degraded performance in multimodal models, and (2) that deliberately introducing misalignment—particularly in style-related information—can enhance robustness in representation learning.
2. The explanation of the core idea is clear, and the illustration in Figure 1 makes the concept easy to understand.
3. There are extensive experiments conducted to verify the proposed theorems.

Weaknesses:
1. Since this paper aims to unify opposing views on cross-modal misalignment—and one of these views argues that misalignment can lead to “hallucination” in multimodal models—it would be valuable to include a more detailed analysis, preferably theoretical, of the hallucination aspect to strengthen the argument.

2. In the theoretical results, the assumption that the generative mappings are diffeomorphisms (e.g., as stated on page 4, line 147) may be overly idealized and unrealistic in practical settings. A discussion of this assumption’s implications and possible relaxations would enhance the paper’s applicability.

---

> ### Author Rebuttal · Authors · 2025-07-29
>
> We sincerely thank you for your thoughtful review and recognition of our work’s theoretical depth, clarity, and significance. Your constructive suggestions on modeling assumptions, latent structure, and presentation are valuable and will help strengthen the paper. Below, we address your questions point by point.
>
> ---
> > **W1** *"... one of these views argues that misalignment can lead to 'hallucination' ... it would be valuable to include a more detailed analysis ..."*
>
> We thank you for surfacing this important line of inquiry. While our current theoretical framework does not explicitly model hallucination, we view it as a consequential practical symptom of unresolved cross-modal misalignment. In MMCL-based models, hallucination can manifest in two prominent scenarios:
>
> - When important visual attributes are omitted or rarely mentioned (selection bias) or inconsistently described across samples (perturbation bias), the resulting representations may fail to retain these factors. In practice, if the learned representation has higher capacity than the subspace of invariant semantics, the remaining dimensions may be filled with spurious or dataset-specific correlations. These can surface as hallucinated or non-factual semantic information during generation or matching.
>
> - When key semantic variables are consistently mislabeled in captions (e.g., both sea lions and seals captioned as "seal"), MMCL can learn falsely grounded associations—that is, associations that are consistently aligned across modalities but semantically incorrect. These may propagate into downstream tasks, producing outputs that reflect dataset artifacts rather than visually grounded meaning.
>
> In essence, hallucination arises when training captions lack semantic fidelity, either through omission or distortion, especially for concepts expected to generalize at test time. As emphasized in Corollary 4.1, high-fidelity alignment is crucial for learning grounded, reliable representations, offering an implicit explanation for when hallucination is likely to occur.
>
> A rigorous theoretical treatment of hallucination would require extending our framework to include decoding mechanisms and generative modeling dynamics. While non-trivial, we view this as an exciting direction for future work and will include this discussion in the final version.
>
> ---
> > **W2** *"... the assumption that the generative mappings are diffeomorphisms ... may be overly idealized ..."*
>
> We agree that the assumption of diffeomorphic generative mappings can be idealized in practice. We adopt this assumption primarily for theoretical tractability, especially without assuming parametric latent distributions or structural priors. Below, we discuss both its implications and potential relaxations:
>
>  - Implications: The diffeomorphism assumption ensures that the true latent semantic factors are recoverable from the observed modalities via a well-behaved inverse. This structural regularity is essential for identifiability analysis; without it, semantic recovery becomes ill-posed or approximate under stricter assumptions. Our use of this assumption follows established precedents in nonlinear ICA and causal representation learning [1,2,3], where assumptions on diffeomorphistic generative process are common to obtain rigurous identifiability guarantees.
>
>  - Possible relaxations: While assuming diffeomorphic generative mappings provides clean theoretical guarantees, we acknowledge that this assumption could be relaxed in future work. For instance, one may consider weaker conditions such as injectivity or explore models with discrete latent factors combined with measure-preserving or piecewise invertible mappings. However, relaxing the diffeomorphism assumption typically necessitates more advanced mathematical machinery and may require stronger inductive biases to maintain theoretical rigor. These relaxations offer promising directions but also increase complexity and may reduce generality.
>
> Importantly, our experiments show that deep models still perform well in settings where the diffeomorphism assumption does not strictly hold (e.g., on MPI3D-Complex dataset). This suggests the assumption serves as a sufficient but not necessary condition for recovery, common in many theory-driven studies.
>
> Thank you again for raising this point, and we will incorporate this discussion on both implications and possible  relaxations into the final version of the paper.
>
> [1] *Von Kügelgen, Julius, et al. "Self-supervised learning with data augmentations provably isolates content from style." NeurIPS 2021*.
> [2] *Brehmer, Johann, et al. "Weakly supervised causal representation learning." NeurIPS 2022*.
> [3] *Zhang, Kun, et al. "Causal representation learning from multiple distributions: a general setting." ICML 2024*.
>
> ---
> > **Q1** *"... two loss functions are introduced ... to clarify the rationale behind choosing different loss functions ..."*
>
> We appreciate your question and clarify that the use of two different loss functions, $\mathcal{L}_ {\small \mathrm{MMCL}}$ and $\mathcal{L}_ {\small \mathrm{SymAlignMaxEnt}}$, is intentional and corresponds to the distinct goals of the respective experiments:
>
> - In Section 5.1, we employ the loss $\mathcal{L} _ {\small \mathrm {SymAlignMaxEnt}}$ to validate our theoretical analysis, which is derived from $\mathcal{L}_{ \small \mathrm{MMCL}}$ under idealized conditions such as infinite observational data (see Page 3, line 92–93). This loss is designed for analytical tractability and reflects the formal identifiability guarantees of our theory (Page 3, line 98). As is standard in identifiability studies related to contrastive learning (Page 3, line 98), such idealized objectives provide a principled lens to rigurously analyze the effects of misalignment on multimodal representation learning.
>
> - In Sections 5.2 and 5.3, we turn to practical evaluation on realistic image-text datasets, where the canonical MMCL loss is adopted (Page 3, line 89). This loss, used in systems such as CLIP, ALIGN, reflects standard training objectives for multimodal contrastive learning and allows us to examine how our theoretical insights manifest in real-world scenarios.
>
> The core difference between the two losses lies in the geometry of the representation space, particularly in how they normalize and align semantic manifolds. Despite this difference, our theoretical framework targets block-wise semantic identifiability up to invertible transformations, rather than exact functional recovery, which remains compatible with both objectives.
>
> We will further clarify this design choice in the final version, and thank you for bringing attention to this important detail.
>
> ---
> > **Q2** *"In Figure 4 (page 7) ... unclear ... figure description on page 8 would greatly improve clarity ..."*
>
> Thank you! We will revise the caption for clarity in the final version.
>
> ---
> > **Q3** *"... all models are based on simple MLP ... whether the proposed theoretical insights hold under more complex models, such as transformer-based architectures ..."*
>
> We appreciate this thoughtful question. The identifiability guarantees in Theorem 4.1 are derived under general assumptions and are not tied to any specific architecture, as long as the model class is sufficiently expressive to minimize the MMCL objective.
>
> We deliberately use simple MLPs in our experiments to validate the theory under controlled conditions. MLPs are standard in identifiability studies, and in the context of our study, they allow us to isolate the effects of semantic misalignment without confounding factors introduced by architectural inductive biases. In contrast, more complex models (e.g., CNNs or Transformers) may benefit performance on practical datasets, but also involve heavier parameterization of complexity that may obscure the mechanisms under study—especially in limited-sample regimes.
>
> While our theory remains valid for such architectures, understanding how architectural inductive biases interact with misalignment dynamics is an important and orthogonal direction, and we acknowledge this as valuable future work.
>
> ---
> > **Q4** *"... text-specific variables are independent of semantic variables $s$, with “grammar” given as an example ... whether certain text-specific variables are inherently entangled with semantic ..."*
>
> We appreciate this thoughtful observation. The term *"grammar"* may have unintentionally overstated the independence between linguistic form and semantics. Our intention was to refer to text-specific variables that influence expression but are not visually grounded, such as stylistic phrasing or rhetorical choices. We agree that certain grammatical features—such as verb tense or modality—can carry semantic content, particularly in textual contexts.
>
> In our formulation, if such features contribute to visually grounded meaning, they are included as part of the shared semantic variable $\mathbf{s}$. If they do not correspond to observable visual attributes but are statistically correlated, they can be interpreted within our framework as a form of selection bias (with the modalities mirrored), i.e., information present in text but omitted in the image modality. This reflects a modeling simplification to keep the exposition concise, rather than a fundamental gap, as the asymmetry is already accounted for by our selection bias formalism.
>
> To prevent misinterpretation, we will revise the explanation of terminology in the final version, replacing “grammar” with “stylistic or expressive form”. We appreciate your careful reading.
>
> ---
> We hope our reply clarifies your concerns, and thank you again for your time.

---

> > ### Comment · Reviewer_t7nT · 2025-08-04
> > **Keep my score**
> >
> > Thank you for your detailed response. I will keep my current score.

---

> > > ### Author Response · Authors · 2025-08-05
> > >
> > > Thank you again for your valuable comments and engagement throughout the discussion. We sincerely appreciate the time you dedicated to reviewing our work.

---

### Official Review · Reviewer_HLeK · 2025-07-02

**Clarity:** 4
**Significance:** 3
**Originality:** 3
**Rating:** 4
**Confidence:** 2

**Summary:**

This paper proposes a framework for characterizing multimodal misalignment through a latent variable model for the underlying generative process of image and text. This model describes two types of potential misalignment: selection bias, where only a subset of the latent variables are used in generating text, and perturbation bias, where the ground truth semantic information is corrupted to generate erroneous text. Under the LVM framing, the paper demonstrates that multimodal contrastive learning recovers the unbiased semantic variables but excludes the misaligned components. Experiments on synthetic numerical data verify this theoretical finding, as well as experiments on a real-world dataset with seven mutually independent ground-truth factors and a synthetic dataset with causal relations between latent variables.

**Questions:**

Questions:
1. How might practitioners identify selection and perturbation biases in their dataset?
2. In a situation where the amount of bias varies per sample (such as because of differences in human annotations), how would this affect the representations learned?
3. When might misalignment be useful to capture synergistic interactions in multimodal data? For example, sarcasm is difficult to predict given unimodal data, but is easier given multimodal context, such as a video.

**Ethical Concerns:**

["NO or VERY MINOR ethics concerns only"]

**Final Justification:**

The authors have addressed my questions on the practical aspects of the work -- how it may be extended to modalities beyond vision and text, how practitioners may identify selection and perturbation bias, and how synergistic interactions between modalities that yield new information can be identified. I maintain my positive rating.

**Limitations:**

Yes.

**Paper Formatting Concerns:**

None.

**Quality:**

3

**Strengths And Weaknesses:**

Strengths
1. The problem of understanding when multimodal misalignment is desirable is important and novel.
2. The paper is well-written and the intuition behind the theoretical findings is well-explained.
3.Experiments on both simulated numerical data and real-world modalities support the theoretical findings.
Weaknesses
1. Experiments are limited to image and text.
2. In a real world setting, it is often unclear to what extent there is selection and perturbation bias. This work does not seem to address how one would quantify these biases, which would be relevant to practitioners.

---

> ### Author Rebuttal · Authors · 2025-07-28
>
> Thank you for reviewing our work and your feedback. We appreciate your recognition of our theoretical framing, clarity, and empirical validation. While some of your excellent questions extend beyond our current scope, we see them as exciting directions built on our foundation. We respond to each point below.
>
> ---
> > *"Experiments are limited to image and text."*
>
> We appreciate the suggestion. Our focus on image-text pairs is a deliberate and principled choice aimed at isolating the core phenomenon of interest: semantic misalignment in multimodal learning. Image-text is among the most widely studied modality pairings in large-scale representation learning (e.g., CLIP, ALIGN), and it offers practical advantages for analyzing semantic filtering—due to its scale, accessibility, and interpretability.
>
> Importantly, our framework is modality-agnostic and applies to any setting involving two views: one being a high-dimensional entangled observation (e.g., image), and the other a structured auxiliary modality that embeds human conceptual knowledge, such as text. This makes language a natural fit for modeling human-induced biases like semantic selection and perturbation.
>
> We select image-text pairs not for their exclusivity, but because they allow controlled, interpretable evaluations of theoretical predictions, using well-established datasets like MPI3D and Causal3DIdent. Other modalities (e.g., depth maps, audio data, tactile tensor grids) are compatible with our framework in principle, but currently may lack the structured datasets and semantic annotations necessary for rigorous, controllable analysis. Exploring these modalities is a valuable direction, though outside the current scope of our theory-driven study.
>
> ---
> > *"In a real world setting, it is often unclear to what extent there is selection and perturbation bias. This work does not seem to address how one would quantify these biases, which would be relevant to practitioners."*
>
> We appreciate your emphasis on this practical point. Our work is primarily focused on formalizing the impact of semantic misalignment on representation learning and identifiability, rather than on empirically quantifying dataset-specific bias. That said, we fully acknowledge that estimating the extent and structure of such biases (particularly in large-scale, web-curated datasets) is an important challenge for improving interpretability, data quality, and downstream robustness.
>
> We view this as a challenging but valuable problem, and a natural next step that builds on our theoretical contributions—representing a complementary direction beyond the current scope of this study.
>
> For preliminary directions, we refer to our response to your next question. We will incorporate a discussion of this topic in the final version.
>
> ---
> > **Q1** *"How might practitioners identify selection and perturbation biases in their dataset?"*
>
> This is an great and timely question. While our current work does not implement empirical detection mechanisms, our framework suggests several actionable directions for practitioners:
>
> - Selection Bias: Practitioners can begin by defining a target concept vocabulary (e.g., derived from ontology or human annotation) and computing the caption coverage rate for each concept across the dataset. Averaging coverage across categories may help reveal systematic underrepresentation or omission of certain semantic types.
>
> - Perturbation Bias: Since this bias involves semantic distortions, one practical approach is to generate reference captions using a strong, well-aligned image-to-text model (e.g., fine-tuned BLIP or GPT-4V), and compare them with existing captions. Metrics such as semantic similarity, factual consistency, or entity-level precision/recall can serve as approximate proxies for quantifying distortion or hallucination.
>
> These techniques involve design choices and approximations, but they offer a useful starting point for diagnosing selection and perturbation biases in real-world multimodal datasets—ultimately supporting more reliable and semantically grounded vision-language model training.
>
> ---
> > **Q2** *"In a situation where the amount of bias varies per sample (such as because of differences in human annotations), how would this affect the representations learned?"*
>
> We appreciate this insightful observation. The impact of instance-level bias variation depends on the type of misalignment:
>
> - Selection Bias: If the subset of semantic variables included in the text varies across samples, it introduces a missing-not-at-random (MNAR) pattern in the latent space, a well-known challenge in missing data theory, particularly in the context of latent semantic representations. In this case, the selection bias parameter $\theta$ becomes a sample-dependent random variable, potentially governed by annotation conventions, cultural norms, or attentional cues, especially in large-scale web data. Modeling this rigorously would require specifying a distribution over selection subsets, possibly incorporating structured priors, a challenging but important direction that we identify for future work. We explicitly acknowledge this limitation in Appendix H.1, and emphasize that our current theory provides a principled abstraction that can be extended to such sample-specific stochastic selection scenarios.
>
> - Perturbation Bias: Our framework already accommodates sample-level variation in perturbation. As noted in Assumption 3.2, we do not assume that all samples share the same perturbed semantics. Instead, we allow a random subset $A \subset \mathbb{I}_\rho$ with non-zero probability $p(A) > 0$, such that perturbations occur stochastically across the dataset. This formulation captures instance-level variation in perturbation patterns, and our identifiability result holds under this generalization.
>
> ---
> > **Q3** *"When might misalignment be useful to capture synergistic interactions in multimodal data? For example, sarcasm is difficult to predict given unimodal data, but is easier given multimodal context, such as a video."*
>
> Thank you for this excellent and thought-provoking question. It points to two distinct cases of multimodal synergy:
>
> - Deterministic cross-modal mappings:
> Some semantics that initially appear non-shared (e.g., facial expression $\rightarrow$ emotional tone)  can be deterministically inferred across modalities. Our framework naturally accommodates this case: deterministic cross-modal mappings (i.e., $z^{(2)} = f(z^{(1)})$) can simply be absorbed into either modality’s encoder, thus preserving semantic alignment. Hence, although we present a shared latent semantic space for conceptual clarity, our identifiability guarantees fully extend to these scenarios.
> - Emergent semantics requiring interaction-specific latent variables:
> In contrast, phenomena such as sarcasm, irony, or humor arise from non-additive, modality-interactive effects, e.g., the incongruity between neutral text and exaggerated prosody or facial expressions. Such semantics may not be inferable from a single modality alone, and explicitly modeling stable, interaction-specific latent variables becomes necessary. Importantly, we believe these multimodal interactions exhibit structured and recurring patterns rather than arbitrary noise, and our principled abstraction provides a foundational framework upon which future interaction-driven models can build.
>
> Thus, while our current framework does not directly model these interactions, by isolating invariant semantics under structured misalignment, it sets the stage for capturing emergent multimodal meanings. We see this as an exciting complementary direction and will incorporate this discussion into the final version.
>
> ---
> We hope our reply clarifies your concerns, and thank you again for your time.

---

> > ### Comment · Reviewer_HLeK · 2025-08-04
> >
> > Thank you for the detailed response. I appreciate the clarification and discussion on capturing synergistic interactions, and I will keep my positive evaluation.

---

> > > ### Author Response · Authors · 2025-08-05
> > >
> > > Thank you again for your feedback and for engaging with us during the discussion. We've really valued the interaction.

---

### Official Review · Reviewer_iYK6 · 2025-07-03

**Clarity:** 3
**Significance:** 2
**Originality:** 2
**Rating:** 4
**Confidence:** 4

**Summary:**

This paper analyzes how cross-modal misalignment affects multimodal contrastive learning (MMCL). A latent variable model with selection and perturbation biases is proposed to characterize misalignment, revealing which semantic factors remain identifiable. Theoretical results are validated on real and synthetic data, showing that moderate misalignment can enhance generalization by acting as a regularizer.

**Questions:**

1.	Can the proposed theory be extended beyond contrastive learning frameworks to settings like multimodal generation or joint classification? In particular, is it applicable to asymmetric tasks such as image-to-text generation?
2.	In practical scenarios, how can the strength of selection or perturbation bias be estimated? Are there learnable or calibratable metrics that can quantify the degree of misalignment?
3.	The paper suggests that misalignment can act as a regularizer, but what are the precise conditions under which this becomes beneficial? At what point does misalignment turn from helpful bias into detrimental noise suppression?
4.	Many existing works (e.g., CLIP with hard negative sampling or caption filtering) aim to mitigate misalignment. Can such techniques be theoretically or empirically integrated with this framework? Are they compatible in principle?
5.	Is it possible to design corrective mechanisms based on estimated selection or perturbation bias to improve text generation or data sampling? Could such targeted interventions form a promising direction for future work?

**Ethical Concerns:**

["NO or VERY MINOR ethics concerns only"]

**Final Justification:**

The authors’ response, including their experimental results and clarifications, has addressed part of my concerns. Therefore, I have decided to raise my score.

**Limitations:**

yes

**Quality:**

3

**Strengths And Weaknesses:**

S1.	The paper proposes a latent variable model with selection and perturbation biases to precisely characterize cross-modal misalignment, offering strong theoretical grounding.
S2.	The paper establishes a block-identifiability framework that reconciles seemingly conflicting views on whether misalignment is harmful or beneficial, providing practical guidance for real-world applications.
S3.	The paper validates the theory through comprehensive experiments on both synthetic and real-world datasets (e.g., MPI3D, Causal3DIdent), covering misalignment severity, variable dependencies, and downstream tasks.

W1.	The theoretical analysis is tightly coupled with the MMCL framework (contrastive learning + symmetric max-entropy loss), limiting its applicability to other paradigms such as generative modeling or multi-task learning.
W2.	While the modeling of selection and perturbation biases is representative, it does not account for more complex forms of misalignment such as temporal desynchronization, modality dropout, or entity mismatch.
W3.	The downstream tasks are simplified and mainly focus on semantic variable prediction or synthetic targets, lacking evaluation on real-world tasks like visual question answering, retrieval.
W4.	The paper asserts that perturbation bias does not structurally propagate, but it lacks a deeper analysis of how misalignment may affect causal pathways or downstream generalization in practical systems.
W5.	Implementation and training details are somewhat underreported; while the core theory is solid, missing descriptions of hyperparameters, baselines, and robustness tests limit reproducibility.

---

> ### Author Rebuttal · Authors · 2025-07-28
>
> We appreciate the thoughtful assessment and the recognition of our contributions to the theoretical depth, identifiability framing, and empirical validation. We address the remaining concerns and discussion points below.
>
> ---
> > **W1** *"The theoretical analysis is tightly coupled with the MMCL ... limiting ... other paradigms such as generative modeling or multi-task learning."*
>
> We would like to clarify that our study focuses on identifiability analysis under MMCL, as stated in the abstract, introduction, and theoretical sections. This choice is principled: MMCL with InfoNCE is a widely adopted paradigm in vision-language learning (e.g., CLIP), offering conceptual clarity and practical significance. Centering on a single objective is also consistent with prior work in identifiability theory, such as nonlinear ICA with likelihood-based modeling [1] and unimodal contrastive learning [2].
>
> While our framework is developed within MMCL, its insights into how semantic misalignment shapes representation learning remain broadly relevant. Applying these ideas to downstream tasks such as generation or multi-task learning would require integrating decoding mechanisms or task-specific objectives—directions we view as promising but beyond the current scope of this theory-focused work.
>
> [1] *Khemakhem, Ilyes, et al. "Variational autoencoders and nonlinear ica: A unifying framework." AISTATS, 2020*.
> [2] *Zimmermann, Roland S., et al. "Contrastive learning inverts the data generating process." ICML, 2021*.
>
> ---
> > **W2** *"While the modeling of selection and perturbation biases is representative, it does not account for ... misalignment such as temporal desynchronization, modality dropout, or entity mismatch."*
>
> We appreciate this observation and agree that multimodal data presents a variety of misalignment forms. Our study specifically focuses on semantic misalignment in image-text pairs, which is a setting of growing practical relevance and theoretical interest due to its prevalence in large-scale pretraining pipelines (e.g., CLIP, ALIGN). By modeling selection and perturbation biases, we aim to capture common patterns of omission and distortion in this widely used modality pair.
>
> We acknowledge that other types of misalignment, such as temporal desynchronization, modality dropout, or entity mismatch, are also important and may require different modeling assumptions and methodological tools. Our current abstraction serves as a focused and tractable step toward formalizing semantic misalignment. Extending the framework to encompass broader or more heterogeneous modalities is an exciting and valuable direction for future work.
>
> ---
> > **W3** *"... downstream tasks ... simplified ... focus on semantic variable prediction or synthetic targets, lacking evaluation on real-world tasks like visual question answering, retrieval."*
>
> We respectfully clarify that our downstream evaluations were deliberately designed to be controlled, interpretable, and causally grounded, in order to rigorously validate our theoretical predictions and isolate the effects of selection and perturbation biases. While VQA or retrieval can offer support, they also introduce extra supervision or inductive biases beyond our theory, which may confound interpretation of results. As emphasized in Corollary 4.1 and 4.2, our findings offer principled guidance for improving data quality regarding model versatility and robustness in practical multimodal systems. Extending the framework to complex downstream tasks is an important but orthogonal extension for future work.
>
> ---
> > **W4** *"... perturbation bias does not structurally propagate ... lacks deeper analysis of how misalignment may affect causal pathways or downstream generalization ..."*
>
> We appreciate your interest in the causal implications of misalignment. Our assumption that perturbation bias does not causally propagate (Assumption 3.2) is a deliberate and principled modeling choice. In practice, such perturbations often reflect textual distortions—e.g., omission or rewriting of certain attributes by annotators—without intervening on the visual content. This abstraction allows us to isolate the effects of semantic misalignment without committing to specific causal structures among latent variables.
>
> Regarding downstream generalization, we directly address this in Corollary 4.2. When biased semantics align with environment-sensitive factors, MMCL implicitly filters out these spurious signals, thereby enhancing generalization. We empirically validate this in Section 5.1: under domain shifts, domain-sensitive factors lead to degraded performance when preserved, but robustness improves when they are filtered (by intentional perturbation), as predicted by our theory.
>
> ---
> > **W5** *"... implementation and training details ... underreported ... missing hyperparameters, baselines, robustness tests ..."*
>
> We respectfully disagree with this assessment. Comprehensive implementation and training details are reported throughout the paper and appendices:
>  - Section 5 outlines the experimental setup, datasets, and evaluation procedures.
>  - Appendices D.1, E.1, and F.1 include full model specifications, training hyperparameters, and evaluation settings.
>  - All reported results are averaged over three random seeds, and standard deviations are included in performance plots to assess robustness.
>
> Since our focus is on validating theoretical claims under controlled conditions, benchmark comparisons are not applicable. However, we provide a fully reproducible anonymous codebase with all configurations and scripts for transparency.
>
> ---
> > **Q1** *"Can... extended beyond contrastive learning frameworks ... such as image-to-text generation?"*
>
> Yes, with caveats. While our work focuses on MMCL, the core principle, that semantic misalignment influences which representations are learned, applies beyond contrastive learning. Extending the theory to generative tasks (e.g., image-to-text) would require incorporating decoding mechanisms, which lies outside our current scope. Nevertheless, we believe the framework offers foundational insights into how alignment quality shapes learned semantics, and may inform future efforts toward improving fidelity and safety in generative settings.
>
> ---
> > **Q2** *"... how can ... bias be estimated? Are there... metrics ... quantify ... misalignment?"*
>
> Yes, with caveats. While our study does not implement bias estimation mechanisms, the proposed framework suggests plausible directions for approximating misalignment:
>  - Selection Bias: One can define a target concept vocabulary (e.g., based on human annotations or a domain ontology) and compute the frequency with which each concept appears in image captions. The aggregate coverage can serve as a proxy for semantic omission, though it depends on the quality and granularity of the chosen vocabulary.
>  - Perturbation Bias: This is more nuanced, as it involves distortions rather than omission. A possible approach is to generate reference captions using strong image-to-text models (e.g., fine-tuned BLIP or GPT-4V) and compare them to dataset captions using approximate semantic similarity or factual consistency metrics. While these metrics are only proxies and may themselves be noisy, they can help identify candidate distortions or inconsistencies at scale.
>
> ---
> > **Q3** *"... misalignment can act as a regularizer, ... under which this becomes beneficial? ... misalignment turn from helpful bias into detrimental noise suppression?"*
>
> This is a valuable question, and we address it formally in Corollary 4.2. Misalignment becomes beneficial when it systematically affects environment-sensitive (i.e., non-robust or domain-specific) semantics, allowing contrastive learning to filter out unstable cues and focus on domain-invariant features, thus acting as an implicit regularizer.
>
> Importantly, the key factor is which semantics are biased, not how strongly. Mild omission of task-relevant features can degrade performance, while even strong distortion may help if it targets spurious factors. The tipping point lies in the semantic identity of the biased subset, not the absolute noise level.
>
> ---
> > **Q4** *"Can ... CLIP with hard negative sampling or caption filtering ... integrated with this framework ... compatible in principle?"*
>
> Yes, both techniques are compatible with our framework and align with the insights in Corollary 4.1. Caption filtering directly reduces selection and perturbation bias by removing imcomplete or distorted captions, improving semantic alignment across modalities. Hard negative sampling, while not addressing misalignment per se, enhances semantic discrimination and complements alignment by refining decision boundaries. Together, these practices support the theoretical predictions of our model and can be viewed as practical extensions.
>
> ---
> > **Q5** *"Is it possible to design corrective mechanisms ... to improve text generation or data sampling? ... promising direction for future work?"*
>
> Yes, and we agree this is a highly promising direction. Since misalignment introduces structured inductive biases, learning corrective mechanisms that estimate and mitigate these patterns can meaningfully improve data quality and downstream performance. One practical approach is to use high-quality image-to-text models (e.g., BLIP, GPT-4V) to generate reference captions and compare them with existing captions. Semantic divergence between the two can serve as a proxy signal for selection or perturbation bias. While current models may underperform on rare or subjective concepts, human-in-the-loop feedback or filtering can further enhance correction quality. We believe semi-automatic pipelines for mitigating misalignment will become increasingly feasible as tools improve.
>
> ---
> We thank you again for the thoughtful feedback. Points like Q3–Q5 point to valuable extensions beyond our current scope, and we will incorporate relevant discussion in the final version.

---

> > ### Comment · Reviewer_iYK6 · 2025-08-05
> >
> > Thank you for your detailed response. I will consider raising my score.

---

> > > ### Author Response · Authors · 2025-08-05
> > >
> > > Thank you for your consideration and timely feedback. We greatly appreciate your openness to raising the score and would be happy to clarify any remaining concerns. Thank you again for your time and effort in reviewing our work!

---

### Official Review · Reviewer_1M1L · 2025-07-03

**Clarity:** 3
**Significance:** 3
**Originality:** 3
**Rating:** 5
**Confidence:** 3

**Summary:**

This paper provides a theoretical framework to understand the impact of cross-modal misalignment in multimodal contrastive learning (MMCL), focusing on image-text pairs. The authors identify two opposing views in existing literature: one that treats misalignment as noise to be filtered, and another that suggests it can be beneficial for learning robust representations.
To reconcile these views, the paper introduces a latent variable model (LVM) that formalizes misalignment through two mechanisms affecting the text modality, namely selection bias and perturbation bias.

The main theoretical contribution is a proof that MMCL, without any modification, inherently learns representations that are "block-identifiable" with the subset of semantic variables that are invariant to these biases. In other words, the model learns to represent only the semantic content that is consistently and accurately shared across both modalities, while systematically discarding misaligned (omitted or perturbed) information and modality-specific noise.

The authors then derive two key practical insights from this result: (1) To build comprehensive foundation models, one must minimize misalignment by creating detailed and faithful captions. (2) For tasks requiring out-of-distribution (OOD) generalization, misalignment can be strategically introduced to filter out spurious or environment-sensitive variables, thereby improving robustness. These theoretical claims are validated through extensive experiments on a synthetic dataset, the MPI3D-Complex dataset, and the Causal3DIdent dataset, showing that the model reliably identifies the unbiased, shared semantic factors.

**Questions:**

- The entire theoretical framework rests on the generative functions being smoothly invertible. However, this is rarely true in practice. It's seemingly also violated in the paper's own experiments (e.g. Causal3DIdent). If the theory holds even when its assumptions are violated, does this suggest the theorem is not capturing the essential mechanism at play?

- The proof relies on Assumption 3.2, where textual perturbations do not propagate to causally dependent variables. Given that real language often has its own internal causal logic, how do you justify this simplifying assumption, and how would your identifiability guarantees degrade if it were violated?

**Ethical Concerns:**

["NO or VERY MINOR ethics concerns only"]

**Final Justification:**

As written in my original review and the follow-up response, while I still have reserved opinions about the applicability of the theorems due to its assumptions, I acknowledge that this work makes interesting and novel contributions. I would therefore keep my rating.

**Limitations:**

yes

**Paper Formatting Concerns:**

None.

**Quality:**

3

**Strengths And Weaknesses:**

Strengths:
- **Timely Unification**: The paper tackles a crucial and confusing issue in multimodal learning—the dual role of misalignment—and offers a clear, unifying theoretical perspective. The ability to explain why misalignment can be both harmful and beneficial depending on the end goal is a significant conceptual contribution.

- **Strong Theoretical Result**: The core identifiability theorem (Thm. 4.1) is powerful. Demonstrating that a standard learning objective like the InfoNCE loss implicitly performs this sophisticated semantic filtering is a non-obvious and important finding. The proof that this holds regardless of the latent causal structure is particularly strong.

- **Clear Practical Implications**: The paper successfully bridges theory and practice by providing actionable advice for practitioners. The corollaries on pretraining vs. robust OOD learning give clear guidance on data curation strategies, which is highly valuable for the community.

- **Thorough Empirical Validation**: The experiments are well-designed and systematically test the paper's claims. The use of three different datasets provides convincing evidence for the theoretical results. The ablations further strengthen the conclusions.

Weaknesses:
- **Oversimplified Model of Misalignment**: While the formalization of selection and perturbation bias is a strength, it represents a highly structured form of noise. Misalignment in large, web-scraped datasets (e.g., LAION) is likely to be far more chaotic, containing unstructured factual errors, complex paraphrasing, and ambiguous references that don't fit neatly into the proposed categories. The experiments, which use template-based text, also do not capture this complexity.

---

> ### Author Rebuttal · Authors · 2025-07-27
>
> Thank you for your thoughtful review. We appreciate your recognition of our theoretical contributions, practical implications, and empirical validation. Below, we respond to your insightful questions and further clarify key modeling choices and assumptions raised in your comments.
>
> ---
> #### **Oversimplified Model of Misalignment**
> > *"While the formalization of selection and perturbation bias is a strength, it represents a highly structured form of noise. Misalignment in large, web-scraped datasets (e.g., LAION) is likely to be far more chaotic, containing unstructured factual errors, complex paraphrasing, and ambiguous references that don't fit neatly into the proposed categories."*
>
> We agree that real-world misalignment is diverse and often unstructured, particularly in large-scale web-scraped corpora. However, our goal is to provide a principled and analytically tractable abstraction that captures the two dominant sources of semantic misalignment in vision-language data: (1) omission of relevant semantics (modeled as selection bias), and (2) inclusion of distorted or stochastically perturbed semantics (modeled as perturbation bias).
>
> These two categories subsume a broad spectrum of real-world issues, including ambiguous references (which omit disambiguating context $\rightarrow$ selection bias) and unstructured, random factual errors ($\rightarrow$ perturbation bias). As for linguistic-side distortions such as complex paraphrasing, we treat them as modality-specific factors (e.g., stylistic variation, denoted $\mathbf{m}_t$ in our formulation). We also explicitly acknowledge this modeling limitation in Appendix H.1.
>
> Crucially, this abstraction allows for formal reasoning about how such misalignment affects representation learning, which would be intractable under unconstrained or ad hoc noise models. As such, we view our framework as offering first-order, foundational insights into the effect of semantic misalignment rather than a full taxonomy of noise processes. Extending the framework to incorporate richer or data-driven misalignment structures remains a valuable and promising future direction.
>
> ---
> > *"The experiments, which use template-based text, also do not capture this complexity."*
>
> Our original experiments were designed to validate the core theoretical claims under controlled settings. To further assess the practical relevance of our theory on real-world, large-scale data (and to also address Reviewer mroJ’s related concern), we include a case study using OpenCLIP trained on LAION-400M. Specifically, we evaluate zero-shot image-to-text matching across 146 visual concepts, grouped into 15 semantic categories (e.g., color, object). For each concept, we collected up to 200 CC-licensed Flickr images and estimated the frequency of their appearance in LAION-400M captions as a proxy for selection bias.
>
> We report the average F1 scores using OpenCLIP ViT-B/32 and ViT-L/14:
>
> ||Color|Object|Clothing|Food|Vehicle|Scene|Weather|Animal|Role|Texture|Trait|Viewpoint|Emotion|Postprocess|Stereotype|
> |-|-|-|-|-|-|-|-|-|-|-|-|-|-|-|-|
> |Average Coverage (%)|2.1622|1.6288|1.1747|0.8140|0.6114|0.5365|0.5284|0.4682|0.2147|0.1470|0.0935|0.0312|0.0238|0.0118|0.0003|
> |ViT-B/32 F1 Score (%)|81.1|87.7|84.6|86.6|90.0|97.0|76.3|97.6|69.2|33.2|21.2|29.4|19.5|41.8|34.8|
> |ViT-L/14 F1 Score (%)|79.0|89.5|86.6|91.0|90.1|94.6|75.9|98.9|66.4|37.1|19.4|31.0|20.1|44.1|40.2|
>
> The results show that visual concepts with extremely low caption coverage yield dramatically worse F1 scores, aligning with our theoretical prediction. This suggests that our theoretical framework remains predictive even under realistic misalignment in large-scale models. We will include this study in the final version to strengthen the empirical grounding of our work.
>
> ---
> #### **Smooth Invertibility Assumption**
> > *"The entire theoretical framework rests on the generative functions being smoothly invertible. However, this is rarely true in practice. It's seemingly also violated in the paper's own experiments (e.g. Causal3DIdent)."*
>
> It is correct that our theory assumes smooth invertibility, which may not strictly hold in practice. Nevertheless, this assumption is standard in identifiability theory [1–3], enabling formal guarantees under well-defined conditions. Without it, identifiability is generally unprovable or requires stronger assumptions on other aspects.
>
> That said, empirical robustness beyond this assumption is often observed. For example, even in discrete settings with non-injective mappings, such as Bernoulli observations from logistic models, approximate identifiability can still emerge in practice [4]. We are aware of such results and acknowledge them as empirical support for our findings, especially in settings like Causal3DIdent, where our theory remains predictive despite relaxed assumptions.
>
> [1] *Von Kügelgen, Julius, et al. "Self-supervised learning with data augmentations provably isolates content from style."   NeurIPS 2021*.
> [2] *Brehmer, Johann, et al. "Weakly supervised causal representation learning." NeurIPS 2022*.
> [3] *Zhang, Kun, et al. "Causal representation learning from multiple distributions: a general setting." ICML 2024*.
> [4] *Khemakhem, Ilyes, et al. "Variational autoencoders and nonlinear ica: A unifying framework." AISTATS, 2020*.
>
> ---
> > *"If the theory holds even when its assumptions are violated, does this suggest the theorem is not capturing the essential mechanism at play?"*
>
> No, this does not imply that the theory fails to capture the essential mechanism. Rather, it highlights that the mechanism identified, namely, alignment on unbiased shared semantics via contrastive learning, holds beyond the idealized conditions needed for proof. The theory provides a principled explanation for this mechanism and clarifies when and why it should be expected to hold, even if real-world success extends further than the formal boundary.
>
> ---
> #### **Assumption 3.2**
> > *"The proof relies on Assumption 3.2, where textual perturbations do not propagate to causally dependent variables. Given that real language often has its own internal causal logic, how do you justify this simplifying assumption?"*
>
> Assumption 3.2 posits that textual perturbations do not causally propagate to other latent variables. We recognize that natural language can exhibit internal causal structure. However, this assumption is motivated by two practical considerations grounded in how image captions are produced:
>
> - Intentional perturbation of unwanted semantics (e.g., due to sensitive or ethical concerns). In some settings, annotators may deliberately perturb certain sensitive factors, such as age or religion. These perturbations target specific factors, even if those factors are causally related to others in the observational modality. For example, although age may correlate with clothing color in real-world photos, perturbing age in captions does not imply that clothing color must also be modified. This supports our assumption that misalignment mechanisms can be selectively applied without inducing causal propagation.
>
> - Human perceptual limitations. Annotators often aim to produce accurate and detailed captions, but due to cognitive or attentional constraints, they may inadvertently perturb certain semantic concepts. These perturbations often follow patterns—for example, certain semantic factors (like material and shape) may be jointly perturbed more frequently than others, forming entangled perturbation patterns. In such cases, while semantic entanglement exists within the perturbable set, the perturbation pattern does not necessarily follow the causal laws of real observations, but instead reflects human cognitive nuances. Crucially, such patterns do not causally propagate to other semantics. This fits naturally into our modeling: the range of the perturbable subset can be interpreted as a restricted subset of the selected semantic powerset $\mathcal{P} _ {\text{proper}}(\mathbb{I} _ \theta)$, capturing typical co-perturbation patterns without requiring assumptions on the latent causal pathways underlying observational data.
>
> In both cases, Assumption 3.2, grounded in practical rationale, provides a pragmatic abstraction that enables us to analyze recoverable semantics without relying on structural assumptions about the latent causal graph or modeling human cognitive nuances.
>
> ---
> > *"... and how would your identifiability guarantees degrade if it (i.e., Assumption 2) were violated?"*
>
> If perturbations were allowed to causally propagate across latent semantic factors, i.e., like those interventional settings, the distinction between biased and unbiased semantics would become less sharply defined, as the injected bias could indirectly affect additional, descendent variables (and those descendent variables also become biased due to causal propogation). In this scenario, the learned representation may no longer cleanly recover the truly unbiased semantics, but instead reflect a blurred or partially contaminated mixture, depending on how far the perturbation propagates, due to we can not determine the intrinsic representation dimensionality without knowing the true latent causal strucutre. Thus, formal guarantees would require strong assumptions on the latent causal graph and the dynamics of perturbation spread (whether certain latent factor is dominated by its exogenous variable), both of which would substantially increase theoretical complexity and reduce the generality of the model. By contrast, our current abstraction deliberately avoids these commitments, allowing us to make latent-structure-agnostic, tractable statements about semantic recovery.
>
> ---
> We hope our reply clarifies your concerns, and thank you again for your time.

---

> > ### Comment · Reviewer_1M1L · 2025-08-04
> > **Thanks**
> >
> > I would like to thank the authors for their comprehensive response. Although I still have reserved opinions on the simplified assumptions, I do acknowledge that many things would not be discussable theoretically if all these assumptions do not hold. I guess in general I am a bit anti-theory :) Nevertheless, I recognize the value of this work and would keep my positive rating. I wish good luck to the authors' submission.

---

> > > ### Author Response · Authors · 2025-08-05
> > >
> > > We appreciate your time and the constructive feedback, as well as your responsiveness during the discussion. Thank you again for reviewing our work!

---

### Note · Authors · 2025-08-11

We thank the reviewers for their constructive feedback and engagement.

---
There is broad consensus that our work makes a **novel and significant theoretical contribution** to understanding cross-modal misalignment. Reviewers highlighted our **unified theoretical framing** that reconciles opposing views and explains when misalignment is harmful versus beneficial (`1M1L`, `iYK6`, `t7nT`). The core identifiability theorem was described as "powerful" and **non-obvious**, offering principled insight into which semantics are recoverable under bias (`1M1L`, `t7nT`). Our work was commended for its **practical guidance** grounded in theory (`1M1L`, `iYK6`), **clarity of exposition** with well-motivated intuition and effective illustrations (`HLeK`, `t7nT`), and **extensive, well-designed experiments** with reproducible code (`1M1L`, `HLeK`, `t7nT`, `mroJ`).

---
During the rebuttal phase, we:
- **Clarified scope** (`iYK6`) – intentionally focus on MMCL for conceptual clarity and rigor, with extensions to generative/multi-task paradigms left for future work.

- **Discussed complex misalignment types** (`iYK6`) – clarified selection/perturbation bias as a principled, extensible abstraction, and outlined directions for extending to other cases.

- **Illustrated downstream relevance** (`mroJ`, `1M1L`) – added a large-scale OpenCLIP–LAION-400M case study showing that low-caption-coverage concepts have lower retrieval F1, consistent with theory.

- **Justified assumptions** (`1M1L`, `t7nT`) –  explained smooth invertibility as a standard, sufficient condition in identifiability theory, discussed possible relaxations, and observed robustness when not strictly met.

- **Linked hallucination** (`t7nT`) – analyzed as a symptom of low-fidelity captions during multimodal pretraining, connected to Corollary 4.1 on semantic fidelity.

- **Clarified bias estimation** (`HLeK`) – suggested practical methods to estimate selection and perturbation bias, to guide future empirical work.

- **Will improve clarity** (`t7nT`, `HLeK`) –  refine figure captions, relevant discussion, and loss-function rationale in the final version.

---

### Decision · Program_Chairs · 2025-09-17

**Decision:**

Accept (spotlight)

**Comment:**

This paper introduces a theoretical framework for cross-modal misalignment in multimodal contrastive learning, formalizing it as selection and perturbation biases. Reviewers agreed the work is novel and impactful, with clear theory and supportive experiments, further strengthened by the additional OpenCLIP–LAION case study in rebuttal. While concerns remain about simplifying assumptions and broader empirical validation, the reviewers found the clarifications convincing. Overall, the paper makes a meaningful theoretical contribution and is recommended for acceptance.